# Provable Guarantees for Neural Networks via Gradient Feature Learning

**Zhenmei Shi**[*]**, Junyi Wei**[*]**, Yingyu Liang**
University of Wisconsin, Madison
`zhmeishi@cs.wisc.edu,jwei53@wisc.edu,yliang@cs.wisc.edu`

## Abstract

Neural networks have achieved remarkable empirical performance, while the current theoretical analysis is not adequate for understanding their success, e.g., the Neural Tangent Kernel approach fails to capture their key feature learning ability, while recent analyses on feature learning are typically problem-specific. This work proposes a unified analysis framework for two-layer networks trained by gradient descent. The framework is centered around the principle of feature learning from gradients, and its effectiveness is demonstrated by applications in several prototypical problems such as mixtures of Gaussians and parity functions. The framework also sheds light on interesting network learning phenomena such as feature learning beyond kernels and the lottery ticket hypothesis.

## 1 Introduction

Neural network (NN) learning has achieved remarkable empirical success and has been a main driving force for the recent progress in machine learning and artificial intelligence. On the other hand, theoretical understandings significantly lag behind. Traditional analysis approaches are not adequate due to the overparameterization of practical networks and the non-convex optimization in the training via gradient descent. One line of work (e.g. [9, 31, 38, 60, 71, 123] and many others) shows under proper conditions, heavily overparameterized networks are approximately linear models over data-independent features, i.e., a linear function on the Neural Tangent Kernel (NTK). While making weak assumptions about the data and thus applicable to various settings, this approach requires the network learning to be approximately using fixed data-independent features (i.e., the kernel regime, or fixed feature methods). It thus fails to capture the feature learning ability of networks (i.e., to learn a feature mapping for the inputs which allow accurate prediction), which is widely believed to be the key factor to their empirical success in many applications (e.g., [54, 77, 117, 119]). To study feature learning in networks, a recent line of work (e.g. [5, 6, 14, 33, 52, 72, 76, 116] and others) shows examples where networks provably enjoy advantages over fixed feature methods (including NTK), under different settings and assumptions. While providing more insights, these studies typically focus on specific problems, and their analyses exploit the specific properties of the problems and appear to be unrelated to each other. *Is there a common principle for feature learning in networks via gradient descent? Is there a unified analysis framework that can clarify the principle and also lead to provable error guarantees for prototypical problem settings?*

In this work, we take a step toward this goal by proposing a gradient feature learning framework for analyzing two-layer network learning by gradient descent. (1) The framework makes essentially no assumption about the data distribution and can be applied to various problems. Furthermore, it is centered around features from gradients, clearly illustrating how gradient descent leads to feature learning in networks and subsequently accurate predictions. (2) It leads to error guarantees competitive with the optimal in a family of networks that use the features induced by gradients on the

---

[*]Equal contribution.

37th Conference on Neural Information Processing Systems (NeurIPS 2023).

data distribution. Then for a specific problem with structured data distributions, if the optimal in the induced family is small, the framework gives a small error guarantee.

We then apply the framework to several prototypical problems: mixtures of Gaussians, parity functions, linear data, and multiple-index models. These have been used for studying network learning (in particular, for the feature learning ability), but with different and seemingly unrelated analyses. In contrast, straightforward applications of our framework give small error guarantees, where the main effort is to compute the optimal in the induced family. Furthermore, in some cases, such as parities, we can handle more general data distributions than in the existing work.

Finally, we also demonstrate that the framework sheds light on several interesting network learning phenomena or implications such as feature learning beyond the kernel regime, lottery ticket hypothesis (LTH), simplicity bias, learning over different data distributions, and new perspectives about roadmaps forward. Due to space limitations, we present implications about features beyond the kernel regime and LTH in the main body but defer the other implications in Appendix C with a brief here. (1) For simplicity bias, it is generally believed that the optimization has some *implicit regularization* effect that restricts learning dynamics to a low capacity subset of the whole hypothesis class, so can lead to good generalization [53, 90]. Our framework provides an explanation that the learning first learns simpler functions and then more sophisticated ones. (2) For learning over different data distributions, we provide data-dependent non-vacuous guarantees, as our framework can be viewed as using the optimal gradient-induced NN to measure or quantify the "complexity" of the problem. For easier problems, this quantity is smaller, and our framework can give a better error bound to derive guarantees. (3) For new perspectives about roadmaps forward, our framework suggests the strong representation power of NN is actually the key to successful learning, while traditional ones suggest strong representation power leads to vacuous generalization bounds [19, 33]. Thus, we suggest a different analysis road. Traditional analysis typically first reasons about the optimal based on the whole function class then analyzes how NN learns proper features and reaches the optimal. In contrast, our framework defines feature family first, and then reasons about the optimal based on it.

## 2 Related Work

**Neural Networks Learning Analysis.** Recently there has been an increasing interest in the analysis of network learning. One line of work connects the sufficiently over-parameterized neural network to linear methods around its initialization like NTK (e.g. [9, 11, 20, 21, 31, 38, 49, 60, 62, 69, 71, 78, 82, 91, 93, 95, 114, 121, 122] and more), so that the neural network training is a convex problem. The key idea is that it suffices to consider the first-order Tyler expansion of the neural network around the origin when the initialization is large enough. However, NTK lies in the lazy training (kernel) regime that excludes feature learning [29, 50, 68, 113]. Many studies (e.g. [2, 5, 6, 8, 12, 14, 22, 26, 33, 37, 51, 52, 57, 58, 70, 72, 73, 76, 99, 112, 115, 116] and more) show that neural networks take advantage over NTK empirically and theoretically. Another line of work is the mean-field (MF) analysis of neural networks (e.g. [27, 28, 36, 79, 80, 100, 106] and more). The insight is to see the training dynamics of a sufficiently large-width neural network as a PDE. It uses a smaller initialization than the NTK so that the parameters may move away from the initialization. However, the MF does not provide explicit convergence rates and requires an unrealistically large width of the neural network. One more line of work is neural networks max-margin analysis (e.g. [30, 47, 48, 56, 61, 63, 74, 75, 83, 85, 86, 107, 109] and more). They need a strong assumption that the convergence starts from weights having perfect training accuracy, while feature learning happens in the early stage of training. To explain the success of neural networks beyond the limitation mentioned above, some work introduces the low intrinsic dimension of data distributions [17, 18, 23, 24, 25, 44, 67, 104, 108, 124]. Another recent line of work is that a trained network can exactly recover the ground truth or optimal solution or teacher network [3, 4, 10, 39, 84, 87, 94, 96, 120], but they have strong assumptions on data distribution or model structure, e.g., Gaussian marginals. [1, 40, 55, 110, 111] show that training dynamics of neural networks have multiple phases, e.g., feature learning at the beginning, and then dynamics in convex optimization which requires proxy convexity [43] or PL condition [65] or special data structure.

**Feature Learning Based on Gradient Analysis.** A recent line of work is studying how features emerge from the gradient. [7, 46] consider linear separable data and show that the first few gradient steps can learn good features, and the later steps learn a good network on neurons with these features. [33, 45, 105] have similar conclusions on non-linear data (e.g., parity functions), while in their problems one feature is sufficient for accurate prediction (i.e., single-index data model).

[32] considers multiple-index with low-degree polynomials as labeling functions and shows that a one-step gradient update can learn multiple features that lead to accurate prediction. [13, 81] studies one gradient step feature improvements at different learning rates. [97] proposes Recursive Feature Machines to show the mechanism of recursively feature learning but without giving a final loss guarantee. These studies consider specific problems and exploit properties of the data to analyze the gradient delicately, while our work provides a general framework applicable to different problems.

# 3 Gradient Feature Learning Framework

**Problem Setup.** We denote $[n] := \{1, 2, \ldots, n\}$ and $\tilde{O}(\cdot), \tilde{\Theta}(\cdot), \tilde{\Omega}(\cdot)$ to omit the $\log$ term inside. Let $\mathcal{X} \subseteq \mathbb{R}^d$ denote the input space, $\mathcal{Y} \subseteq \mathbb{R}$ the label space. Let $\mathcal{D}$ be an arbitrary data distribution over $\mathcal{X} \times \mathcal{Y}$. Denote the class of two-layer networks with $m$ neurons as:

$$\mathcal{F}_{d,m} := \left\{ f_{(\mathbf{a},\mathbf{W},\mathbf{b})} \mid f_{(\mathbf{a},\mathbf{W},\mathbf{b})}(\mathbf{x}) := \mathbf{a}^\top \left[ \sigma(\mathbf{W}^\top \mathbf{x} - \mathbf{b}) \right] = \sum_{i \in [m]} \mathbf{a}_i \left[ \sigma(\langle \mathbf{w}_i, \mathbf{x} \rangle - \mathbf{b}_i) \right] \right\}, \quad (1)$$

where $\sigma(z) = \max(z, 0)$ is the ReLU activation function, $\mathbf{a} \in \mathbb{R}^m$ is the second layer weight, $\mathbf{W} \in \mathbb{R}^{d \times m}$ is the first layer weight, $\mathbf{w}_i$ is the $i$-th column of $\mathbf{W}$ (i.e., the weight for the $i$-th neuron), and $\mathbf{b} \in \mathbb{R}^m$ is the bias for the neurons. For technical simplicity, we only train $\mathbf{a}, \mathbf{W}$ but not $\mathbf{b}$. Let superscript $(t)$ denote the time step, e.g., $f_{(\mathbf{a}^{(t)}, \mathbf{W}^{(t)}, \mathbf{b})}$ denote the network at time step $t$. Denote $\Xi := (\mathbf{a}, \mathbf{W}, \mathbf{b})$, $\Xi^{(t)} := (\mathbf{a}^{(t)}, \mathbf{W}^{(t)}, \mathbf{b})$. The goal of neural network learning is to minimize the expected risk, i.e., $\mathcal{L}_\mathcal{D}(f) := \mathbb{E}_{(\mathbf{x},y) \sim \mathcal{D}} \mathcal{L}_{(\mathbf{x},y)}(f)$, where $\mathcal{L}_{(\mathbf{x},y)}(f) = \ell(y f(\mathbf{x}))$ is the loss on an example $(\mathbf{x}, y)$ for some loss function $\ell(\cdot)$, e.g., the hinge loss $\ell(z) = \max\{0, 1 - z\}$, and the logistic loss $\ell(z) = \log[1 + \exp(-z)]$. We also consider $\ell_2$ regularization. The regularized loss with regularization coefficient $\lambda$ is $\mathcal{L}_\mathcal{D}^\lambda(f) := \mathcal{L}_\mathcal{D}(f) + \frac{\lambda}{2}(\|\mathbf{W}\|_F^2 + \|\mathbf{a}\|_2^2)$. Given a training set with $n$ i.i.d. samples $\mathcal{Z} = \{(\mathbf{x}^{(l)}, y^{(l)})\}_{l \in [n]}$ from $\mathcal{D}$, the empirical risk and its regularized version are:

$$\widetilde{\mathcal{L}}_\mathcal{Z}(f) := \frac{1}{n} \sum_{l \in [n]} \mathcal{L}_{(\mathbf{x}^{(l)}, y^{(l)})}(f), \qquad \widetilde{\mathcal{L}}_\mathcal{Z}^\lambda(f) := \widetilde{\mathcal{L}}_\mathcal{Z}(f) + \frac{\lambda}{2}(\|\mathbf{W}\|_F^2 + \|\mathbf{a}\|_2^2). \quad (2)$$

Then the training process is summarized in Algorithm 1.

---

**Algorithm 1** Network Training via Gradient Descent

---

Initialize $(\mathbf{a}^{(0)}, \mathbf{W}^{(0)}, \mathbf{b})$
**for** $t = 1$ **to** $T$ **do**
  Sample $\mathcal{Z}^{(t-1)} \sim \mathcal{D}^n$
  $\mathbf{a}^{(t)} = \mathbf{a}^{(t-1)} - \eta^{(t)} \nabla_\mathbf{a} \widetilde{\mathcal{L}}_{\mathcal{Z}^{(t-1)}}^{\lambda^{(t)}}(f_{\Xi^{(t-1)}}), \quad \mathbf{W}^{(t)} = \mathbf{W}^{(t-1)} - \eta^{(t)} \nabla_\mathbf{W} \widetilde{\mathcal{L}}_{\mathcal{Z}^{(t-1)}}^{\lambda^{(t)}}(f_{\Xi^{(t-1)}})$
**end for**

---

In the whole paper, we need some natural assumptions about the data and the loss.

**Assumption 3.1.** *We assume $\mathbb{E}[\|\mathbf{x}\|_2] \leq B_{x1}$, $\mathbb{E}[\|\mathbf{x}\|_2^2] \leq B_{x2}$, $\|\mathbf{x}\|_2 \leq B_x$ and for any label $y$, we have $|y| \leq 1$. We assume the loss function $\ell(\cdot)$ is a 1-Lipschitz convex decreasing function, normalized $\ell(0) = 1$, $|\ell'(0)| = \Theta(1)$, and $\ell(\infty) = 0$.*

**Remark 3.2.** *The above are natural assumptions. Most input distributions have the bounded norms required, and the typical binary classification $\mathcal{Y} = \{\pm 1\}$ satisfies the requirement. Also, the most popular loss functions satisfy the assumption, e.g., the hinge loss and logistic loss.*

## 3.1 Warm Up: A Simple Setting with Frozen First Layer

To illustrate some high-level intuition, we first consider a simple setting where the first layer is frozen after one gradient update, i.e., no updates to $\mathbf{W}$ for $t \geq 2$ in Algorithm 1.

The first idea of our framework is to provide guarantees compared to the optimal in a family of networks. Here let us consider networks with specific weights for the first layer:

**Definition 3.3.** *For some fixed $\mathbf{W} \in \mathbb{R}^{d \times m}, \mathbf{b} \in \mathbb{R}^d$, and a parameter $B_{a2}$, consider the following family of networks $\mathcal{F}_{\mathbf{W},\mathbf{b},B_{a2}}$, and the optimal approximation network loss in this family:*

$$\mathcal{F}_{\mathbf{W},\mathbf{b},B_{a2}} := \left\{ f_{(\mathbf{a},\mathbf{W},\mathbf{b})} \in \mathcal{F}_{d,m} \mid \|\mathbf{a}\|_2 \leq B_{a2} \right\}, \qquad \mathrm{OPT}_{\mathbf{W},\mathbf{b},B_{a2}} := \min_{f \in \mathcal{F}_{\mathbf{W},\mathbf{b},B_{a2}}} \mathcal{L}_\mathcal{D}(f). \quad (3)$$

The second idea is to compare to networks using features from gradient descent. As an illustrative example, we now provide guarantees compared to networks with first layer weights $\mathbf{W}^{(1)}$ (i.e., the weights after the first gradient step):

> **Theorem 3.4** (Simple Setting). *Assume $\widetilde{\mathcal{L}}_{\mathcal{Z}}\left(f_{(\mathbf{a},\mathbf{W}^{(1)},\mathbf{b})}\right)$ is L-smooth to $\mathbf{a}$. Let $\eta^{(t)} = \frac{1}{L}, \lambda^{(t)} = 0$, for all $t \in \{2, 3, \ldots, T\}$. Training by Algorithm 1 with no updates for the first layer after the first gradient step, w.h.p., there exists $t \in [T]$ such that*
> $$\mathcal{L}_{\mathcal{D}}(f_{(\mathbf{a}^{(t)},\mathbf{W}^{(1)},\mathbf{b})}) \leq \mathrm{OPT}_{\mathbf{W}^{(1)},\mathbf{b},B_{a2}} + O\left(\frac{L(\|\mathbf{a}^{(1)}\|_2^2 + B_{a2}^2)}{T} + \sqrt{\frac{B_{a2}^2(\|\mathbf{W}^{(1)}\|_F^2 B_x^2 + \|\mathbf{b}\|_2^2)}{n}}\right).$$

Intuitively, the theorem shows that if the weight $\mathbf{W}^{(1)}$ after a one-step gradient gives a good set of neurons in the sense that there exists a classifier on top of these neurons with low loss, then the network will learn to approximate this good classifier and achieve low loss. The proof is based on standard convex optimization and the Rademacher complexity (details in Appendix D.1).

Such an approach, while simple, has been used to obtain interesting results on network learning in existing work, which shows that $\mathbf{W}^{(1)}$ can indeed give good neurons due to the structure of the special problems considered (e.g., parities on uniform inputs [15], or polynomials on a subspace [32]). However, it is unclear whether such intuition can still yield useful guarantees for other problems. So, for our purpose of building a general framework covering more prototypical problems, the challenge is what features from gradient descent should be considered so that the family of networks for comparison can achieve a low loss on other problems. The other challenge is that we would like to consider the typical case where the first layer weights are not frozen. In the following, we will introduce the core concept of Gradient Features to address the first challenge, and stipulate proper geometric properties of Gradient Features for the second challenge.

## 3.2 Core Concepts in the Gradient Feature Learning Framework

Now, we will introduce the core concept in our framework, Gradient Features, and use it to build the family of networks to derive guarantees. As mentioned, we consider the setting where the first layer is not frozen. After the network learns good features, to ensure the updates in later gradient steps of the first layer are still benign for feature learning, we need some geometric conditions about the gradient features, which are measured by parameters in the definition of Gradient Features. The conditions are general enough, so that, as shown in Section 4, many prototypical problems satisfy them and the induced family of networks enjoys low loss, leading to useful guarantees. We begin by considering what features can be learned via gradients. Note that the gradient w.r.t. $\mathbf{w}_i$ is

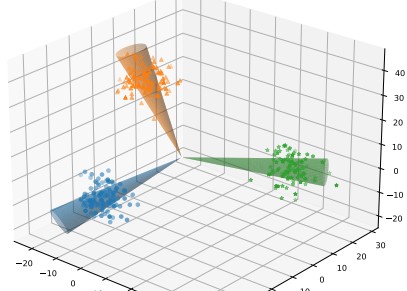

Gradient Feature being cones under Mixture of Gaussians data

$$\frac{\partial \mathcal{L}_{\mathcal{D}}(f)}{\partial \mathbf{w}_i} = \mathbf{a}_i \mathbb{E}_{(\mathbf{x},y)} \left[\ell'(yf(\mathbf{x}))y\left[\sigma'\left(\langle \mathbf{w}_i, \mathbf{x} \rangle - \mathbf{b}_i\right)\right] \mathbf{x}\right]$$

$$= \mathbf{a}_i \mathbb{E}_{(\mathbf{x},y)} \left[\ell'(yf(\mathbf{x}))y\mathbf{x}\mathbb{I}[\langle \mathbf{w}_i, \mathbf{x} \rangle > \mathbf{b}_i]\right].$$

Inspired by this, we define the following notion:

**Definition 3.5** (Simplified Gradient Vector). *For any $\mathbf{w} \in \mathbb{R}^d, b \in \mathbb{R}$, a Simplified Gradient Vector is*

$$G(\mathbf{w}, b) := \mathbb{E}_{(\mathbf{x},y)\sim\mathcal{D}}[y\mathbf{x}\mathbb{I}[\mathbf{w}^\top \mathbf{x} > b]]. \quad (4)$$

**Remark 3.6.** *Note that the definition of $G(\mathbf{w}, b)$ ignores the term $\ell'(yf(\mathbf{x}))$ in the gradient, where $f$ is the model function. In the early stage of training (or the first gradient step), $\ell'(\cdot)$ is approximately a constant, i.e., $\ell'(yf(\mathbf{x})) \approx \ell'(0)$ due to the symmetric initialization (see Equation (8)).*

Figure 1: An illustration of Gradient Feature, i.e., Definition 3.7 with random initialization (Gaussian), under Mixture of three Gaussian clusters in 3-dimension data space with blue/green/orange color. The Gradient Feature stays in three cones, where each center of the cone aligns with the corresponding Gaussian cluster center.

**Definition 3.7** (Gradient Feature). *For a unit vector $D \in \mathbb{R}^d$ with $\|D\|_2 = 1$, and a $\gamma \in (0, 1)$, a direction neighborhood (cone) $\mathcal{C}_{D,\gamma}$ is defined as:*

$$\mathcal{C}_{D,\gamma} := \{\mathbf{w} \mid |\langle \mathbf{w}, D \rangle|/\|\mathbf{w}\|_2 > (1 - \gamma)\}. \quad (5)$$

*Let $\mathbf{w} \in \mathbb{R}^d$, $b \in \mathbb{R}$ be random variables drawn from some distribution $\mathcal{W}, \mathcal{B}$. A Gradient Feature set with parameters $p, \gamma, B_G$ is defined as:*

$$S_{p,\gamma,B_G}(\mathcal{W}, \mathcal{B}) := \left\{ (D, s) \mid \Pr_{\mathbf{w}, b} \left[ G(\mathbf{w}, b) \in \mathcal{C}_{D,\gamma}, \|G(\mathbf{w}, b)\|_2 \geq B_G, s = b/|b| \right] \geq p \right\}. \quad (6)$$

**Remark 3.8.** *When clear from context, write it as $S_{p,\gamma,B_G}$. Gradient features (see Figure 1 for illustration) are simply normalized vectors $D$ that are given (approximately) by the simplified gradient vectors. (Similarly, the normalized scalar $s$ is given by the bias $b$.) To be a useful gradient feature, we require the direction to be "hit" by sufficiently large simplified gradient vectors with sufficient large probability, so as to be distinguished from noise and remain useful throughout the gradient steps. Later we will use the gradient features when $\mathcal{W}, \mathcal{B}$ are the initialization distributions.*

To make use of the gradient features, we consider the following family of networks using these features and with bounded norms, and will provide guarantees compared to the best in this family:

**Definition 3.9** (Gradient Feature Induced Networks). *The Gradient Feature Induced Networks are:*

$$\mathcal{F}_{d,m,B_F,S} := \left\{ f_{(\mathbf{a}, \mathbf{W}, \mathbf{b})} \in \mathcal{F}_{d,m} \mid \forall i \in [m], \ |\mathbf{a}_i| \leq B_{a1}, \|\mathbf{a}\|_2 \leq B_{a2}, (\mathbf{w}_i, \mathbf{b}_i/|\mathbf{b}_i|) \in S, \ |\mathbf{b}_i| \leq B_b \right\},$$

*where $S$ is some Gradient Feature set and $B_F := (B_{a1}, B_{a2}, B_b)$ are some parameters.*

**Remark 3.10.** *In above definition, the weight and bias of a neuron are simply the scalings of some item in the feature set $S$ (for simplicity the scaling of $\mathbf{w}_i$ is absorbed into the scaling of $\mathbf{a}_i$ and $\mathbf{b}_i$).*

**Definition 3.11** (Optimal Approximation via Gradient Features). *The optimal approximation network and loss using Gradient Feature Induced Networks $\mathcal{F}_{d,r,B_F,S}$ are defined as:*

$$f^* := \underset{f \in \mathcal{F}_{d,r,B_F,S}}{\arg\min} \mathcal{L}_{\mathcal{D}}(f), \qquad \mathrm{OPT}_{d,r,B_F,S} := \min_{f \in \mathcal{F}_{d,r,B_F,S}} \mathcal{L}_{\mathcal{D}}(f). \quad (7)$$

### 3.3 Provable Guarantee via Gradient Feature Learning

To obtain the guarantees, we first specify the symmetric initialization. It is convenient for the analysis and is typical in existing analysis (e.g., [7, 32, 33, 105]), though some other initialization can also work. Formally, we train a two-layer network with $4m$ neurons, $f_{(\mathbf{a}, \mathbf{W}, \mathbf{b})} \in \mathcal{F}_{d,4m}$. We initialize $\mathbf{a}_i^{(0)}, \mathbf{w}_i^{(0)}$ from Gaussians and $\mathbf{b}_i$ from a constant for $i \in \{1, \ldots, m\}$, and initialize the parameters for $i \in \{m+1, \ldots, 4m\}$ accordingly to get a zero output initial network. Specifically:

$$\text{for } i \in \{1, \ldots, m\}: \quad \mathbf{a}_i^{(0)} \sim \mathcal{N}(0, \sigma_a^2), \mathbf{w}_i^{(0)} \sim \mathcal{N}(0, \sigma_w^2 I), \mathbf{b}_i = \tilde{b},$$

$$\text{for } i \in \{m+1, \ldots, 2m\}: \quad \mathbf{a}_i^{(0)} = -\mathbf{a}_{i-m}^{(0)}, \mathbf{w}_i^{(0)} = -\mathbf{w}_{i-m}^{(0)}, \mathbf{b}_i = -\mathbf{b}_{i-m}, \quad (8)$$

$$\text{for } i \in \{2m+1, \ldots, 4m\}: \quad \mathbf{a}_i^{(0)} = -\mathbf{a}_{i-2m}^{(0)}, \mathbf{w}_i^{(0)} = \mathbf{w}_{i-2m}^{(0)}, \mathbf{b}_i = \mathbf{b}_{i-2m},$$

where $\sigma_a^2, \sigma_w^2, \tilde{b} > 0$ are hyper-parameters. After initialization, $\mathbf{a}, \mathbf{W}$ are updated as in Algorithm 1. We are now ready to present our main result in the framework.

---

**Theorem 3.12** (Main Result). *Assume Assumption 3.1. For any $\epsilon, \delta \in (0, 1)$, if $m \leq e^d$ and*

$$m = \Omega\left( \frac{1}{p\epsilon^4} \left( r B_{a1} B_{x1} \sqrt{\frac{B_b}{B_G}} \right)^4 + \frac{1}{\sqrt{\delta}} + \frac{1}{p} \left( \log\left( \frac{r}{\delta} \right) \right)^2 \right),$$

$$T = \Omega\left( \frac{1}{\epsilon} \left( \frac{\sqrt{r} B_{a2} B_b B_{x1}}{(mp)^{\frac{1}{4}}} + m\tilde{b} \right) \left( \frac{\sqrt{\log m}}{\sqrt{B_b B_G}} + \frac{1}{B_{x1}(mp)^{\frac{1}{4}}} \right) \right),$$

$$\frac{n}{\log n} = \tilde{\Omega}\left( \frac{m^3 p B_x^2 B_{a2}^4 B_b}{\epsilon^2 r^2 B_{a1}^2 B_G} + \frac{(mp)^{\frac{1}{2}} B_{x2}}{B_b B_G} + \frac{B_x^2}{B_{x2}} + \frac{1}{p} + \left( \frac{1}{B_G^2} + \frac{1}{B_{x1}^2} \right) \frac{B_{x2}}{|\ell'(0)|^2} + \frac{Tm}{\delta} \right),$$

*then with initialization (8) and proper hyper-parameter values, we have with probability $\geq 1 - \delta$ over the initialization and training samples, there exists $t \in [T]$ in Algorithm 1 with:*

$$\Pr[\mathrm{sign}(f_{\Xi^{(t)}}(\mathbf{x})) \neq y] \leq \mathcal{L}_{\mathcal{D}}\left( f_{\Xi^{(t)}} \right)$$

$$\leq \mathrm{OPT}_{d,r,B_F,S_{p,\gamma,B_G}} + r B_{a1} B_{x1} \sqrt{2\gamma + O\left( \frac{\sqrt{B_{x2} \log n}}{B_G |\ell'(0)| n^{\frac{1}{2}}} \right)} + \epsilon.$$

---

Intuitively, the theorem shows when a data distribution admits a small approximation error by some "ground-truth" network with $r$ neurons using gradient features from $S_{p,\gamma,B_G}$ (i.e., a small optimal approximate loss $\mathrm{OPT}_{d,r,B_F,S_{p,\gamma,B_G}}$), the gradient descent training can successfully learn good neural networks with sufficiently many $m$ neurons.

Now we discuss the requirements and the error guarantee. Viewing boundedness parameters $B_{a1}, B_{x1}$ etc. as constants, then the number $m$ of neurons learned is roughly $\tilde{\Theta}\left(\frac{r^4}{p\epsilon^4}\right)$, a polynomial overparameterization compared to the "ground-truth" network. The proof shows that such an overparameterization is needed such that some neurons can capture the gradient features given by gradient descent. This is consistent with existing analysis about overparameterization network learning, and also consistent with existing empirical observations.

The error bound consists of three terms. The last term $\epsilon$ can be made arbitrarily small, while the other two depend on the concrete data distribution. Specifically, with larger $r$ and $\gamma$, the second term increases. While the first term (the optimal approximation loss) decreases, since a larger $r$ means a larger "ground-truth" network family, and a larger $\gamma$ means a larger Gradient Feature set $S_{p,\gamma,B_G}$. So, there is a trade-off between these two terms. When we later apply the framework to concrete problems (e.g., mixtures of Gaussians, parity functions), we will show that depending on the specific data distribution, we can choose the proper values for $r, \gamma$ to make the error small. This then leads to error guarantees for the concrete problems and demonstrates the unifying power of the framework. Please refer to Appendix D.3 for more discussion about our problem setup and our core concept, e.g., parameter choice, early stopping, the role of $s$, activation functions, and so on.

**Proof Sketch.** The intuition in the proof of Theorem 3.12 is closely related to the notion of Gradient Features. First, the gradient descent will produce gradients that approximate the features in $S_{p,\gamma,B_G}$. Then, the gradient descent update gives a good set of neurons, such that there exists an accurate classifier using these neurons with loss comparable to the optimal approximation loss. Finally, the training will learn to approximate the accurate classifier, resulting in the desired error guarantee. The complete proof is in Appendix D (the population version in Appendix D.2 and the empirical version in Appendix D.4), including the proper values for hyper-parameters such as $\eta^{(t)}$ in Theorem D.17. Below, we briefly sketch the key ideas and omit the technical details.

We first show that a large subset of neurons has gradients at the first step as good features. (The claim can be extended to multiple steps; for simplicity, we follow existing work (e.g., [33, 105]) and present only the first step.) Let $\nabla_i$ denote the gradient of the $i$-th neuron $\nabla_{\mathbf{w}_i}\mathcal{L}_{\mathcal{D}}(f_{\Xi^{(0)}})$. Denote the subset of neurons with nice gradients approximating feature $(D, s)$ as:

$$G_{(D,s),Nice} := \left\{ i \in [2m] : s = \mathbf{b}_i/|\mathbf{b}_i|, \langle \nabla_i, D \rangle > (1-\gamma)\|\nabla_i\|_2, \|\nabla_i\|_2 \geq \left|\mathbf{a}_i^{(0)}\right| B_G \right\}. \quad (9)$$

**Lemma 3.13** (Feature Emergence). *For any $r$ size subset $\{(D_1, s_1), \ldots, (D_r, s_r)\} \subseteq S_{p,\gamma,B_G}$, with probability at least $1 - re^{-\Theta(mp)}$, for all $j \in [r]$, we have $|G_{(D_j,s_j),Nice}| \geq \frac{mp}{4}$.*

This is because $\nabla_i = \ell'(0)\mathbf{a}_i^{(0)}\mathbb{E}_{(\mathbf{x},y)}\left[y\sigma'\left[\left\langle \mathbf{w}_i^{(0)}, \mathbf{x}\right\rangle - \mathbf{b}_i\right]\mathbf{x}\right] = \ell'(0)\mathbf{a}_i^{(0)}G(\mathbf{w}_i^{(0)}, \mathbf{b}_i)$. Now consider $s_j = +1$ (the case $-1$ is similar). Since $\mathbf{w}_i$ is initialized by Gaussians, by $\nabla_i$'s connection to Gradient Features, we can see that for all $i \in [m]$, $\Pr\left[i \in G_{(D_j,+1),Nice}\right] \geq \frac{p}{2}$. The lemma follows from concentration via a large enough $m$, i.e., sufficient overparameterization. The gradients allow obtaining a set of neurons approximating the "ground-truth" network with comparable loss:

**Lemma 3.14** (Existence of Good Networks). *For any $\delta \in (0,1)$, with proper hyper-parameter values, with probability at least $1 - \delta$, there is $\tilde{\mathbf{a}}$ such that $\|\tilde{\mathbf{a}}\|_0 = O\left(r\sqrt{mp}\right)$ and $f_{(\tilde{\mathbf{a}},\mathbf{W}^{(1)},\mathbf{b})}(\mathbf{x}) = \sum_{i=1}^{4m}\tilde{\mathbf{a}}_i\sigma\left(\left\langle \mathbf{w}_i^{(1)}, \mathbf{x}\right\rangle - \mathbf{b}_i\right)$ satisfies*

$$\mathcal{L}_{\mathcal{D}}(f_{(\tilde{\mathbf{a}},\mathbf{W}^{(1)},\mathbf{b})}) \leq \mathrm{OPT}_{d,r,B_F,S_{p,\gamma,B_G}} + \sqrt{2}rB_{a1}B_{x1}\left(\sqrt{\gamma} + \sqrt{\frac{2B_b}{\sqrt{mp}B_G}}\right).$$

Given the good set of neurons, we finally show that the remaining gradient steps can learn an accurate classifier. Intuitively, with small step sizes $\eta^{(t)}$, the weights of the first layer $\mathbf{w}_i$ do not change too much (stay in a neighborhood) while the second layer weights grow, and thus the learning is similar to convex learning using the good set of neurons. Technically, we adopt the online convex optimization analysis (Theorem D.5) in [33] to get the final loss guarantee in Theorem 3.12.

# 4 Applications in Special Cases

In this section we will apply the gradient feature learning framework to some specific problems, corresponding to concrete data distributions $\mathcal{D}$. We primarily focus on prototypical problems for analyzing feature learning in networks. We will present here the results for mixtures of Gaussians and parity functions, and include the complete proofs and some other results in Appendix E.

## 4.1 Mixtures of Gaussians

Mixtures of Gaussians are among the most fundamental and widely used statistical models. Recently, it has been used to study neural network learning, in particular, the effect of gradient descent for feature learning of two-layer neural networks and the advantage over fixed feature methods [46, 99].

**Data Distributions.** We follow notations from [99]. The data are from a mixture of $r$ high-dimensional Gaussians, and each Gaussian is assigned to one of two possible labels in $\mathcal{Y} = \{\pm 1\}$. Let $\mathcal{S}(y) \subseteq [r]$ denote the set of indices of Gaussians associated with the label $y$. The data distribution is then: $q(\mathbf{x}, y) = q(y)q(\mathbf{x}|y), q(\mathbf{x}|y) = \sum_{j \in \mathcal{S}(y)} p_j \mathcal{N}_j(\mathbf{x})$, where $\mathcal{N}_j(\mathbf{x})$ is a multivariate normal distribution with mean $\mu_j$, covariance $\Sigma_j$, and $p_j$ are chosen such that $q(\mathbf{x}, y)$ is correctly normalized. We will make some assumptions about the Gaussians, for which we first introduce some notations.

$$D_j := \frac{\mu_j}{\|\mu_j\|_2}, \quad \tilde{\mu}_j := \mu_j/\sqrt{d}, \quad B_{\mu 1} := \min_{j \in [r]} \|\tilde{\mu}_j\|_2, \quad B_{\mu 2} := \max_{j \in [r]} \|\tilde{\mu}_j\|_2, \quad p_B := \min_{j \in [r]} p_j.$$

**Assumption 4.1.** *Let $8 \leq \tau \leq d$ be a parameter that will control our final error guarantee. Assume*

- *Equiprobable labels: $q(-1) = q(+1) = 1/2$.*
- *For all $j \in [r]$, $\Sigma_j = \sigma_j I_{d \times d}$. Let $\sigma_B := \max_{j \in [r]} \sigma_j$ and $\sigma_{B+} := \max\{\sigma_B, B_{\mu 2}\}$.*
- *$r \leq 2d$, $p_B \geq \frac{1}{2d}$, $\Omega\left(1/d + \sqrt{\tau \sigma_{B+}^2 \log d/d}\right) \leq B_{\mu 1} \leq B_{\mu 2} \leq d$.*
- *The Gaussians are well-separated: for all $i \neq j \in [r]$, we have $-1 \leq \langle D_i, D_j \rangle \leq \theta$, where*

$$0 \leq \theta \leq \min\left\{\frac{1}{2r}, \frac{\sigma_{B+}}{B_{\mu 2}}\sqrt{\frac{\tau \log d}{d}}\right\}.$$

**Remark 4.2.** *The first two assumptions are for simplicity; they can be relaxed. We can generalize our analysis to the mixture of Gaussians with unbalanced label probabilities and general covariances. The third assumption is to make sure that each Gaussian has a good amount of probability mass to be learned. The remaining assumptions are to make sure that the Gaussians are well-separated and can be distinguished by the learning algorithm.*

We are now ready to apply the framework to these data distributions, for which we only need to compute the Gradient Feature set and the corresponding optimal approximation loss.

**Lemma 4.3** (Mixtures of Gaussians: Gradient Features). *$(D_j, +1) \in S_{p, \gamma, B_G}$ for all $j \in [r]$, where*

$$p = \frac{B_{\mu 1}}{\sqrt{\tau \log d}\sigma_{B+} \cdot d^{\Theta\left(\tau \sigma_{B+}^2 / B_{\mu 1}^2\right)}}, \quad \gamma = \frac{1}{d^{0.9\tau - 1.5}}, \quad B_G = p_B B_{\mu 1}\sqrt{d} - O\left(\frac{\sigma_{B+}}{d^{0.9\tau}}\right).$$

*Let $f^*(\mathbf{x}) = \sum_{j=1}^{r} \frac{y_{(j)}}{\sqrt{\tau \log d}\sigma_{B+}} \left[\sigma\left(\langle D_j, \mathbf{x}\rangle - 2\sqrt{\tau \log d}\sigma_{B+}\right)\right]$ whose hinge loss is at most $\frac{3}{d^\tau} + \frac{4}{d^{0.9\tau - 1}\sqrt{\tau \log d}}$.*

Given the values on gradient feature parameters $p, \gamma, B_G$ and the optimal approximation loss $\mathrm{OPT}_{d, r, B_F, S_{p, \gamma, B_G}}$, the framework immediately leads to the following guarantee:

---

**Theorem 4.4** (Mixtures of Gaussians: Main Result). *Assume Assumption 4.1. For any $\epsilon, \delta \in (0, 1)$, when Algorithm 1 uses hinge loss with*

$$m = \mathrm{poly}\left(\frac{1}{\delta}, \frac{1}{\epsilon}, d^{\Theta\left(\tau \sigma_{B+}^2 / B_{\mu 1}^2\right)}, r, \frac{1}{p_B}\right) \leq e^d, \quad T = \mathrm{poly}(m), \quad n = \mathrm{poly}(m)$$

*and proper hyper-parameters, then with probability at least $1 - \delta$, there exists $t \in [T]$ such that*

$$\Pr[\mathrm{sign}(f_{\Xi^{(t)}}(\mathbf{x})) \neq y] \leq \frac{\sqrt{2}r}{d^{0.4\tau - 0.8}} + \epsilon.$$

---

The theorem shows that gradient descent can learn to a small error via learning the gradient features, given proper hyper-parameters. In particular, we need sufficient overparameterization (a sufficiently large number $m$ of neurons). When $\sigma_{B+}{}^2/B_{\mu 1}^2$ is a constant which is the prototypical interesting case, and we choose a constant $\tau$, then $m$ is polynomial in the key parameters $\frac{1}{\delta}, \frac{1}{\epsilon}, d, r, \frac{1}{p_B}$, and the error bound is inverse polynomial in $d$. The complete proof is given in Appendix E.2.

[46] studies (almost) linear separable cases while our setting includes non-linear separable cases, e.g., XOR. [99] mainly studies neural network classification on 4 Gaussian clusters with XOR structured labels, while our setting is much more general, e.g., our cluster number can extend up to $2d$.

### 4.1.1 Mixtures of Gaussians: Beyond the Kernel Regime

As discussed in the introduction, it is important for the analysis to go beyond fixed feature methods such as NTK (i.e., the kernel regime), so as to capture the feature learning ability which is believed to be the key factor for the empirical success. We first review the fixed feature methods. Following [33], suppose $\Psi$ is a data-independent feature mapping of dimension $N$ with bounded features, i.e., $\Psi : \mathcal{X} \to [-1, 1]^N$. For $B > 0$, the family of linear models on $\Psi$ with bounded norm $B$ is $\mathcal{H}_B = \{h(\tilde{\mathbf{x}}) : h(\tilde{\mathbf{x}}) = \langle \Psi(\tilde{\mathbf{x}}), w \rangle, \|w\|_2 \leq B\}$. This can capture linear models on fixed finite-dimensional feature maps, e.g., NTK, and also infinite dimensional feature maps, e.g., kernels like RBF, that can be approximated by feature maps of polynomial dimensions [64, 98, 105].

Our framework indeed goes beyond fixed features and shows features from gradients are more powerful than features from random initialization, e.g., NTK. Our framework can show the advantage of network learning over kernel methods under the setting of [99] (4 Gaussian clusters with XOR structured labels). For large enough $d$, our framework only needs roughly $\Omega(\log d)$ neurons and $\Omega((\log d)^2)$ samples to achieve arbitrary small constant error (see Theorem E.18 when $\sigma_B = 1$), while fixed feature methods need $\Omega(d^2)$ features and $\Omega(d^2)$ samples to achieve nontrivial errors (as proved in [99]). Moreover, [99] uses ODE to simulate the optimization process for the 2-layer networks learning XOR-shaped Gaussian mixture with $\Omega(1)$ neurons and gives convincing evidence that $\Omega(d)$ samples is enough to learn it, yet they do not give a rigorous convergence guarantee for this problem. We successfully derive a convergence guarantee and we require a much smaller sample size $\Omega((\log d)^2)$. For the proof (detailed in Appendix E.3), we only need to calculate the $p, \gamma, B_G$ of the data distribution carefully and then inject these numbers into Theorem 3.12.

## 4.2 Parity Functions

Parity functions are a canonical family of learning problems in computational learning theory, usually for showing theoretical computational barriers [103]. The typical sparse parties over $d$-dim binary inputs $\phi \in \{\pm 1\}^d$ are $\prod_{i \in A} \phi_i$ where $A \subseteq [d]$ is a subset of dimensions. Recent studies have shown that when the distribution of inputs $\phi$ has structures rather than uniform, neural networks can perform feature learning and finally learn parity functions with a small error, while methods without feature learning, e.g. NTK, cannot achieve as good results [33, 76, 105]. Thus, this has been a prototypical setting for studying feature learning phenomena in networks. Here we consider a generalization of this problem and show that our framework can show successful learning via gradient descent.

**Data Distributions.** Suppose $\mathbf{M} \in \mathbb{R}^{d \times D}$ is an unknown dictionary with $D$ columns that can be regarded as patterns. For simplicity, assume $d = D$ and $\mathbf{M}$ is orthonormal. Let $\phi \in \mathbb{R}^d$ be a hidden representation vector. Let $A \subseteq [D]$ be a subset of size $rk$ corresponding to the class relevant patterns and $r$ is an odd number. Then the input is generated by $\mathbf{M}\phi$, and some function on $\phi_A$ generates the label. WLOG, let $A = \{1, \ldots, rk\}$, $A^\perp = \{rk + 1, \ldots, d\}$. Also, we split $A$ such that for all $j \in [r]$, $A_j = \{(j-1)k + 1, \ldots, jk\}$. Then the input $\mathbf{x}$ and the class label $y$ are given by:

$$\mathbf{x} = \mathbf{M}\phi, y = g^*(\phi_A) = \text{sign}\Big( \sum_{j \in [r]} \text{XOR}(\phi_{A_j}) \Big), \tag{10}$$

where $g^*$ is the ground-truth labeling function mapping from $\mathbb{R}^{rk}$ to $\mathcal{Y} = \{\pm 1\}$, $\phi_A$ is the sub-vector of $\phi$ with indices in $A$, and $\text{XOR}(\phi_{A_j}) = \prod_{l \in A_j} \phi_l$ is the parity function. We still need to specify the distribution $\mathcal{X}$ of $\phi$, which determines the structure of the input distribution:

$$\mathcal{X} := (1 - 2rp_A)\mathcal{X}_U + \sum_{j \in [r]} p_A(\mathcal{X}_{j,+} + \mathcal{X}_{j,-}). \tag{11}$$

For all corresponding $\phi_{A^\perp}$ in $\mathcal{X}$, we have $\forall l \in A^\perp$, independently: $\phi_l = \begin{cases} +1, & \text{w.p. } p_o \\ -1, & \text{w.p. } p_o \\ 0, & \text{w.p. } 1 - 2p_o \end{cases}$,

where $p_o$ controls the signal noise ratio: if $p_o$ is large, then there are many nonzero entries in $A^\perp$ which are noise interfering with the learning of the ground-truth labeling function on $A$. For corresponding $\phi_A$, any $j \in [r]$, we have

- In $\mathcal{X}_{j,+}$, $\phi_{A_j} = [+1, +1, \ldots, +1]^\top$ and $\phi_{A \setminus A_j}$ only have zero elements.
- In $\mathcal{X}_{j,-}$, $\phi_{A_j} = [-1, -1, \ldots, -1]^\top$ and $\phi_{A \setminus A_j}$ only have zero elements.
- In $\mathcal{X}_U$, we have $\phi_A$ draw from $\{+1, -1\}^{rk}$ uniformly.

In short, we have $r$ parity functions each corresponding to a block of $k$ dimensions; $\mathcal{X}_{j,+}$ and $\mathcal{X}_{j,-}$ stands for the component providing a strong signal for the $j$-th parity; $\mathcal{X}_U$ corresponds to uniform distribution unrelated to any parity and providing weak learning signal; $A^\perp$ is the noise part. The label depends on the sum of the $r$ parity functions.

**Assumption 4.5.** *Let $8 \leq \tau \leq d$ be a parameter that will control our final error guarantee. Assume $k$ is an odd number and: $k \geq \Omega(\tau \log d)$, $\quad d \geq rk + \Omega(\tau r \log d)$, $\quad p_o = O\left(\frac{rk}{d-rk}\right)$, $\quad p_A \geq \frac{1}{d}$.*

**Remark 4.6.** *We set up the problem to be more general than the parity function learning in existing work. If $r = 1$, the labeling function reduces to the traditional $k$-sparse parties of $d$ bits. The assumptions require $k, d$, and $p_A$ to be sufficiently large so as to provide enough large signals for learning. Note that when $k = \frac{d}{16}, r = 1, p_o = \frac{1}{2}$, our analysis also holds, which shows our framework is beyond the kernel regime (discuss in detail in Section 4.2.1).*

To apply our framework, again we only need to compute the Gradient Feature set and the corresponding optimal loss. We first define the Gradient Features: For all $j \in [r]$, let $D_j = \frac{\sum_{l \in A_j} \mathbf{M}_l}{\|\sum_{l \in A_j} \mathbf{M}_l\|_2}$.

**Lemma 4.7** (Parity Functions: Gradient Features). *We have $(D_j, +1), (D_j, -1) \in S_{p,\gamma,B_G}$ for all $j \in [r]$, where*

$$p = \Theta\left(\frac{1}{\sqrt{\tau r \log d} \cdot d^{\Theta(\tau r)}}\right), \qquad \gamma = \frac{1}{d^{\tau-2}}, \qquad B_G = \sqrt{k} p_A - O\left(\frac{\sqrt{k}}{d^\tau}\right). \qquad (12)$$

*With gradient features from $S_{p,\gamma,B_G}$, let $f^*(\mathbf{x}) = \sum_{j=1}^{r} \sum_{i=0}^{k} (-1)^{i+1} \sqrt{k} \left[\sigma\left(\langle D_j, \mathbf{x}\rangle - \frac{2i-k-1}{\sqrt{k}}\right) - 2\sigma\left(\langle D_j, \mathbf{x}\rangle - \frac{2i-k}{\sqrt{k}}\right) + \sigma\left(\langle D_j, \mathbf{x}\rangle - \frac{2i-k+1}{\sqrt{k}}\right)\right]$ whose hinge loss is 0.*

Above, we show that $D_j$ is the "indicator function" for the subset $A_j$ so that we can build the optimal neural network based on such directions. Given the values on gradient feature parameters and the optimal approximation loss, the framework immediately leads to the following guarantee:

---

**Theorem 4.8** (Parity Functions: Main Result). *Assume Assumption 4.5. For any $\epsilon, \delta \in (0, 1)$, when Algorithm 1 uses hinge loss with*

$$m = \text{poly}\left(\frac{1}{\delta}, \frac{1}{\epsilon}, d^{\Theta(\tau r)}, k, \frac{1}{p_A}\right) \leq e^d, \quad T = \text{poly}(m), \quad n = \text{poly}(m)$$

*and proper hyper-parameters, then with probability at least $1 - \delta$, there exists $t \in [T]$ such that*

$$\Pr[\text{sign}(f_{\Xi^{(t)}}(\mathbf{x})) \neq y] \leq \frac{3r\sqrt{k}}{d^{(\tau-3)/2}} + \epsilon.$$

---

The theorem shows that gradient descent can learn to a small error in this problem. We also need sufficient overparameterization: When $r$ is a constant (e.g., $r = 1$ in existing work), and we choose a constant $\tau$, $m$ is polynomial in $\frac{1}{\delta}, \frac{1}{\epsilon}, d, k, \frac{1}{p_A}$, and the error bound is inverse polynomial in $d$. The proof is in Appendix E.4. Our setting is more general than that in [33, 76] which corresponds to $\mathbf{M} = I, r = 1, p_A = \frac{1}{4}, p_o = \frac{1}{2}$. [105] study single index learning, where one feature direction is enough for a two-layer network to recover the label, while our setting considers $r$ directions $D_1, \ldots, D_r$, so the network needs to learn multiple directions to get a small error.

#### 4.2.1 Parity Functions: Beyond the Kernel Regime

Again, we show that our framework indeed goes beyond fixed features under parity functions. Our problem setting in Section 4.2 is general enough to include the problem setting in [33]. Their lower bound for fixed feature methods directly applies to our case and leads to the following:

**Proposition 4.9.** *There exists a data distribution in the parity learning setting in Section 4.2 with* $\mathbf{M} = I, r = 1, p_A = \frac{1}{4}, k = \frac{d}{16}, p_o = \frac{1}{2}$, *such that all* $h \in \mathcal{H}_B$ *have hinge-loss at least* $\frac{1}{2} - \frac{\sqrt{N}B}{2^k \sqrt{2}}$.

This means to get an inverse-polynomially small loss, fixed feature models need to have an exponentially large size, i.e., either the number of features $N$ or the norm $B$ needs to be exponential in $k$. In contrast, Theorem 4.8 shows our framework guarantees a small loss with a polynomially large model, runtime, and sample complexity. Clearly, our framework is beyond the fixed feature methods.

**Parities on Uniform Inputs.** When $r = 1, p_A = 0$, our problem setting will degenerate to the classic sparse parity function on a uniform input distribution. This has also been used for analyzing network learning [16]. For this case, our framework can get a $k2^{O(k)} \log(k)$ network width bound and a $O(d^k)$ sample complexity bound, matching those in [16]. This then again confirms the advantage of network learning over kernel methods that requires $d^{\Omega(k)}$ dimensions as shown in [16]. See the full statement in Theorem E.31, details in Appendix E.5, and alternative analysis in Appendix E.6.

## 5 Further Implications and Conclusion

Our general framework sheds light on several interesting phenomena in NN learning observed in practice. Feature learning beyond the kernel regime has been discussed in Section 4.1.1 and Section 4.2.1. Here we discuss the LTH and defer more implications such as simplicity bias, learning over different data distributions, and new perspectives about roadmaps forward in Appendix C.

**Lottery Ticket Hypothesis (LTH).** Another interesting phenomenon is the LTH [41]: randomly-initialized networks contain subnetworks that when trained in isolation reach test accuracy comparable to the original network in a similar number of iterations. Later studies (e.g., [42]) show that LTH is more stable when subnetworks are found in the network after a few gradient steps.

Our framework provides an explanation for two-layer networks: the lottery ticket subnetwork contains exactly those neurons whose gradient feature approximates the weights of the "ground-truth" network $f^*$; they may not exist at initialization but can be found after the first gradient step. More precisely, Lemma 3.14 shows that after the first gradient step, there is a *sparse* second-layer weight $\tilde{\mathbf{a}}$ with $\|\tilde{\mathbf{a}}\|_0 = O\left(r\sqrt{mp}\right)$, such that using this weight on the hidden neurons gives a network with a small loss. Let $U$ be the support of $\tilde{\mathbf{a}}$. Equivalently, there is a small-loss subnetwork $f_{\Xi}^U$ with only neurons in $U$ and with second-layer weight $\tilde{\mathbf{a}}_U$ on these neurons. Following the same proof of Theorem 3.12:

**Proposition 5.1.** *In the same setting of Theorem 3.12 but only considering the subnetwork supported on $U$ after the first gradient step, with the same requirements on $m$ and $T$, with proper hyper-parameter values, we have the same guarantee: with probability $\geq 1 - \delta$, there is $t \in [T]$ with*

$$\Pr[\text{sign}(f_{\Xi^{(t)}}^U)(\mathbf{x}) \neq y] \leq \text{OPT}_{d,r,B_F,S_{p,\gamma},B_G} + rB_{a1}B_{x1}\sqrt{2\gamma + O\left(\frac{\sqrt{B_{x2}\log n}}{B_G\sqrt{n}}\right)} + \epsilon.$$

This essentially formally proves LTH for two-layer networks, showing (a) the existence of the winning lottery subnetwork and (b) that gradient descent on the subnetwork can learn to similar loss in similar runtime as on the whole network. In particular, (b) is novel and not analyzed in existing work.

We provide our work's broader impacts and limitations (e.g., statement of recovering existing results and some failure cases beyond our framework) in Appendix A and Appendix B respectively.

**Conclusion.** We propose a general framework for analyzing two-layer neural network learning by gradient descent and show that it can lead to provable guarantees for several prototypical problem settings for analyzing network learning. In particular, our framework goes beyond fixed feature methods, e.g., NTK. It sheds light on several interesting phenomena in NN learning, e.g., the lottery ticket hypothesis and simplicity bias. Future directions include: (1) How to extend the framework to deeper networks? (2) While the current framework focuses on the gradient features in the early gradient steps, whether feature learning also happens in later steps and if so how to formalize that?

## Acknowledgements

The work is partially supported by Air Force Grant FA9550-18-1-0166, the National Science Foundation (NSF) Grants 2008559-IIS, 2023239-DMS, and CCF-2046710.

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

# Appendix

**Contents**

Appendix A discusses the potential societal impact of our work. Appendix B describes the limitations of our work. In Appendix C, we present our framework implications about simplicity bias. The complete proof of our main results is given in Appendix D. We present the case study of linear data in Appendix E.1, mixtures of Gaussians in Appendix E.2 and Appendix E.3, parity functions in Appendix E.4, Appendix E.5 and Appendix E.6, and multiple-index models in Appendix E.7. We put the auxiliary lemmas in Appendix F.

## A   Broader Impacts

Our paper is purely theoretical in nature, and thus we do not anticipate an immediate negative ethical impact. We provide a unified theoretical framework that can be applied to different theoretical problems. We propose the two key ideas of gradient feature and gradient feature-induced neural networks not only to show their ability to unify several current works but also to open a new direction of thinking with respect to the learning process. These notations have the potential to be extended to multi-layer gradient features and multi-step learning, and this work is only our first step.

On the other hand, this work may lead to a better understanding and inspire the development of improved network learning methods, which may have a positive impact on the theoretical machine-learning community. It may also be beneficial to engineering-inclined machine-learning researchers.

## B   Limitations

**Recover Existing Results.**    The framework may or may not recover the width or sample complexity bounds in existing work.

1. The framework can give matching bounds as the existing work in some cases, like parities over uniform inputs (Appendix E.5).
2. In some other cases, it gives polynomial error bounds not the same as those in the existing work (e.g., for parities over structured inputs). This is because our work is analyzing general cases, and thus may not give better than or the same bounds as those in special cases, since special cases have more properties that can be exploited to get potentially better bounds. On the other hand, our bounds can already show the advantage over kernel methods (e.g., Proposition 4.9).

We would like to emphasize that our contribution is providing an analysis framework that can (1) formalize the unifying principles of learning features from gradients in network training, and (2) give

polynomial error bounds for prototypical problems. Our focus is not to recover the guarantees in existing work.

**Failure Cases.** There are some failure cases that gradient feature learning framework cannot cover:

1. In [101], they constructed a function that is easy to approximate using a 3-layer network but not approximable by any 2-layer network. Since the function is not approximable by any 2-layer network, it cannot be approximated by the gradient-induced networks as well, so OPT will be large. As a result, the final error will be large.
2. In uniform parity data distribution, considering an odd number of features rather than even, i.e., $k$ is an odd number in Assumption E.28, we can show that our gradient feature set is empty even when $p$ in Equation (6) is exponentially small, thus the OPT is a positive constant since the gradient induced network can only be constants. Meanwhile, the neural network won't be able to learn this data distribution because its gradient is always 0 through the training, and the final error equals OPT.

The first case corresponds to the approximation hardness of 2-layer networks, while the second case gives a learning hardness example. The above two cases show that if there is an approximation or learning hardness, our gradient feature learning framework may be vacuous because the optimal model in the gradient feature class has a large risk, then the ground-truth mapping from inputs to labels is not learnable by gradient descent. These analyses are consistent with previous works [15, 101].

## C   More Further Implications

Our general framework also sheds some light on several interesting phenomena in neural network (NN) learning observed in practice. Feature learning beyond the kernel regime has been discussed in Section 4.1.1 and Section 4.2.1. The lottery ticket hypothesis (LTH) has been discussed in Section 5. Below we discuss other implications.

**Implicit Regularization/Simplicity Bias.** It is now well known that practical NN are overparameterized and traditional uniform convergence bounds cannot adequately explain their generalization performance [59, 88, 118]. It is generally believed that the optimization has some *implicit regularization* effect that restricts learning dynamics to a subset of the whole hypothesis class, which is not of high capacity so can lead to good generalization [53, 90]. Furthermore, learning dynamics tend to first learn simple functions and then learn more and more sophisticated ones (referred to as simplicity bias) [89, 102]. However, it remains elusive to formalize such simplicity bias.

Our framework provides a candidate explanation: the learning dynamics first learn to approximate the best network in a smaller family of gradient feature induced networks $\mathcal{F}_{d,r,B_F,S}$ and then learn to approximate the best in a larger family. Consider the number of neurons $r$ for illustration. Let $r_1 \ll r_2$, and let $T_1$ and $T_2$ be their corresponding runtime bounds for $T$ in the main Theorem 3.12. Clearly, $T_1 \ll T_2$. Then, at time $T_1$, the theorem guarantees the learning dynamics learn to approximate the best in the family $\mathcal{F}_{d,r_1,B_F,S}$ with $r_1$ neurons, but not for the larger family $\mathcal{F}_{d,r_2,B_F,S}$. Later, at time $T_2$, the learning dynamics learn to approximate the best in the larger family $\mathcal{F}_{d,r_2,B_F,S}$. That is, the learning first learns simpler functions and then more sophisticated ones where the simplicity bias is measured by the size of the family of gradient feature-induced networks. The implicit regularization is then restricting to networks approximating smaller families of gradient feature-induced networks. Furthermore, we can also conclude that for an SGD-optimized NN, its actual representation power is from the subset of NN based on gradient features, instead of the whole set of NN. This view helps explain the simplicity bias/implicit regularization phenomenon of NN learning in practice.

**Learning over Different Data Distributions.** Our framework articulates the following key principles (pointed out for specific problems in existing work but not articulated more generally):

- Role of gradient: the gradient leads to the emergence of good features, which is useful for the learning of upper layers in later stages.
- From features to solutions: learned features in early steps will not be distorted, if not improved, in later stages. The training dynamic for upper layers will eventually learn a good combination of hidden neurons based on gradient features, giving a good solution.

Then, more interesting insights are obtained from the generality of the framework. To build a general framework, the meaningful error guarantees should be data-dependent, since NN learning on general data distributions is hard and data-independent guarantees will be vacuous [34, 35]. Comparing the optimal in a family of "ground-truth" functions (inspired by agnostic learning in learning theory) is a useful method to obtain the data-dependent bound. We further construct the "ground-truth" functions using properties of the training dynamics, i.e., gradient features. This greatly facilitates the analysis of the training dynamics and is the key to obtaining the final guarantees. On the other hand, the framework can also be viewed as using the optimal by gradient-induced NN to measure or quantify the "complexity" of the problem. For easier problems, this quantity is smaller, and our framework can give a better error bound. So this provides a united way to derive guarantees for specific problems.

**New Perspectives about Roadmaps Forward.**    We argue a new perspective about the connection between the strong representation power and the successful learning of NN. Traditionally, the strong representation power of NN is the key reason for hardness results of NN learning: NN has strong representation power and can encode hard learning questions, so they are hard to learn. See the proof in SQ bound from [33] or NP-hardness from [19]. The strong representation power also causes trouble for the statistical aspect: it leads to vacuous generalization bounds when traditional uniform convergence tools are used.

Our framework suggests a perspective in sharp contrast: the strong representation power of NN with gradient features is actually the key to successful learning. More concretely, the optimal error of the gradient feature-induced NN being small (i.e., strong representation power for a given data distribution) can lead to a small guarantee, which is the key to successful learning. The above new perspective suggests a different analysis road than traditional ones. Traditional analysis typically first reasons about the optimal based on the whole function class, i.e. the ground truth, then analyze how NN learns proper features and reaches the optimal. In contrast, our framework defines feature family first, and then reasons about the optimal based on it.

Our framework provides the foundation for future work on analyzing gradient-based NN learning, which may inspire future directions including but not limited to (1) defining a new feature family for 2-layer NN rather than gradient feature, (2) considering deep NN and introducing new gradient features (e.g., gradient feature notion for upper layers), (3) defining different gradient feature family at different training stages (e.g., gradient feature notion for later stages). In particular, the challenges in the later-stage analysis are: (a) the weights in the later stage will not be as normal as the initialization, and we need new tools to analyze their properties; (b) to show that the later-stage features eventually lead to a good solution, we may need new analysis tools for the non-convex optimization due to the changes in the first layer weights.

## D   Gradient Feature Learning Framework

We first prove a Simplified Gradient Feature Learning Framework in Appendix D.1, which only considers one-step gradient feature learning. Then, we prove our Gradient Feature Learning Framework, e.g., no freezing of the first layer. In Appendix D.2, we consider population loss to simplify the proof. Then, we provide more discussion about our problem setup and our core concept in Appendix D.3. Finally, we prove our Gradient Feature Learning Framework under empirical loss considering sample complexity in Appendix D.4.

### D.1   Simplified Gradient Feature Learning Framework

---

**Algorithm 2** Training by Algorithm 1 with no updates for the first layer after the first gradient step

---

   Initialize $f_{(\mathbf{a}^{(0)}, \mathbf{W}^{(0)}, \mathbf{b})} \in \mathcal{F}_{d,m}$; Sample $\mathcal{Z} \sim \mathcal{D}^n$
   Get $(\mathbf{a}^{(1)}, \mathbf{W}^{(1)}, \mathbf{b})$ by one gradient step update and fix $\mathbf{W}^{(1)}, \mathbf{b}$
   **for** $t = 2$ **to** $T$ **do**
     $\mathbf{a}^{(t)} = \mathbf{a}^{(t-1)} - \eta^{(t)} \nabla_{\mathbf{a}} \widetilde{\mathcal{L}}_{\mathcal{Z}} \big( f_{\Xi^{(t-1)}} \big)$
   **end for**

---

**Theorem 3.4** (Simple Setting). *Assume $\widetilde{\mathcal{L}}_{\mathcal{Z}} \big( f_{(\mathbf{a}, \mathbf{W}^{(1)}, \mathbf{b})} \big)$ is L-smooth to $\mathbf{a}$. Let $\eta^{(t)} = \frac{1}{L}, \lambda^{(t)} = 0$, for all $t \in \{2, 3, \ldots, T\}$. Training by Algorithm 1 with no updates for the first layer after the*

*first gradient step, w.h.p., there exists $t \in [T]$ such that $\mathcal{L}_\mathcal{D}(f_{(\mathbf{a}^{(t)}, \mathbf{W}^{(1)}, \mathbf{b})}) \leq \mathrm{OPT}_{\mathbf{W}^{(1)}, \mathbf{b}, B_{a2}} +$*
$O\left( \frac{L(\|\mathbf{a}^{(1)}\|_2^2 + B_{a2}^2)}{T} + \sqrt{\frac{B_{a2}^2(\|\mathbf{W}^{(1)}\|_F^2 B_x^2 + \|\mathbf{b}\|_2^2)}{n}} \right).$

*Proof of Theorem 3.4.* Recall that

$$\mathcal{F}_{\mathbf{W}, \mathbf{b}, B_{a2}} := \left\{ f_{(\mathbf{a}, \mathbf{W}, \mathbf{b})} \in \mathcal{F}_{d,m} \mid \|\mathbf{a}\|_2 \leq B_{a2} \right\}, \qquad \mathrm{OPT}_{\mathbf{W}, \mathbf{b}, B_{a2}} := \min_{f \in \mathcal{F}_{\mathbf{W}, \mathbf{b}, B_{a2}}} \mathcal{L}_\mathcal{D}(f). \quad (13)$$

We denote $f^* = \mathrm{argmin}_{f \in \mathcal{F}_{\mathbf{W}, \mathbf{b}, B_{a2}}} \mathcal{L}_\mathcal{D}(f)$ and $\tilde{f}^* = \mathrm{argmin}_{f \in \mathcal{F}_{\mathbf{W}, \mathbf{b}, B_{a2}}} \widetilde{\mathcal{L}}_\mathcal{Z}(f)$. We use $\mathbf{a}^*$ and $\tilde{\mathbf{a}}^*$ to denote their second layer weights respectively. Then, we have

$$\mathcal{L}_\mathcal{D}(f_{(\mathbf{a}^{(t)}, \mathbf{W}^{(1)}, \mathbf{b})}) = \mathcal{L}_\mathcal{D}(f_{(\mathbf{a}^{(t)}, \mathbf{W}^{(1)}, \mathbf{b})}) - \widetilde{\mathcal{L}}_\mathcal{Z}(f_{(\mathbf{a}^{(t)}, \mathbf{W}^{(1)}, \mathbf{b})}) \tag{14}$$
$$+ \widetilde{\mathcal{L}}_\mathcal{Z}(f_{(\mathbf{a}^{(t)}, \mathbf{W}^{(1)}, \mathbf{b})}) - \widetilde{\mathcal{L}}_\mathcal{Z}(f_{(\tilde{\mathbf{a}}^*, \mathbf{W}^{(1)}, \mathbf{b})}) \tag{15}$$
$$+ \widetilde{\mathcal{L}}_\mathcal{Z}(f_{(\tilde{\mathbf{a}}^*, \mathbf{W}^{(1)}, \mathbf{b})}) - \widetilde{\mathcal{L}}_\mathcal{Z}(f_{(\mathbf{a}^*, \mathbf{W}^{(1)}, \mathbf{b})}) \tag{16}$$
$$+ \widetilde{\mathcal{L}}_\mathcal{Z}(f_{(\mathbf{a}^*, \mathbf{W}^{(1)}, \mathbf{b})}) - \mathcal{L}_\mathcal{D}(f_{(\mathbf{a}^*, \mathbf{W}^{(1)}, \mathbf{b})}) \tag{17}$$
$$+ \mathcal{L}_\mathcal{D}(f_{(\mathbf{a}^*, \mathbf{W}^{(1)}, \mathbf{b})}) \tag{18}$$
$$\leq \left| \mathcal{L}_\mathcal{D}(f_{(\mathbf{a}^{(t)}, \mathbf{W}^{(1)}, \mathbf{b})}) - \widetilde{\mathcal{L}}_\mathcal{Z}(f_{(\mathbf{a}^{(t)}, \mathbf{W}^{(1)}, \mathbf{b})}) \right| \tag{19}$$
$$+ \left| \widetilde{\mathcal{L}}_\mathcal{Z}(f_{(\mathbf{a}^{(t)}, \mathbf{W}^{(1)}, \mathbf{b})}) - \widetilde{\mathcal{L}}_\mathcal{Z}(f_{(\tilde{\mathbf{a}}^*, \mathbf{W}^{(1)}, \mathbf{b})}) \right| \tag{20}$$
$$+ 0 \tag{21}$$
$$+ \left| \widetilde{\mathcal{L}}_\mathcal{Z}(f_{(\mathbf{a}^*, \mathbf{W}^{(1)}, \mathbf{b})}) - \mathcal{L}_\mathcal{D}(f_{(\mathbf{a}^*, \mathbf{W}^{(1)}, \mathbf{b})}) \right| \tag{22}$$
$$+ \mathrm{OPT}_{\mathbf{W}^{(1)}, \mathbf{b}, B_{a2}}. \tag{23}$$

Fixing $\mathbf{W}^{(1)}, \mathbf{b}$ and optimizing $\mathbf{a}$ only is a convex optimization problem. Note that $\eta \leq \frac{1}{L}$, where $\widetilde{\mathcal{L}}_\mathcal{Z}$ is $L$-smooth to $\mathbf{a}$. Thus with gradient descent, we have

$$\frac{1}{T} \sum_{t=1}^{T} \widetilde{\mathcal{L}}_\mathcal{Z}\left( f_{(\mathbf{a}^{(t)}, \mathbf{W}^{(1)}, \mathbf{b})} \right) - \widetilde{\mathcal{L}}_\mathcal{Z}\left( f_{(\mathbf{a}^*, \mathbf{W}^{(1)}, \mathbf{b})} \right) \leq \frac{\|\mathbf{a}^{(1)} - \mathbf{a}^*\|_2^2}{2T\eta}. \tag{24}$$

Then our theorem gets proved by Lemma F.9 and generalization bounds based on Rademacher complexity. $\qquad\square$

### D.2 Gradient Feature Learning Framework under Expected Risk

We consider the following training process under population loss to simplify the proof. We prove our Gradient Feature Learning Framework under empirical loss considering sample complexity in Appendix D.4.

---
**Algorithm 3** Network Training via Gradient Descent

---
Initialize $(\mathbf{a}^{(0)}, \mathbf{W}^{(0)}, \mathbf{b})$ as in Equation (8)
**for** $t = 1$ **to** $T$ **do**
$\quad \mathbf{a}^{(t)} = \mathbf{a}^{(t-1)} - \eta^{(t)} \nabla_\mathbf{a} \mathcal{L}_\mathcal{D}^{\lambda^{(t)}}(f_{\Xi^{(t-1)}})$
$\quad \mathbf{W}^{(t)} = \mathbf{W}^{(t-1)} - \eta^{(t)} \nabla_\mathbf{W} \mathcal{L}_\mathcal{D}^{\lambda^{(t)}}(f_{\Xi^{(t-1)}})$
**end for**

---

Given an input distribution, we can get a Gradient Feature set $S_{p, \gamma, B_G}$ and $f^*(\mathbf{x}) = \sum_{j=1}^{r} \mathbf{a}_j^* \sigma(\langle \mathbf{w}_j^*, \mathbf{x} \rangle - \mathbf{b}_j^*)$, where $f^* \in \mathcal{F}_{d, r, B_F, S_{p, \gamma, B_G}}$ is a Gradient Feature Induced networks defined in Definition 3.11. Considering training by Algorithm 3, we have the following results.

**Theorem D.1** (Gradient Feature Learning Framework under Expected Risk). *Assume Assumption 3.1. For any $\epsilon, \delta \in (0,1)$, if $m \leq e^d$ and*

$$m = \Omega\left(\frac{1}{p}\left(\frac{rB_{a1}B_{x1}}{\epsilon}\sqrt{\frac{B_b}{B_G}}\right)^4 + \frac{1}{\sqrt{\delta}} + \frac{1}{p}\left(\log\left(\frac{r}{\delta}\right)\right)^2\right), \tag{25}$$

$$T = \Omega\left(\frac{1}{\epsilon}\left(\frac{\sqrt{r}B_{a2}B_bB_{x1}}{(mp)^{\frac{1}{4}}} + m\tilde{b}\right)\left(\frac{\sqrt{\log m}}{\sqrt{B_bB_G}} + \frac{1}{B_{x1}(mp)^{\frac{1}{4}}}\right)\right), \tag{26}$$

*then with proper hyper-parameter values, we have with probability $\geq 1 - \delta$, there exists $t \in [T]$ in Algorithm 3 with*

$$\Pr[\text{sign}(f_{\Xi^{(t)}}(\mathbf{x})) \neq y] \leq \mathcal{L}_\mathcal{D}\left(f_{\Xi^{(t)}}\right) \leq \text{OPT}_{d,r,B_F,S_{p,\gamma,B_G}} + rB_{a1}B_{x1}\sqrt{2\gamma} + \epsilon. \tag{27}$$

See the full statement and proof in Theorem D.9. Below, we show some lemmas used in the analysis of population loss.

### D.2.1 Feature Learning

We first show that a large subset of neurons has gradients at the first step as good features.

**Definition D.2** (Nice Gradients Set. Equivalent to Equation (9)). *We define*

$$G_{(D,+1),Nice} := \left\{i \in [m] : \left\langle \mathbf{w}_i^{(1)}, D\right\rangle > (1-\gamma)\left\|\mathbf{w}_i^{(1)}\right\|_2, \left\|\mathbf{w}_i^{(1)}\right\|_2 \geq \left|\eta^{(1)}\ell'(0)\mathbf{a}_i^{(0)}\right|B_G\right\}$$

$$G_{(D,-1),Nice} := \left\{i \in [2m]\setminus[m] : \left\langle \mathbf{w}_i^{(1)}, D\right\rangle > (1-\gamma)\left\|\mathbf{w}_i^{(1)}\right\|_2, \left\|\mathbf{w}_i^{(1)}\right\|_2 \geq \left|\eta^{(1)}\ell'(0)\mathbf{a}_i^{(0)}\right|B_G\right\}$$

*where $\gamma, B_G$ is the same in the Definition 3.7.*

**Lemma D.3** (Feature Emergence. Full Statement of Lemma 3.13). *Let $\lambda^{(1)} = \frac{1}{\eta^{(1)}}$. For any $r$ size subset $\{(D_1, s_1), \ldots, (D_r, s_r)\} \subseteq S_{p,\gamma,B_G}$, with probability at least $1 - 2re^{-cmp}$ where $c > 0$ is a universal constant, we have that for all $j \in [r]$, $|G_{(D_j,s_j),Nice}| \geq \frac{mp}{4}$.*

*Proof of Lemma D.3.* By symmetric initialization and Lemma F.1, we have for all $i \in [2m]$

$$\mathbf{w}_i^{(1)} = -\eta^{(1)}\ell'(0)\mathbf{a}_i^{(0)}\mathbb{E}_{(\mathbf{x},y)}\left[y\sigma'\left[\left\langle \mathbf{w}_i^{(0)}, \mathbf{x}\right\rangle - \mathbf{b}_i\right]\mathbf{x}\right] \tag{28}$$

$$= -\eta^{(1)}\ell'(0)\mathbf{a}_i^{(0)}G(\mathbf{w}_i^{(0)}, \mathbf{b}_i). \tag{29}$$

For all $j \in [r]$, as $(D_j, s_j) \in S_{p,\gamma,B_G}$, by Lemma F.3,
(1) if $s_j = +1$, for all $i \in [m]$, we have

$$\Pr\left[i \in G_{(D_j,s_j),Nice}\right] \tag{30}$$

$$= \Pr\left[\frac{\left\langle \mathbf{w}_i^{(1)}, D_j\right\rangle}{\left\|\mathbf{w}_i^{(1)}\right\|_2} > (1-\gamma), \left\|\mathbf{w}_i^{(1)}\right\|_2 \geq \left|\eta^{(1)}\ell'(0)\mathbf{a}_i^{(0)}\right|B_G\right] \tag{31}$$

$$= \Pr\left[\frac{\left\langle \mathbf{w}_i^{(1)}, D_j\right\rangle}{\left\|\mathbf{w}_i^{(1)}\right\|_2} > (1-\gamma), \left\|\mathbf{w}_i^{(1)}\right\|_2 \geq \left|\eta^{(1)}\ell'(0)\mathbf{a}_i^{(0)}\right|B_G, \frac{\mathbf{b}_i}{|\mathbf{b}_i|} = s_j\right] \tag{32}$$

$$\geq \Pr\left[G(\mathbf{w}_i^{(0)}, \mathbf{b}_i) \in \mathcal{C}_{D_j,\gamma}, \|G(\mathbf{w}_i^{(0)}, \mathbf{b}_i)\|_2 \geq B_G, \frac{\mathbf{b}_i}{|\mathbf{b}_i|} = s_j, \mathbf{a}_i^{(0)}\left\langle G(\mathbf{w}_i^{(0)}, \mathbf{b}_i), D_j\right\rangle > 0\right]$$

$$\geq \frac{p}{2}, \tag{33}$$

(2) if $s_j = -1$, for all $i \in [2m]\setminus[m]$, similarly we have

$$\Pr\left[i \in G_{(D_j,s_j),Nice}\right] \geq \frac{p}{2}. \tag{34}$$

By concentration inequality, (Chernoff's inequality under small deviations), we have

$$\Pr\left[|G_{(D_j,s_j),Nice}| < \frac{mp}{4}\right] \leq 2e^{-cmp}. \tag{35}$$

We complete the proof by union bound. $\qquad\square$

### D.2.2 Good Network Exists

Then, the gradients allow for obtaining a set of neurons approximating the "ground-truth" network with comparable loss.

**Lemma D.4** (Existence of Good Networks. Full Statement of Lemma 3.14). *Let* $\lambda^{(1)} = \frac{1}{\eta^{(1)}}$. *For any* $B_\epsilon \in (0, B_b)$, *let* $\sigma_a = \Theta\left(\frac{\tilde{b}}{-\ell'(0)\eta^{(1)}B_G B_\epsilon}\right)$ *and* $\delta = 2re^{-\sqrt{mp}}$. *Then, with probability at least* $1 - \delta$ *over the initialization, there exists* $\tilde{\mathbf{a}}_i$'s *such that* $f_{(\tilde{\mathbf{a}}, \mathbf{W}^{(1)}, \mathbf{b})}(\mathbf{x}) = \sum_{i=1}^{4m} \tilde{\mathbf{a}}_i \sigma\left(\left\langle \mathbf{w}_i^{(1)}, \mathbf{x}\right\rangle - \mathbf{b}_i\right)$ *satisfies*

$$\mathcal{L}_{\mathcal{D}}(f_{(\tilde{\mathbf{a}}, \mathbf{W}^{(1)}, \mathbf{b})}) \leq rB_{a1}\left(\frac{B_{x1}^2 B_b}{\sqrt{mp}B_G B_\epsilon} + B_{x1}\sqrt{2\gamma} + B_\epsilon\right) + \mathrm{OPT}_{d,r,B_F,S_{p,\gamma},B_G}, \tag{36}$$

*and* $\|\tilde{\mathbf{a}}\|_0 = O\left(r(mp)^{\frac{1}{2}}\right)$, $\|\tilde{\mathbf{a}}\|_2 = O\left(\frac{B_{a2}B_b}{\tilde{b}(mp)^{\frac{1}{4}}}\right)$, $\|\tilde{\mathbf{a}}\|_\infty = O\left(\frac{B_{a1}B_b}{\tilde{b}(mp)^{\frac{1}{2}}}\right)$.

*Proof of Lemma D.4.* Recall $f^*(\mathbf{x}) = \sum_{j=1}^r \mathbf{a}_j^* \sigma(\langle \mathbf{w}_j^*, \mathbf{x}\rangle - \mathbf{b}_j^*)$, where $f^* \in \mathcal{F}_{d,r,B_F,S_{p,\gamma},B_G}$ is defined in Definition 3.11 and let $s_j^* = \frac{\mathbf{b}_j^*}{|\mathbf{b}_j^*|}$. By Lemma D.3, with probability at least $1 - \delta_1$, $\delta_1 = 2re^{-cmp}$, for all $j \in [r]$, we have $|G_{(\mathbf{w}_j^*, s_j^*), Nice}| \geq \frac{mp}{4}$. Then for all $i \in G_{(\mathbf{w}_j^*, s_j^*), Nice} \subseteq [2m]$, we have $-\ell'(0)\eta^{(1)}G(\mathbf{w}_i^{(0)}, \mathbf{b}_i)\frac{\mathbf{b}_j^*}{\tilde{b}}$ only depend on $\mathbf{w}_i^{(0)}$ and $\mathbf{b}_i$, which is independent of $\mathbf{a}_i^{(0)}$. Given Definition 3.7, we have

$$-\ell'(0)\eta^{(1)}\|G(\mathbf{w}_i^{(0)}, \mathbf{b}_i)\|_2\frac{\mathbf{b}_j^*}{\tilde{b}} \in \left[\ell'(0)\eta^{(1)}B_{x1}\frac{B_b}{\tilde{b}}, -\ell'(0)\eta^{(1)}B_{x1}\frac{B_b}{\tilde{b}}\right]. \tag{37}$$

We split $[r]$ into $\Gamma = \{j \in [r] : |\mathbf{b}_j^*| < B_\epsilon\}$, $\Gamma_- = \{j \in [r] : \mathbf{b}_j^* \leq -B_\epsilon\}$ and $\Gamma_+ = \{j \in [r] : \mathbf{b}_j^* \geq B_\epsilon\}$. Let $\epsilon_a = \frac{B_{x1}B_b}{\sqrt{mp}B_G B_\epsilon}$. Then we know that for all $j \in \Gamma_+ \cup \Gamma_-$, for all $i \in G_{(\mathbf{w}_j^*, s_j^*), Nice}$, we have

$$\Pr_{\mathbf{a}_i^{(0)} \sim \mathcal{N}(0, \sigma_a^2)}\left[\left|-\mathbf{a}_i^{(0)}\ell'(0)\eta^{(1)}\|G(\mathbf{w}_i^{(0)}, \mathbf{b}_i)\|_2\frac{|\mathbf{b}_j^*|}{\tilde{b}} - 1\right| \leq \epsilon_a\right] \tag{38}$$

$$= \Pr_{\mathbf{a}_i^{(0)} \sim \mathcal{N}(0, \sigma_a^2)}\left[1 - \epsilon_a \leq -\mathbf{a}_i^{(0)}\ell'(0)\eta^{(1)}\|G(\mathbf{w}_i^{(0)}, \mathbf{b}_i)\|_2\frac{|\mathbf{b}_j^*|}{\tilde{b}} \leq 1 + \epsilon_a\right] \tag{39}$$

$$= \Pr_{g \sim \mathcal{N}(0,1)}\left[1 - \epsilon_a \leq g\Theta\left(\frac{\|G(\mathbf{w}_i^{(0)}, \mathbf{b}_i)\|_2|\mathbf{b}_j^*|}{B_G B_\epsilon}\right) \leq 1 + \epsilon_a\right] \tag{40}$$

$$= \Pr_{g \sim \mathcal{N}(0,1)}\left[(1 - \epsilon_a)\Theta\left(\frac{B_G B_\epsilon}{\|G(\mathbf{w}_i^{(0)}, \mathbf{b}_i)\|_2|\mathbf{b}_j^*|}\right) \leq g \leq (1 + \epsilon_a)\Theta\left(\frac{B_G B_\epsilon}{\|G(\mathbf{w}_i^{(0)}, \mathbf{b}_i)\|_2|\mathbf{b}_j^*|}\right)\right]$$

$$= \Theta\left(\frac{\epsilon_a B_G B_\epsilon}{\|G(\mathbf{w}_i^{(0)}, \mathbf{b}_i)\|_2|\mathbf{b}_j^*|}\right) \tag{41}$$

$$\geq \Omega\left(\frac{\epsilon_a B_G B_\epsilon}{B_{x1}B_b}\right) \tag{42}$$

$$= \Omega\left(\frac{1}{\sqrt{mp}}\right). \tag{43}$$

Thus, with probability $\Omega\left(\frac{1}{\sqrt{mp}}\right)$ over $\mathbf{a}_i^{(0)}$, we have

$$\left|-\mathbf{a}_i^{(0)}\ell'(0)\eta^{(1)}\|G(\mathbf{w}_i^{(0)}, \mathbf{b}_i)\|_2\frac{|\mathbf{b}_j^*|}{\tilde{b}} - 1\right| \leq \epsilon_a, \quad \left|\mathbf{a}_i^{(0)}\right| = O\left(\frac{\tilde{b}}{-\ell'(0)\eta^{(1)}B_G B_\epsilon}\right). \tag{44}$$

Similarly, for $j \in \Gamma$, for all $i \in G_{(\mathbf{w}_j^*, s_j^*), Nice}$, with probability $\Omega\left(\frac{1}{\sqrt{mp}}\right)$ over $\mathbf{a}_i^{(0)}$, we have

$$\left|-\mathbf{a}_i^{(0)}\ell'(0)\eta^{(1)}\|G(\mathbf{w}_i^{(0)}, \mathbf{b}_i)\|_2\frac{B_\epsilon}{\tilde{b}} - 1\right| \leq \epsilon_a, \quad \left|\mathbf{a}_i^{(0)}\right| = O\left(\frac{\tilde{b}}{-\ell'(0)\eta^{(1)}B_G B_\epsilon}\right). \tag{45}$$

For all $j \in [r]$, let $\Lambda_j \subseteq G_{(\mathbf{w}_j^*, s_j^*), Nice}$ be the set of $i$'s such that condition Equation (44) or Equation (45) are satisfied. By Chernoff bound and union bound, with probability at least $1 - \delta_2$, $\delta_2 = re^{-\sqrt{mp}}$, for all $j \in [r]$ we have $|\Lambda_j| \geq \Omega(\sqrt{mp})$.

We have for $\forall j \in \Gamma_+ \cup \Gamma_-, \forall i \in \Lambda_j$,

$$\left| \frac{|\mathbf{b}_j^*|}{\tilde{b}} \left\langle \mathbf{w}_i^{(1)}, \mathbf{x} \right\rangle - \left\langle \mathbf{w}_j^*, \mathbf{x} \right\rangle \right| \tag{46}$$

$$\leq \left\| -\mathbf{a}_i^{(0)} \ell'(0) \eta^{(1)} \| G(\mathbf{w}_i^{(0)}, \mathbf{b}_i)\|_2 \frac{|\mathbf{b}_j^*|}{\tilde{b}} \frac{\mathbf{w}_i^{(1)}}{\|\mathbf{w}_i^{(1)}\|_2} - \frac{\mathbf{w}_i^{(1)}}{\|\mathbf{w}_i^{(1)}\|_2} + \frac{\mathbf{w}_i^{(1)}}{\|\mathbf{w}_i^{(1)}\|_2} - \mathbf{w}_j^* \right\| \|\mathbf{x}\|_2 \tag{47}$$

$$\leq (\epsilon_a + \sqrt{2\gamma}) \|\mathbf{x}\|_2. \tag{48}$$

Similarly, for $\forall j \in \Gamma, \forall i \in \Lambda_j$,

$$\left| \frac{B_\epsilon}{\tilde{b}} \left\langle \mathbf{w}_i^{(1)}, \mathbf{x} \right\rangle - \left\langle \mathbf{w}_j^*, \mathbf{x} \right\rangle \right| \leq (\epsilon_a + \sqrt{2\gamma}) \|\mathbf{x}\|_2. \tag{49}$$

If $i \in \Lambda_j, j \in \Gamma_+ \cup \Gamma_-$, set $\tilde{\mathbf{a}}_i = \mathbf{a}_j^* \frac{|\mathbf{b}_j^*|}{|\Lambda_j|\tilde{b}}$, if $i \in \Lambda_j, j \in \Gamma$, set $\tilde{\mathbf{a}}_i = \mathbf{a}_j^* \frac{B_\epsilon}{|\Lambda_j|\tilde{b}}$, otherwise set $\tilde{\mathbf{a}}_i = 0$, we have $\|\tilde{\mathbf{a}}\|_0 = O\left(r(mp)^{\frac{1}{2}}\right), \|\tilde{\mathbf{a}}\|_2 = O\left(\frac{B_{a2}B_b}{\tilde{b}(mp)^{\frac{1}{4}}}\right), \|\tilde{\mathbf{a}}\|_\infty = O\left(\frac{B_{a1}B_b}{\tilde{b}(mp)^{\frac{1}{2}}}\right)$.

Finally, we have

$$\mathcal{L}_\mathcal{D}(f_{(\tilde{\mathbf{a}}, \mathbf{W}^{(1)}, \mathbf{b})}) \tag{50}$$

$$= \mathcal{L}_\mathcal{D}(f_{(\tilde{\mathbf{a}}, \mathbf{W}^{(1)}, \mathbf{b})}) - \mathcal{L}_\mathcal{D}(f^*) + \mathcal{L}_\mathcal{D}(f^*) \tag{51}$$

$$\leq \mathbb{E}_{(\mathbf{x}, y)} \left[ \left| f_{(\tilde{\mathbf{a}}, \mathbf{W}^{(1)}, \mathbf{b})}(\mathbf{x}) - f^*(\mathbf{x}) \right| \right] + \mathcal{L}_\mathcal{D}(f^*) \tag{52}$$

$$\leq \mathbb{E}_{(\mathbf{x}, y)} \left[ \left| \sum_{i=1}^m \tilde{\mathbf{a}}_i \sigma\left( \left\langle \mathbf{w}_i^{(1)}, \mathbf{x} \right\rangle - \tilde{b} \right) + \sum_{i=m+1}^{2m} \tilde{\mathbf{a}}_i \sigma\left( \left\langle \mathbf{w}_i^{(1)}, \mathbf{x} \right\rangle + \tilde{b} \right) - \sum_{j=1}^r \mathbf{a}_j^* \sigma(\left\langle \mathbf{w}_j^*, \mathbf{x} \right\rangle - \mathbf{b}_j^*) \right| \right]$$
$$+ \mathcal{L}_\mathcal{D}(f^*) \tag{53}$$

$$\leq \mathbb{E}_{(\mathbf{x}, y)} \left[ \left| \sum_{j \in \Gamma_+} \sum_{i \in \Lambda_j} \mathbf{a}_j^* \frac{1}{|\Lambda_j|} \left| \frac{|\mathbf{b}_j^*|}{\tilde{b}} \sigma\left( \left\langle \mathbf{w}_i^{(1)}, \mathbf{x} \right\rangle - \tilde{b} \right) - \sigma(\left\langle \mathbf{w}_j^*, \mathbf{x} \right\rangle - \mathbf{b}_j^*) \right| \right| \right] \tag{54}$$

$$+ \mathbb{E}_{(\mathbf{x}, y)} \left[ \left| \sum_{j \in \Gamma_-} \sum_{i \in \Lambda_j} \mathbf{a}_j^* \frac{1}{|\Lambda_j|} \left| \frac{|\mathbf{b}_j^*|}{\tilde{b}} \sigma\left( \left\langle \mathbf{w}_i^{(1)}, \mathbf{x} \right\rangle + \tilde{b} \right) - \sigma(\left\langle \mathbf{w}_j^*, \mathbf{x} \right\rangle - \mathbf{b}_j^*) \right| \right| \right] \tag{55}$$

$$+ \mathbb{E}_{(\mathbf{x}, y)} \left[ \left| \sum_{j \in \Gamma} \sum_{i \in \Lambda_j} \mathbf{a}_j^* \frac{1}{|\Lambda_j|} \left| \frac{B_\epsilon}{\tilde{b}} \sigma\left( \left\langle \mathbf{w}_i^{(1)}, \mathbf{x} \right\rangle - \tilde{b} \right) - \sigma(\left\langle \mathbf{w}_j^*, \mathbf{x} \right\rangle - \mathbf{b}_j^*) \right| \right| \right] + \mathcal{L}_\mathcal{D}(f^*) \tag{56}$$

$$\leq \mathbb{E}_{(\mathbf{x}, y)} \left[ \left| \sum_{j \in \Gamma_+} \sum_{i \in \Lambda_j} \mathbf{a}_j^* \frac{1}{|\Lambda_j|} \left| \frac{|\mathbf{b}_j^*|}{\tilde{b}} \left\langle \mathbf{w}_i^{(1)}, \mathbf{x} \right\rangle - \left\langle \mathbf{w}_j^*, \mathbf{x} \right\rangle \right| \right| \right] \tag{57}$$

$$+ \mathbb{E}_{(\mathbf{x}, y)} \left[ \left| \sum_{j \in \Gamma_-} \sum_{i \in \Lambda_j} \mathbf{a}_j^* \frac{1}{|\Lambda_j|} \left| \frac{|\mathbf{b}_j^*|}{\tilde{b}} \left\langle \mathbf{w}_i^{(1)}, \mathbf{x} \right\rangle - \left\langle \mathbf{w}_j^*, \mathbf{x} \right\rangle \right| \right| \right] \tag{58}$$

$$+ \mathbb{E}_{(\mathbf{x}, y)} \left[ \left| \sum_{j \in \Gamma} \sum_{i \in \Lambda_j} \mathbf{a}_j^* \frac{1}{|\Lambda_j|} \left| \frac{B_\epsilon}{\tilde{b}} \left\langle \mathbf{w}_i^{(1)}, \mathbf{x} \right\rangle + B_\epsilon - \left\langle \mathbf{w}_j^*, \mathbf{x} \right\rangle \right| \right| \right] + \mathcal{L}_\mathcal{D}(f^*) \tag{59}$$

$$\leq r\|\mathbf{a}^*\|_\infty (\epsilon_a + \sqrt{2\gamma}) \mathbb{E}_{(\mathbf{x}, y)} \|\mathbf{x}\|_2 + |\Gamma| \|\mathbf{a}^*\|_\infty B_\epsilon + \mathcal{L}_\mathcal{D}(f^*) \tag{60}$$

$$\leq r B_{x1} B_{a1} (\epsilon_a + \sqrt{2\gamma}) + |\Gamma| B_{a1} B_\epsilon + \text{OPT}_{d, r, B_F, S_{p, \gamma}, B_G}. \tag{61}$$

We finish the proof by union bound and $\delta \geq \delta_1 + \delta_2$. $\qquad\square$

### D.2.3 Learning an Accurate Classifier

We will use the following theorem from existing work to prove that gradient descent learns a good classifier (Theorem D.9). Theorem D.1 is simply a direct corollary of Theorem D.9.

**Theorem D.5** (Theorem 13 in [33]). *Fix some $\eta$, and let $f_1, \ldots, f_T$ be some sequence of convex functions. Fix some $\theta_1$, and assume we update $\theta_{t+1} = \theta_t - \eta \nabla f_t(\theta_t)$. Then for every $\theta^*$ the following holds:*

$$\frac{1}{T} \sum_{t=1}^{T} f_t(\theta_t) \leq \frac{1}{T} \sum_{t=1}^{T} f_t(\theta^*) + \frac{1}{2\eta T} \|\theta^*\|_2^2 + \|\theta_1\|_2 \frac{1}{T} \sum_{t=1}^{T} \|\nabla f_t(\theta_t)\|_2 + \eta \frac{1}{T} \sum_{t=1}^{T} \|\nabla f_t(\theta_t)\|_2^2.$$

To apply the theorem we first present a few lemmas bounding the change in the network during steps.

**Lemma D.6** (Bound of $\Xi^{(0)}, \Xi^{(1)}$). *Assume the same conditions as in Lemma D.4, and $d \geq \log m$, with probability at least $1 - \delta - \frac{1}{m^2}$ over the initialization, $\|\mathbf{a}^{(0)}\|_\infty = O\left(\frac{\tilde{b}\sqrt{\log m}}{-\ell'(0)\eta^{(1)}B_G B_\epsilon}\right)$, and for all $i \in [4m]$, we have $\|\mathbf{w}_i^{(0)}\|_2 = O\left(\sigma_w \sqrt{d}\right)$. Finally, $\|\mathbf{a}^{(1)}\|_\infty = O\left(-\eta^{(1)}\ell'(0)(B_{x1}\sigma_w\sqrt{d} + \tilde{b})\right)$, and for all $i \in [4m]$, $\|\mathbf{w}_i^{(1)}\|_2 = O\left(\frac{\tilde{b}\sqrt{\log m}B_{x1}}{B_G B_\epsilon}\right)$.*

*Proof of Lemma D.6.* By Lemma F.4, we have $\|\mathbf{a}^{(0)}\|_\infty = O\left(\frac{\tilde{b}\sqrt{\log m}}{-\ell'(0)\eta^{(1)}B_G B_\epsilon}\right)$ with probability at least $1 - \frac{1}{2m^2}$ by property of maximum i.i.d Gaussians. For any $i \in [4m]$, by Lemma F.5 and $d \geq \log m$, we have

$$\Pr\left(\frac{1}{\sigma_w^2}\left\|\mathbf{w}_i^{(0)}\right\|_2^2 \geq d + 2\sqrt{4d\log(m)} + 8\log(m)\right) \leq O\left(\frac{1}{m^4}\right). \tag{62}$$

Thus, by union bound, with probability at least $1 - \frac{1}{2m^2}$, for all $i \in [4m]$, we have $\|\mathbf{w}_i^{(0)}\|_2 = O\left(\sigma_w\sqrt{d}\right)$.

For all $i \in [4m]$, we have

$$|\mathbf{a}_i^{(1)}| = -\eta^{(1)}\ell'(0)\left|\mathbb{E}_{(\mathbf{x},y)}\left[y\left[\sigma\left(\left\langle\mathbf{w}_i^{(0)}, \mathbf{x}\right\rangle - b_i\right)\right]\right]\right| \tag{63}$$

$$\leq -\eta^{(1)}\ell'(0)(\|\mathbf{w}_i^{(0)}\|_2\mathbb{E}_{(\mathbf{x},y)}[\|\mathbf{x}\|_2] + \tilde{b}) \tag{64}$$

$$\leq O\left(-\eta^{(1)}\ell'(0)(B_{x1}\sigma_w\sqrt{d} + \tilde{b})\right). \tag{65}$$

$$\|\mathbf{w}_i^{(1)}\|_2 = -\eta^{(1)}\ell'(0)\left\|\mathbf{a}_i^{(0)}\mathbb{E}_{(\mathbf{x},y)}\left[y\sigma'\left[\left\langle\mathbf{w}_i^{(0)}, \mathbf{x}\right\rangle - b_i\right]\mathbf{x}\right]\right\|_2 \tag{66}$$

$$\leq O\left(\frac{\tilde{b}\sqrt{\log m}B_{x1}}{B_G B_\epsilon}\right). \tag{67}$$

$\square$

**Lemma D.7** (Bound of $\Xi^{(t)}$). *Assume the same conditions as in Lemma D.6, and let $\eta = \eta^{(t)}$ for all $t \in \{2, 3, \ldots, T\}$, $0 < T\eta B_{x1} \leq o(1)$, and $0 = \lambda = \lambda^{(t)}$ for all $t \in \{2, 3, \ldots, T\}$, for all $i \in [4m]$, we have*

$$|\mathbf{a}_i^{(t)}| \leq O\left(|\mathbf{a}_i^{(1)}| + \|\mathbf{w}_i^{(1)}\|_2 + \frac{\tilde{b}}{B_{x1}} + \eta\tilde{b}\right) \tag{68}$$

$$\|\mathbf{w}_i^{(t)} - \mathbf{w}_i^{(1)}\|_2 \leq O\left(t\eta B_{x1}|\mathbf{a}_i^{(1)}| + t\eta^2 B_{x1}^2\|\mathbf{w}_i^{(1)}\|_2 + t\eta^2 B_{x1}\tilde{b}\right). \tag{69}$$

*Proof of Lemma D.7.* For all $i \in [4m]$, by Lemma D.6,

$$|\mathbf{a}_i^{(t)}| = \left|(1-\eta\lambda)\mathbf{a}_i^{(t-1)} - \eta\mathbb{E}_{(\mathbf{x},y)}\left[\ell'(yf_{\Xi^{(t-1)}(\mathbf{x})})y\left[\sigma\left(\left\langle\mathbf{w}_i^{(t-1)},\mathbf{x}\right\rangle - \mathbf{b}_i\right)\right]\right]\right| \tag{70}$$

$$\leq \left|(1-\eta\lambda)\mathbf{a}_i^{(t-1)}\right| + \eta\left|\mathbb{E}_{(\mathbf{x},y)}\left[\left[\sigma\left(\left\langle\mathbf{w}_i^{(t-1)},\mathbf{x}\right\rangle - \mathbf{b}_i\right)\right]\right]\right| \tag{71}$$

$$\leq \left|\mathbf{a}_i^{(t-1)}\right| + \eta(B_{x1}\|\mathbf{w}_i^{(t-1)}\|_2 + \tilde{b}) \tag{72}$$

$$\leq \left|\mathbf{a}_i^{(t-1)}\right| + \eta B_{x1}\|\mathbf{w}_i^{(t-1)} - \mathbf{w}_i^{(1)}\|_2 + \eta B_{x1}\|\mathbf{w}_i^{(1)}\|_2 + \eta\tilde{b} \tag{73}$$

$$= \left|\mathbf{a}_i^{(t-1)}\right| + \eta B_{x1}\|\mathbf{w}_i^{(t-1)} - \mathbf{w}_i^{(1)}\|_2 + \eta Z_i, \tag{74}$$

where we denote $Z_i = B_{x1}\|\mathbf{w}_i^{(1)}\|_2 + \tilde{b}$. Then we give a bound of the first layer's weights change,

$$\|\mathbf{w}_i^{(t)} - \mathbf{w}_i^{(1)}\|_2 \tag{75}$$

$$= \left\|(1-\eta\lambda)\mathbf{w}_i^{(t-1)} - \eta\mathbf{a}_i^{(t-1)}\mathbb{E}_{(\mathbf{x},y)}\left[\ell'(yf_{\Xi^{(t-1)}(\mathbf{x})})y\sigma'\left[\left\langle\mathbf{w}_i^{(t-1)},\mathbf{x}\right\rangle - \mathbf{b}_i\right]\mathbf{x}\right] - \mathbf{w}_i^{(1)}\right\|_2 \tag{76}$$

$$\leq \|\mathbf{w}_i^{(t-1)} - \mathbf{w}_i^{(1)}\|_2 + \eta B_{x1}|\mathbf{a}_i^{(t-1)}|. \tag{77}$$

Combine two bounds, we can get

$$|\mathbf{a}_i^{(t)}| \leq |\mathbf{a}_i^{(t-1)}| + \eta Z_i + (\eta B_{x1})^2\sum_{l=1}^{t-2}|\mathbf{a}_i^{(l)}| \tag{78}$$

$$\Leftrightarrow \sum_{l=1}^{t}|\mathbf{a}_i^{(l)}| \leq 2\left(\sum_{l=1}^{t-1}|\mathbf{a}_i^{(l)}|\right) - (1-(\eta B_{x1})^2)\left(\sum_{l=1}^{t-2}|\mathbf{a}_i^{(l)}|\right) + \eta Z_i. \tag{79}$$

Let $h(1) = |\mathbf{a}_i^{(1)}|$, $h(2) = 2|\mathbf{a}_i^{(1)}| + \eta Z_i$ and $h(t+2) = 2h(t+1) - (1-(\eta B_{x1})^2)h(t) + \eta Z_i$ for $n \in \mathbb{N}_+$, by Lemma F.8, we have

$$h(t) = -\frac{Z_i}{\eta B_{x1}^2} + c_1(1-\eta B_{x1})^{(t-1)} + c_2(1+\eta B_{x1})^{(t-1)} \tag{80}$$

$$c_1 = \frac{1}{2}\left(|\mathbf{a}_i^{(1)}| + \frac{Z_i}{\eta B_{x1}^2} - \frac{|\mathbf{a}_i^{(1)}| + \eta Z_i}{\eta B_{x1}}\right) \tag{81}$$

$$c_2 = \frac{1}{2}\left(|\mathbf{a}_i^{(1)}| + \frac{Z_i}{\eta B_{x1}^2} + \frac{|\mathbf{a}_i^{(1)}| + \eta Z_i}{\eta B_{x1}}\right). \tag{82}$$

Thus, by $|c_1| \leq c_2$, and $0 < T\eta B_{x1} \leq o(1)$, we have

$$|\mathbf{a}_i^{(t)}| \leq h(t) - h(t-1) \tag{83}$$

$$= -\eta B_{x1}c_1(1-\eta B_{x1})^{(t-2)} + \eta B_{x1}c_2(1+\eta B_{x1})^{(t-2)} \tag{84}$$

$$\leq 2\eta B_{x1}c_2(1+\eta B_{x1})^t \tag{85}$$

$$\leq O(2\eta B_{x1}c_2). \tag{86}$$

Similarly, by binomial approximation, we also have

$$\|\mathbf{w}_i^{(t)} - \mathbf{w}_i^{(1)}\|_2 \leq \eta B_{x1}h(t-1) \tag{87}$$

$$= \eta B_{x1}\left(-\frac{Z_i}{\eta B_{x1}^2} + c_1(1-\eta B_{x1})^{(t-2)} + c_2(1+\eta B_{x1})^{(t-2)}\right) \tag{88}$$

$$\leq \eta B_{x1}O\left(-\frac{Z_i}{\eta B_{x1}^2} + c_1(1-(t-2)\eta B_{x1}) + c_2(1+(t-2)\eta B_{x1})\right) \tag{89}$$

$$\leq \eta B_{x1}O\left(-\frac{Z_i}{\eta B_{x1}^2} + c_1 + c_2 + (c_2-c_1)t\eta B_{x1}\right) \tag{90}$$

$$\leq \eta B_{x1}O\left(|\mathbf{a}_i^{(1)}| + \frac{|\mathbf{a}_i^{(1)}| + \eta Z_i}{\eta B_{x1}}t\eta B_{x1}\right) \tag{91}$$

$$\leq O\left((\eta|\mathbf{a}_i^{(1)}| + \eta^2 Z_i)tB_{x1}\right). \tag{92}$$

We finish the proof by plugging $Z_i, c_2$ into the bound. $\qquad\square$

**Lemma D.8** (Bound of Loss Gap and Gradient). *Assume the same conditions as in Lemma D.7, for all $t \in [T]$, we have*

$$|\mathcal{L}_\mathcal{D}(f_{(\tilde{\mathbf{a}},\mathbf{W}^{(t)},\mathbf{b})}) - \mathcal{L}_\mathcal{D}(f_{(\tilde{\mathbf{a}},\mathbf{W}^{(1)},\mathbf{b})})| \le B_{x1}\|\tilde{\mathbf{a}}\|_2\sqrt{\|\tilde{\mathbf{a}}\|_0}\max_{i\in[4m]}\|\mathbf{w}_i^{(t)} - \mathbf{w}_i^{(1)}\|_2 \qquad (93)$$

*and for all $t \in [T]$, for all $i \in [4m]$, we have*

$$\left|\frac{\partial\mathcal{L}_\mathcal{D}(f_{\Xi^{(t)}})}{\partial\mathbf{a}_i^{(t)}}\right| \le B_{x1}(\|\mathbf{w}_i^{(t)} - \mathbf{w}_i^{(1)}\|_2 + \|\mathbf{w}_i^{(1)}\|_2) + \tilde{b}. \qquad (94)$$

*Proof of Lemma D.8.* It follows from that

$$|\mathcal{L}_\mathcal{D}(f_{(\tilde{\mathbf{a}},\mathbf{W}^{(t)},\mathbf{b})}) - \mathcal{L}_\mathcal{D}(f_{(\tilde{\mathbf{a}},\mathbf{W}^{(1)},\mathbf{b})})| \qquad (95)$$

$$\le \mathbb{E}_{(\mathbf{x},y)}|f_{(\tilde{\mathbf{a}},\mathbf{W}^{(t)},\mathbf{b})}(\mathbf{x}) - f_{(\tilde{\mathbf{a}},\mathbf{W}^{(1)},\mathbf{b})}(\mathbf{x})| \qquad (96)$$

$$\le \mathbb{E}_{(\mathbf{x},y)}\left[\|\tilde{\mathbf{a}}\|_2\sqrt{\|\tilde{\mathbf{a}}\|_0}\max_{i\in[4m]}\left|\sigma\left[\left\langle\mathbf{w}_i^{(t)},\mathbf{x}\right\rangle - \mathbf{b}_i\right] - \sigma\left[\left\langle\mathbf{w}_i^{(1)},\mathbf{x}\right\rangle - \mathbf{b}_i\right]\right|\right] \qquad (97)$$

$$\le B_{x1}\|\tilde{\mathbf{a}}\|_2\sqrt{\|\tilde{\mathbf{a}}\|_0}\max_{i\in[4m]}\|\mathbf{w}_i^{(t)} - \mathbf{w}_i^{(1)}\|_2. \qquad (98)$$

Also, we have

$$\left|\frac{\partial\mathcal{L}_\mathcal{D}(f_{\Xi^{(t)}})}{\partial\mathbf{a}_i^{(t)}}\right| = \left|\mathbb{E}_{(\mathbf{x},y)}\left[\ell'(yf_{\Xi^{(t)}}(\mathbf{x}))y\left[\sigma\left(\left\langle\mathbf{w}_i^{(t)},\mathbf{x}\right\rangle - \mathbf{b}_i\right)\right]\right]\right| \qquad (99)$$

$$\le B_{x1}\|\mathbf{w}_i^{(t)}\|_2 + \tilde{b} \qquad (100)$$

$$\le B_{x1}(\|\mathbf{w}_i^{(t)} - \mathbf{w}_i^{(1)}\|_2 + \|\mathbf{w}_i^{(1)}\|_2) + \tilde{b}. \qquad (101)$$

$\qquad\square$

We are now ready to prove the main theorem.

**Theorem D.9** (Online Convex Optimization. Full Statement of Theorem D.1). *Consider training by Algorithm 3, and any $\delta \in (0,1)$. Assume $d \ge \log m$. Set*

$$\sigma_w > 0, \quad \tilde{b} > 0, \quad \eta^{(t)} = \eta, \ \lambda^{(t)} = 0 \text{ for all } t \in \{2,3,\ldots,T\},$$

$$\eta^{(1)} = \Theta\left(\frac{\min\{O(\eta), O(\eta\tilde{b})\}}{-\ell'(0)(B_{x1}\sigma_w\sqrt{d} + \tilde{b})}\right), \quad \lambda^{(1)} = \frac{1}{\eta^{(1)}}, \quad \sigma_a = \Theta\left(\frac{\tilde{b}(mp)^{\frac{1}{4}}}{-\ell'(0)\eta^{(1)}B_{x1}\sqrt{B_G B_b}}\right).$$

*Let $0 < T\eta B_{x1} \le o(1)$, $m = \Omega\left(\frac{1}{\sqrt{\delta}} + \frac{1}{p}\left(\log\left(\frac{r}{\delta}\right)\right)^2\right)$. With probability at least $1 - \delta$ over the initialization, there exists $t \in [T]$ such that*

$$\mathcal{L}_\mathcal{D}\left(f_{\Xi^{(t)}}\right) \le \text{OPT}_{d,r,B_F,S_{p,\gamma,B_G}} + rB_{a1}\left(\frac{2B_{x1}}{(mp)^{\frac{1}{4}}}\sqrt{\frac{B_b}{B_G}} + B_{x1}\sqrt{2\gamma}\right) \qquad (102)$$

$$+ \eta\left(\sqrt{r}B_{a2}B_b T\eta B_{x1}^2 + m\tilde{b}\right)O\left(\frac{\sqrt{\log m}B_{x1}(mp)^{\frac{1}{4}}}{\sqrt{B_b B_G}} + 1\right) + O\left(\frac{B_{a2}^2 B_b^2}{\eta T\tilde{b}^2(mp)^{\frac{1}{2}}}\right).$$

*Furthermore, for any $\epsilon \in (0,1)$, set*

$$\tilde{b} = \Theta\left(\frac{B_G^{\frac{1}{4}}B_{a2}B_b^{\frac{3}{4}}}{\sqrt{r}B_{a1}}\right), \quad m = \Omega\left(\frac{1}{p\epsilon^4}\left(rB_{a1}B_{x1}\sqrt{\frac{B_b}{B_G}}\right)^4 + \frac{1}{\sqrt{\delta}} + \frac{1}{p}\left(\log\left(\frac{r}{\delta}\right)\right)^2\right), \quad (103)$$

$$\eta = \Theta\left(\frac{\epsilon}{\left(\frac{\sqrt{r}B_{a2}B_b B_{x1}}{(mp)^{\frac{1}{4}}} + m\tilde{b}\right)\left(\frac{\sqrt{\log m}B_{x1}(mp)^{\frac{1}{4}}}{\sqrt{B_b B_G}} + 1\right)}\right), \quad T = \Theta\left(\frac{1}{\eta B_{x1}(mp)^{\frac{1}{4}}}\right), \quad (104)$$

*we have there exists $t \in [T]$ with*

$$\Pr[\text{sign}(f_{\Xi^{(t)}})(\mathbf{x}) \ne y] \le \mathcal{L}_\mathcal{D}\left(f_{\Xi^{(t)}}\right) \le \text{OPT}_{d,r,B_F,S_{p,\gamma,B_G}} + rB_{a1}B_{x1}\sqrt{2\gamma} + \epsilon. \qquad (105)$$

*Proof of Theorem D.9.* By $m = \Omega\left(\frac{1}{\sqrt{\delta}} + \frac{1}{p}\left(\log\left(\frac{r}{\delta}\right)\right)^2\right)$ we have $2re^{-\sqrt{mp}} + \frac{1}{m^2} \leq \delta$. For any $B_\epsilon \in (0, B_b)$, when $\sigma_a = \Theta\left(\frac{\tilde{b}}{-\ell'(0)\eta^{(1)}B_G B_\epsilon}\right)$, by Theorem D.5, Lemma D.4, Lemma D.8, with probability at least $1 - \delta$ over the initialization, we have

$$\frac{1}{T}\sum_{t=1}^{T}\mathcal{L}_{\mathcal{D}}\left(f_{\Xi^{(t)}}\right) \tag{106}$$

$$\leq \frac{1}{T}\sum_{t=1}^{T}\left|\left(\mathcal{L}_{\mathcal{D}}(f_{(\tilde{\mathbf{a}},\mathbf{W}^{(t)},\mathbf{b})}) - \mathcal{L}_{\mathcal{D}}(f_{(\tilde{\mathbf{a}},\mathbf{W}^{(1)},\mathbf{b})})\right| + \mathcal{L}_{\mathcal{D}}(f_{(\tilde{\mathbf{a}},\mathbf{W}^{(1)},\mathbf{b})})\right) \tag{107}$$

$$+ \frac{\|\tilde{\mathbf{a}}\|_2^2}{2\eta T} + (2\|\mathbf{a}^{(1)}\|_2\sqrt{m} + 4\eta m)\max_{i\in[4m]}\left|\frac{\partial\mathcal{L}_{\mathcal{D}}(f_{\Xi^{(T)}})}{\partial\mathbf{a}_i^{(T)}}\right| \tag{108}$$

$$\leq \text{OPT}_{d,r,B_F,S_{p,\gamma},B_G} + rB_{a1}\left(\frac{B_{x1}^2 B_b}{\sqrt{mp}B_G B_\epsilon} + B_{x1}\sqrt{2\gamma} + B_\epsilon\right) \tag{109}$$

$$+ B_{x1}\|\tilde{\mathbf{a}}\|_2\sqrt{\|\tilde{\mathbf{a}}\|_0}\max_{i\in[4m]}\|\mathbf{w}_i^{(T)} - \mathbf{w}_i^{(1)}\|_2 \tag{110}$$

$$+ \frac{\|\tilde{\mathbf{a}}\|_2^2}{2\eta T} + 4mB_{x1}(\|\mathbf{a}^{(1)}\|_\infty + \eta)\left(\max_{i\in[4m]}\|\mathbf{w}_i^{(T)} - \mathbf{w}_i^{(1)}\|_2 + \max_{i\in[4m]}\|\mathbf{w}_i^{(1)}\|_2 + \frac{\tilde{b}}{B_{x1}}\right). \tag{111}$$

By Lemma D.4, Lemma D.6, Lemma D.7, when $\eta^{(1)} = \Theta\left(\frac{\min\{O(\eta),O(\eta\tilde{b})\}}{-\ell'(0)(B_{x1}\sigma_w\sqrt{d}+\tilde{b})}\right)$, we have

$$\|\tilde{\mathbf{a}}\|_0 = O\left(r(mp)^{\frac{1}{2}}\right), \quad \|\tilde{\mathbf{a}}\|_2 = O\left(\frac{B_{a2}B_b}{\tilde{b}(mp)^{\frac{1}{4}}}\right) \tag{112}$$

$$\|\mathbf{a}^{(1)}\|_\infty = O\left(-\eta^{(1)}\ell'(0)(B_{x1}\sigma_w\sqrt{d} + \tilde{b})\right) \tag{113}$$

$$= \min\{O(\eta), O(\eta\tilde{b})\} \tag{114}$$

$$\max_{i\in[4m]}\|\mathbf{w}_i^{(1)}\|_2 = O\left(\frac{\tilde{b}\sqrt{\log m}B_{x1}}{B_G B_\epsilon}\right) \tag{115}$$

$$\max_{i\in[4m]}\|\mathbf{w}_i^{(T)} - \mathbf{w}_i^{(1)}\|_2 = O\left(T\eta B_{x1}\|\mathbf{a}^{(1)}\|_\infty + T\eta^2 B_{x1}^2\max_{i\in[4m]}\|\mathbf{w}_i^{(1)}\|_2 + T\eta^2 B_{x1}\tilde{b}\right) \tag{116}$$

$$= O\left(T\eta^2 B_{x1}^2\left(\max_{i\in[4m]}\|\mathbf{w}_i^{(1)}\|_2 + \frac{\tilde{b}}{B_{x1}}\right)\right). \tag{117}$$

Set $B_\epsilon = \frac{B_{x1}}{(mp)^{\frac{1}{4}}}\sqrt{\frac{B_b}{B_G}}$, we have $\sigma_a = \Theta\left(\frac{\tilde{b}(mp)^{\frac{1}{4}}}{-\ell'(0)\eta^{(1)}B_{x1}\sqrt{B_G B_b}}\right)$ which satisfy the requirements. Then,

$$\frac{1}{T}\sum_{t=1}^{T}\mathcal{L}_{\mathcal{D}}\left(f_{\Xi^{(t)}}\right) \tag{118}$$

$$\leq \text{OPT}_{d,r,B_F,S_{p,\gamma},B_G} + rB_{a1}\left(\frac{2B_{x1}}{(mp)^{\frac{1}{4}}}\sqrt{\frac{B_b}{B_G}} + B_{x1}\sqrt{2\gamma}\right) \tag{119}$$

$$+ \left(\sqrt{r}B_{a2}B_b T\eta^2 B_{x1}^2\frac{B_{x1}}{\tilde{b}} + m\eta B_{x1}\right)O\left(\frac{\tilde{b}\sqrt{\log m}B_{x1}}{B_G B_\epsilon} + \frac{\tilde{b}}{B_{x1}}\right) + O\left(\frac{B_{a2}^2 B_b^2}{\eta T\tilde{b}^2(mp)^{\frac{1}{2}}}\right)$$

$$\leq \text{OPT}_{d,r,B_F,S_{p,\gamma},B_G} + rB_{a1}\left(\frac{2B_{x1}}{(mp)^{\frac{1}{4}}}\sqrt{\frac{B_b}{B_G}} + B_{x1}\sqrt{2\gamma}\right) \tag{120}$$

$$+ \eta\left(\sqrt{r}B_{a2}B_b T\eta B_{x1}^2 + m\tilde{b}\right)O\left(\frac{\sqrt{\log m}B_{x1}(mp)^{\frac{1}{4}}}{\sqrt{B_b B_G}} + 1\right) + O\left(\frac{B_{a2}^2 B_b^2}{\eta T\tilde{b}^2(mp)^{\frac{1}{2}}}\right). \tag{121}$$

Furthermore, for any $\epsilon \in (0,1)$, set

$$\tilde{b} = \Theta\left(\frac{B_G^{\frac{1}{4}} B_{a2} B_b^{\frac{3}{4}}}{\sqrt{rB_{a1}}}\right), \quad m = \Omega\left(\frac{1}{p\epsilon^4}\left(rB_{a1}B_{x1}\sqrt{\frac{B_b}{B_G}}\right)^4 + \frac{1}{\sqrt{\delta}} + \frac{1}{p}\left(\log\left(\frac{r}{\delta}\right)\right)^2\right), \quad (122)$$

$$\eta = \Theta\left(\frac{\epsilon}{\left(\frac{\sqrt{r}B_{a2}B_bB_{x1}}{(mp)^{\frac{1}{4}}} + m\tilde{b}\right)\left(\frac{\sqrt{\log m}B_{x1}(mp)^{\frac{1}{4}}}{\sqrt{B_bB_G}} + 1\right)}\right), \quad T = \Theta\left(\frac{1}{\eta B_{x1}(mp)^{\frac{1}{4}}}\right), \quad (123)$$

we have

$$\frac{1}{T}\sum_{t=1}^{T}\mathcal{L}_{\mathcal{D}}\left(f_{\Xi^{(t)}}\right) \leq \mathrm{OPT}_{d,r,B_F,S_{p,\gamma,B_G}} + rB_{a1}\left(\frac{2B_{x1}}{(mp)^{\frac{1}{4}}}\sqrt{\frac{B_b}{B_G}} + B_{x1}\sqrt{2\gamma}\right) + \frac{\epsilon}{2} \quad (124)$$

$$+ O\left(\frac{B_{x1}B_{a2}^2B_b^2}{\tilde{b}^2(mp)^{\frac{1}{4}}}\right) \quad (125)$$

$$\leq \mathrm{OPT}_{d,r,B_F,S_{p,\gamma,B_G}} + rB_{a1}B_{x1}\sqrt{2\gamma} + \epsilon. \quad (126)$$

We finish the proof as the 0-1 classification error is bounded by the loss function, e.g., $\mathbb{I}[\mathrm{sign}(f(\mathbf{x})) \neq y] \leq \frac{\ell(yf(\mathbf{x}))}{\ell(0)}$, where $\ell(0) = 1$. $\qquad\square$

### D.3 More Discussion abut Setting

**Range of $\sigma_w$.** In practice, the value of $\sigma_w$ cannot be arbitrary, because its choice will have an effect on the Gradient Feature set $S_{p,\gamma,B_G}$. On the other hand, $d \geq \log m$ is a natural assumption, otherwise, the two-layer neural networks may fall in the NTK regime.

**Parameter Choice.** We use $\lambda = 1/\eta$ in the first step so that the neural network will totally forget its initialization, leading to the feature emergence here. This is a common setting for analysis convenience in previous work, e.g., [32, 33, 105]. We can extend this to other choices (e.g., small initialization and large step size for the first few steps), as long as after the gradient update, the gradient dominates the neuron weights. We use $\lambda = 0$ afterward as the regularization effect is weak in our analysis. We can extend our analysis to $\lambda$ being a small value.

**Early Stopping.** Our analysis divides network learning into two stages: the feature learning stage, and then classifier learning over the good features. The feature learning stage is simplified to one gradient step for the convenience of analysis, while in practice feature learning can happen in multiple steps. The current framework focuses on the gradient features in the early gradient steps, while feature learning can also happen in later steps, in particular for more complicated data. It is an interesting direction to extend the analysis to a longer training horizon.

**Role of $s$.** The $s$ encodes the sign of the bias term, which is important. Recall that we do not update the bias term for simplicity. Let's consider a simple toy example. Assume we have $f_1(x) = a_1\sigma(w_1^\top x + 1)$, $f_2(x) = a_2\sigma(w_2^\top x - 1)$ and $f_3(x) = a_3\sigma(w_3^\top x + 2)$, where $\sigma$ is ReLU activation function which is a homogeneous function.

1. The sign of the bias term is important. We can see that we always have $a_1\sigma(w_1^\top x + 1) \neq a_2\sigma(w_2^\top x - 1)$ for any $a_1, w_1, a_2, w_2$. This means that $f_1(x)$ and $f_2(x)$ are intrinsically different and have different active patterns. Thus, we need to handle the sign of the bias term carefully.
2. The scaling of the bias is absorbed. On the other hand, we can see that $a_1\sigma(w_1^\top x + 1) = a_3\sigma(w_3^\top x + 2)$ when $a_1 = 2a_3, 2w_1 = w_3$. It means that the scale of the bias term is less important, which can be absorbed into other terms.

Thus, we only need to handle bias with different signs carefully.

**Gradient Feature Distribution.** We may define a gradient feature distribution rather than a gradient feature set. However, we find that the technical tools used in this continuous setting are pretty different from the discrete version.

**Activation Functions.** We can change the ReLU activation function to a sublinear activation function, e.g. leaky ReLU, sigmoid, to get a similar conclusion. First, we need to introduce a corresponding gradient feature set, and then we can make it by following the same analysis pipeline. For simplicity, we present ReLU only.

## D.4 Gradient Feature Learning Framework under Empirical Risk with Sample Complexity

In this section, we consider training with empirical risk. Intuitively, the proof is straightforward from the proof for population loss. We can simply replace the population loss with the empirical loss, which will introduce an error term in the gradient analysis. We use concentration inequality to control the error term and show that the error term depends inverse-polynomially on the sample size $n$.

**Definition D.10** (Empirical Simplified Gradient Vector). *Recall* $\mathcal{Z} = \{(\mathbf{x}^{(l)}, y^{(l)})\}_{l \in [n]}$, *for any* $\mathbf{w} \in \mathbb{R}^d$, $b \in \mathbb{R}$, *an Empirical Simplified Gradient Vector is defined as*

$$\widetilde{G}(\mathbf{w}, b) := \frac{1}{n} \sum_{l \in [n]} [y^{(l)} \mathbf{x}^{(l)} \mathbb{I}[\mathbf{w}^\top \mathbf{x}^{(l)} > b]]. \tag{127}$$

**Definition D.11** (Empirical Gradient Feature). *Recall* $\mathcal{Z} = \{(\mathbf{x}^{(l)}, y^{(l)})\}_{l \in [n]}$, *let* $\mathbf{w} \in \mathbb{R}^d$, $b \in \mathbb{R}$ *be random variables drawn from some distribution* $\mathcal{W}, \mathcal{B}$. *An Empirical Gradient Feature set with parameters* $p, \gamma, B_G$ *is defined as:*

$$\widetilde{S}_{p,\gamma,B_G}(\mathcal{W}, \mathcal{B}) := \left\{ (D, s) \ \middle| \ \Pr_{\mathbf{w}, b} \left[ \widetilde{G}(\mathbf{w}, b) \in \mathcal{C}_{D,\gamma} \text{ and } \|\widetilde{G}(\mathbf{w}, b)\|_2 \geq B_G \text{ and } s = \frac{b}{|b|} \right] \geq p \right\}.$$

*When clear from context, write it as* $\widetilde{S}_{p,\gamma,B_G}$.

Considering training by Algorithm 1, we have the following results.

**Theorem 3.12** (Main Result). *Assume Assumption 3.1. For any* $\epsilon, \delta \in (0, 1)$, *if* $m \leq e^d$ *and*

$$m = \Omega \left( \frac{1}{p\epsilon^4} \left( r B_{a1} B_{x1} \sqrt{\frac{B_b}{B_G}} \right)^4 + \frac{1}{\sqrt{\delta}} + \frac{1}{p} \left( \log \left( \frac{r}{\delta} \right) \right)^2 \right),$$

$$T = \Omega \left( \frac{1}{\epsilon} \left( \frac{\sqrt{r} B_{a2} B_b B_{x1}}{(mp)^{\frac{1}{4}}} + m\tilde{b} \right) \left( \frac{\sqrt{\log m}}{\sqrt{B_b B_G}} + \frac{1}{B_{x1}(mp)^{\frac{1}{4}}} \right) \right),$$

$$\frac{n}{\log n} = \tilde{\Omega} \left( \frac{m^3 p B_x^2 B_{a2}^4 B_b}{\epsilon^2 r^2 B_{a1}^2 B_G} + \frac{(mp)^{\frac{1}{2}} B_{x2}}{B_b B_G} + \frac{B_x^2}{B_{x2}} + \frac{1}{p} + \left( \frac{1}{B_G^2} + \frac{1}{B_{x1}^2} \right) \frac{B_{x2}}{|\ell'(0)|^2} + \frac{Tm}{\delta} \right),$$

*then with initialization (8) and proper hyper-parameter values, we have with probability* $\geq 1 - \delta$ *over the initialization and training samples, there exists* $t \in [T]$ *in Algorithm 1 with:*

$$\Pr[\text{sign}(f_{\Xi^{(t)}}(\mathbf{x})) \neq y] \leq \mathcal{L}_\mathcal{D}\left(f_{\Xi^{(t)}}\right)$$

$$\leq \text{OPT}_{d,r,B_F,S_{p,\gamma,B_G}} + r B_{a1} B_{x1} \sqrt{2\gamma + O\left( \frac{\sqrt{B_{x2} \log n}}{B_G |\ell'(0)| n^{\frac{1}{2}}} \right)} + \epsilon.$$

See the full statement and proof in Theorem D.17. Below, we show some lemmas used in the analysis under empirical loss.

**Lemma D.12** (Empirical Gradient Concentration Bound). *When* $\frac{n}{\log n} > \frac{B_x^2}{B_{x2}}$, *with probability at least* $1 - O\left(\frac{1}{n}\right)$ *over training samples, for all* $i \in [4m]$, *we have*

$$\left\| \frac{\partial \widetilde{\mathcal{L}}_\mathcal{Z}(f_\Xi)}{\partial \mathbf{w}_i} - \frac{\partial \mathcal{L}_\mathcal{D}(f_\Xi)}{\partial \mathbf{w}_i} \right\|_2 \leq O\left( \frac{|\mathbf{a}_i| \sqrt{B_{x2} \log n}}{n^{\frac{1}{2}}} \right), \tag{128}$$

$$\left| \frac{\partial \widetilde{\mathcal{L}}_\mathcal{Z}(f_\Xi)}{\partial \mathbf{a}_i} - \frac{\partial \mathcal{L}_\mathcal{D}(f_\Xi)}{\partial \mathbf{a}_i} \right| \leq O\left( \frac{\|\mathbf{w}_i\|_2 \sqrt{B_{x2} \log n}}{n^{\frac{1}{2}}} \right), \tag{129}$$

$$\left| \widetilde{\mathcal{L}}_\mathcal{Z}(f_\Xi) - \mathcal{L}_\mathcal{D}(f_\Xi) \right| \leq O\left( \frac{\left( \|\mathbf{a}\|_0 \|\mathbf{a}\|_\infty (\max_{i \in [4m]} \|\mathbf{w}_i\|_2 B_x + \tilde{b}) + 1 \right) \sqrt{\log n}}{n^{\frac{1}{2}}} \right). \tag{130}$$

*Proof of Lemma D.12.* First, we define,

$$\mathbf{z}^{(l)} = \ell'(y^{(l)} f_{\Xi}(\mathbf{x}^{(l)})) y^{(l)} \left[ \sigma' \left( \left\langle \mathbf{w}_i, \mathbf{x}^{(l)} \right\rangle - \mathbf{b}_i \right) \mathbf{x}^{(l)} \right] \tag{131}$$

$$- \mathbb{E}_{(\mathbf{x},y)} \left[ \ell'(y f_{\Xi}(\mathbf{x})) y \left[ \sigma' \left( \langle \mathbf{w}_i, \mathbf{x} \rangle - \mathbf{b}_i \right) \right] \mathbf{x} \right]. \tag{132}$$

As $|\ell'(z)| \leq 1, |y| \leq 1, |\sigma'(z)| \leq 1$, we have $\mathbf{z}^{(l)}$ is zero-mean random vector with $\left\| \mathbf{z}^{(l)} \right\|_2 \leq 2B_x$ as well as $\mathbb{E}\left[ \left\| \mathbf{z}^{(l)} \right\|_2^2 \right] \leq B_{x2}$. Then by Vector Bernstein Inequality, Lemma 18 in [66], for $0 < z < \frac{B_{x2}}{B_x}$ we have

$$\Pr\left( \left\| \frac{\partial \widetilde{\mathcal{L}}_{\mathcal{Z}}(f_{\Xi})}{\partial \mathbf{w}_i} - \frac{\partial \mathcal{L}_{\mathcal{D}}(f_{\Xi})}{\partial \mathbf{w}_i} \right\|_2 \geq |\mathbf{a}_i| z \right) = \Pr\left( \left\| \frac{1}{n} \sum_{l \in [n]} \mathbf{z}^{(l)} \right\|_2 \geq z \right) \tag{133}$$

$$\leq \exp\left( -n \cdot \frac{z^2}{8B_{x2}} + \frac{1}{4} \right). \tag{134}$$

Thus, let $z = n^{-\frac{1}{2}} \sqrt{B_{x2} \log n}$, with probability at least $1 - O\left(\frac{1}{n}\right)$, we have

$$\left\| \frac{\partial \widetilde{\mathcal{L}}_{\mathcal{Z}}(f_{\Xi})}{\partial \mathbf{w}_i} - \frac{\partial \mathcal{L}_{\mathcal{D}}(f_{\Xi})}{\partial \mathbf{w}_i} \right\|_2 \leq O\left( \frac{|\mathbf{a}_i| \sqrt{B_{x2} \log n}}{n^{\frac{1}{2}}} \right). \tag{135}$$

On the other hand, by Bernstein Inequality, for $z > 0$ we have

$$\Pr\left( \left| \frac{\partial \widetilde{\mathcal{L}}_{\mathcal{Z}}(f_{\Xi})}{\partial \mathbf{a}_i} - \frac{\partial \mathcal{L}_{\mathcal{D}}(f_{\Xi})}{\partial \mathbf{a}_i} \right| > z \| \mathbf{w}_i \|_2 \right) \tag{136}$$

$$= \Pr\left( \left| \frac{1}{n} \sum_{l \in [n]} \left( \ell'(y^{(l)} f_{\Xi}(\mathbf{x}^{(l)})) y^{(l)} \left[ \sigma \left( \left\langle \mathbf{w}_i, \mathbf{x}^{(l)} \right\rangle - \mathbf{b}_i \right) \right] \right. \right. \right. \tag{137}$$

$$\left. \left. \left. - \mathbb{E}_{(\mathbf{x},y)} \left[ \ell'(y f_{\Xi}(\mathbf{x})) y \left[ \sigma \left( \langle \mathbf{w}_i, \mathbf{x} \rangle - \mathbf{b}_i \right) \right] \right] \right) \right| > z \| \mathbf{w}_i \|_2 \right) \tag{138}$$

$$\leq 2 \exp\left( -\frac{\frac{1}{2} n z^2}{B_{x2} + \frac{1}{3} B_x z} \right). \tag{139}$$

Thus, when $\frac{n}{\log n} > \frac{B_x^2}{B_{x2}}$, let $z = n^{-\frac{1}{2}} \sqrt{B_{x2} \log n}$, with probability at least $1 - O\left(\frac{1}{n}\right)$, we have

$$\left| \frac{\partial \widetilde{\mathcal{L}}_{\mathcal{Z}}(f_{\Xi})}{\partial \mathbf{a}_i} - \frac{\partial \mathcal{L}_{\mathcal{D}}(f_{\Xi})}{\partial \mathbf{a}_i} \right| \leq O\left( \frac{\| \mathbf{w}_i \|_2 \sqrt{B_{x2} \log n}}{n^{\frac{1}{2}}} \right). \tag{140}$$

Finally, we have

$$\left| \widetilde{\mathcal{L}}_{\mathcal{Z}} \left( f_{\Xi} \right) - \mathcal{L}_{\mathcal{D}} \left( f_{\Xi} \right) \right| \tag{141}$$

$$= \left| \frac{1}{n} \sum_{l=1}^n \left( \ell \left( y^{(l)} \mathbf{a}^\top \left[ \sigma(\mathbf{W}^\top \mathbf{x}^{(l)} - \mathbf{b}) \right] \right) - \mathbb{E}_{(\mathbf{x},y) \sim \mathcal{D}} \left[ \ell \left( y \mathbf{a}^\top \left[ \sigma(\mathbf{W}^\top \mathbf{x} - \mathbf{b}) \right] \right) \right] \right) \right|. \tag{142}$$

By Assumption 3.1, we have $\ell \left( y^{(l)} \mathbf{a}^\top \left[ \sigma(\mathbf{W}^\top \mathbf{x}^{(l)} - \mathbf{b}) \right] \right) - \mathbb{E}_{(\mathbf{x},y) \sim \mathcal{D}} \left[ \ell \left( y \mathbf{a}^\top \left[ \sigma(\mathbf{W}^\top \mathbf{x} - \mathbf{b}) \right] \right) \right]$ is a zero-mean random variable, with bound $2 \| \mathbf{a} \|_0 \| \mathbf{a} \|_\infty (\max_{i \in [4m]} \| \mathbf{w}_i \|_2 B_x + \tilde{b}) + 2$. By Hoeffding's inequality, for all $z > 0$, we have

$$\Pr\left( \left| \widetilde{\mathcal{L}}_{\mathcal{Z}} \left( f_{\Xi} \right) - \mathcal{L}_{\mathcal{D}} \left( f_{\Xi} \right) \right| \geq z \right) \leq 2 \exp\left( -\frac{z^2 n}{(\| \mathbf{a} \|_0 \| \mathbf{a} \|_\infty (\max_{i \in [4m]} \| \mathbf{w}_i \|_2 B_x + \tilde{b}) + 1)^2} \right).$$

Thus, with probability at least $1 - O\left(\frac{1}{n}\right)$, we have

$$\left| \widetilde{\mathcal{L}}_{\mathcal{Z}} \left( f_{\Xi} \right) - \mathcal{L}_{\mathcal{D}} \left( f_{\Xi} \right) \right| \leq O\left( \frac{\left( \| \mathbf{a} \|_0 \| \mathbf{a} \|_\infty (\max_{i \in [4m]} \| \mathbf{w}_i \|_2 B_x + \tilde{b}) + 1 \right) \sqrt{\log n}}{n^{\frac{1}{2}}} \right). \tag{143}$$

□

The gradients allow for obtaining a set of neurons approximating the "ground-truth" network with comparable loss.

**Lemma D.13** (Existence of Good Networks under Empirical Risk). *Suppose* $\frac{n}{\log n} > \Omega\left(\frac{B_x^2}{B_{x2}} + \frac{1}{p} + \frac{B_{x2}}{B_G^2|\ell'(0)|^2}\right)$. *Let* $\lambda^{(1)} = \frac{1}{\eta^{(1)}}$. *For any* $B_\epsilon \in (0, B_b)$, *let* $\sigma_a = \Theta\left(\frac{\tilde{b}}{-|\ell'(0)|\eta^{(1)}B_G B_\epsilon}\right)$ *and* $\delta = 2re^{-\sqrt{\frac{mp}{2}}}$. *Then, with probability at least* $1 - \delta$ *over the initialization and training samples, there exists* $\tilde{\mathbf{a}}_i$'s *such that* $f_{(\tilde{\mathbf{a}},\mathbf{W}^{(1)},\mathbf{b})}(\mathbf{x}) = \sum_{i=1}^{4m} \tilde{\mathbf{a}}_i \sigma\left(\left\langle \mathbf{w}_i^{(1)}, \mathbf{x} \right\rangle - \mathbf{b}_i\right)$ *satisfies*

$$\mathcal{L}_{\mathcal{D}}(f_{(\tilde{\mathbf{a}},\mathbf{W}^{(1)},\mathbf{b})}) \tag{144}$$

$$\leq rB_{a1}\left(\frac{2B_{x1}^2 B_b}{\sqrt{mp}B_G B_\epsilon} + B_{x1}\sqrt{2\gamma + O\left(\frac{\sqrt{B_{x2}\log n}}{B_G|\ell'(0)|n^{\frac{1}{2}}}\right)} + B_\epsilon\right) + \mathrm{OPT}_{d,r,B_F,S_{p,\gamma,B_G}}, \tag{145}$$

*and* $\|\tilde{\mathbf{a}}\|_0 = O\left(r(mp)^{\frac{1}{2}}\right)$, $\|\tilde{\mathbf{a}}\|_2 = O\left(\frac{B_{a2}B_b}{\tilde{b}(mp)^{\frac{1}{4}}}\right)$, $\|\tilde{\mathbf{a}}\|_\infty = O\left(\frac{B_{a1}B_b}{\tilde{b}(mp)^{\frac{1}{2}}}\right)$.

*Proof of Lemma D.13.* Denote $\rho = O\left(\frac{1}{n}\right)$ and $\beta = O\left(\frac{\sqrt{B_{x2}\log n}}{n^{\frac{1}{2}}}\right)$. Note that by symmetric initialization, we have $\ell'(yf_{\Xi^{(0)}}(\mathbf{x})) = |\ell'(0)|$ for any $\mathbf{x} \in \mathcal{X}$, so that, by Lemma D.12, we have $\left\|\widetilde{G}(\mathbf{w}_i^{(0)}, \mathbf{b}_i) - G(\mathbf{w}_i^{(0)}, \mathbf{b}_i)\right\|_2 \leq \frac{\beta}{|\ell'(0)|}$ with probability at least $1 - \rho$. Thus, by union bound, we can see that $S_{p,\gamma,B_G} \subseteq \widetilde{S}_{p-\rho,\gamma+\frac{\beta}{B_G|\ell'(0)|}, B_G-\frac{\beta}{|\ell'(0)|}}$. Consequently, we have $\mathrm{OPT}_{d,r,B_F,\widetilde{S}_{p-\rho,\gamma+\frac{\beta}{B_G|\ell'(0)|}, B_G-\frac{\beta}{|\ell'(0)|}}} \leq \mathrm{OPT}_{d,r,B_F,S_{p,\gamma,B_G}}$. Exactly follow the proof in Lemma D.4 by replacing $S_{p,\gamma,B_G}$ to $\widetilde{S}_{p-\rho,\gamma+\frac{\beta}{B_G|\ell'(0)|}, B_G-\frac{\beta}{|\ell'(0)|}}$. Then, we finish the proof by $\rho \leq \frac{p}{2}, \frac{\beta}{|\ell'(0)|} \leq (1-1/\sqrt{2})B_G$. $\square$

We will use Theorem D.5 to prove that gradient descent learns a good classifier (Theorem D.17). Theorem 3.12 is simply a direct corollary of Theorem D.17. To apply the theorem we first present a few lemmas bounding the change in the network during steps.

**Lemma D.14** (Bound of $\Xi^{(0)}, \Xi^{(1)}$ under Empirical Risk). *Assume the same conditions as in Lemma D.13, and* $d \geq \log m$, *with probability at least* $1 - \delta - \frac{1}{m^2} - O\left(\frac{m}{n}\right)$ *over the initialization and training samples,* $\|\mathbf{a}^{(0)}\|_\infty = O\left(\frac{\tilde{b}\sqrt{\log m}}{|\ell'(0)|\eta^{(1)}B_G B_\epsilon}\right)$, *and for all* $i \in [4m]$, *we have* $\|\mathbf{w}_i^{(0)}\|_2 = O\left(\sigma_w\sqrt{d}\right)$. *Finally,* $\|\mathbf{a}^{(1)}\|_\infty = O\left(\eta^{(1)}|\ell'(0)|(B_{x1}\sigma_w\sqrt{d}+\tilde{b}) + \eta^{(1)}\frac{\sigma_w\sqrt{dB_{x2}\log n}}{n^{\frac{1}{2}}}\right)$, *and for all* $i \in [4m]$, $\|\mathbf{w}_i^{(1)}\|_2 = O\left(\frac{\tilde{b}\sqrt{\log m}B_{x1}}{B_G B_\epsilon} + \frac{\tilde{b}\sqrt{\log m}B_{x2}\log n}{|\ell'(0)|B_G B_\epsilon n^{\frac{1}{2}}}\right)$.

*Proof of Lemma D.14.* The proof exactly follows the proof of Lemma D.6 with Lemma D.12. $\square$

**Lemma D.15** (Bound of $\Xi^{(t)}$ under Empirical Risk). *Assume the same conditions as in Lemma D.14, and let* $\eta = \eta^{(t)}$ *for all* $t \in \{2,3,\ldots,T\}$, $0 < T\eta B_{x1} \leq o(1)$, *and* $0 = \lambda = \lambda^{(t)}$ *for all* $t \in \{2,3,\ldots,T\}$. *With probability at least* $1 - O\left(\frac{Tm}{n}\right)$ *over training samples, for all* $i \in [4m]$, *for all* $t \in \{2,3,\ldots,T\}$, *we have*

$$|\mathbf{a}_i^{(t)}| \leq O\left(|\mathbf{a}_i^{(1)}| + \|\mathbf{w}_i^{(1)}\|_2 + \frac{\tilde{b}}{\left(B_{x1} + \frac{\sqrt{B_{x2}\log n}}{n^{\frac{1}{2}}}\right)} + \eta\tilde{b}\right) \tag{146}$$

$$\|\mathbf{w}_i^{(t)} - \mathbf{w}_i^{(1)}\|_2 \leq O\left(t\eta\left(B_{x1} + \frac{\sqrt{B_{x2}\log n}}{n^{\frac{1}{2}}}\right)|\mathbf{a}_i^{(1)}| + t\eta^2\left(B_{x1} + \frac{\sqrt{B_{x2}\log n}}{n^{\frac{1}{2}}}\right)^2\|\mathbf{w}_i^{(1)}\|_2 \right.$$

$$\left. + t\eta^2\left(B_{x1} + \frac{\sqrt{B_{x2}\log n}}{n^{\frac{1}{2}}}\right)\tilde{b}\right). \tag{147}$$

*Proof of Lemma D.15.* The proof exactly follows the proof of Lemma D.7 with Lemma D.12. Note that, we have

$$|\mathbf{a}_i^{(t)}| \leq \left|\mathbf{a}_i^{(t-1)}\right| + \eta(B_{x1}\|\mathbf{w}_i^{(t-1)}\|_2 + \tilde{b}) + \eta\frac{\|\mathbf{w}_i^{(t-1)}\|_2\sqrt{B_{x2}\log n}}{n^{\frac{1}{2}}} \tag{148}$$

$$\leq \left|\mathbf{a}_i^{(t-1)}\right| + \eta\left(B_{x1} + \frac{\sqrt{B_{x2}\log n}}{n^{\frac{1}{2}}}\right)\|\mathbf{w}_i^{(t-1)} - \mathbf{w}_i^{(1)}\|_2 + \eta Z_i, \tag{149}$$

where we denote $Z_i = \left(B_{x1} + \frac{\sqrt{B_{x2}\log n}}{n^{\frac{1}{2}}}\right)\|\mathbf{w}_i^{(1)}\|_2 + \tilde{b}$. Similarly, we have

$$\|\mathbf{w}_i^{(t)} - \mathbf{w}_i^{(1)}\|_2 \leq \|\mathbf{w}_i^{(t-1)} - \mathbf{w}_i^{(1)}\|_2 + \eta\left(B_{x1} + \frac{\sqrt{B_{x2}\log n}}{n^{\frac{1}{2}}}\right)|\mathbf{a}_i^{(t-1)}|. \tag{150}$$

We finish the proof by following the same arguments in the proof of Lemma D.7 and union bound. $\square$

**Lemma D.16** (Bound of Loss Gap and Gradient under Empirical Risk). *Assume the same conditions as in Lemma D.15. With probability at least $1 - O\left(\frac{T}{n}\right)$, for all $t \in [T]$, we have*

$$\left|\widetilde{\mathcal{L}}_{\mathcal{Z}^{(t)}}\left(f_{(\tilde{\mathbf{a}},\mathbf{W}^{(t)},\mathbf{b})}\right) - \mathcal{L}_{\mathcal{D}}(f_{(\tilde{\mathbf{a}},\mathbf{W}^{(1)},\mathbf{b})})\right| \tag{151}$$

$$\leq O\left(\frac{\left(\|\tilde{\mathbf{a}}\|_0\|\tilde{\mathbf{a}}\|_\infty(\max_{i\in[4m]}\|\mathbf{w}_i^{(t)}\|_2 B_x + \tilde{b}) + 1\right)\sqrt{\log n}}{n^{\frac{1}{2}}}\right) \tag{152}$$

$$+ B_{x1}\|\tilde{\mathbf{a}}\|_2\sqrt{\|\tilde{\mathbf{a}}\|_0}\max_{i\in[4m]}\|\mathbf{w}_i^{(t)} - \mathbf{w}_i^{(1)}\|_2. \tag{153}$$

*With probability at least $1 - O\left(\frac{T}{n}\right)$, for all $t \in [T]$, $i \in [4m]$ we have*

$$\left|\frac{\partial\widetilde{\mathcal{L}}_{\mathcal{Z}^{(t)}}(f_{\Xi^{(t)}})}{\partial\mathbf{a}_i^{(t)}}\right| \leq B_{x1}(\|\mathbf{w}_i^{(t)} - \mathbf{w}_i^{(1)}\|_2 + \|\mathbf{w}_i^{(1)}\|_2) + \tilde{b} + O\left(\frac{\|\mathbf{w}_i^{(t)}\|_2\sqrt{B_{x2}\log n}}{n^{\frac{1}{2}}}\right). \tag{154}$$

*Proof of Lemma D.16.* By Lemma D.8 and Lemma D.12, with probability at least $1 - O\left(\frac{T}{n}\right)$, for all $t \in [T]$, we have

$$\left|\widetilde{\mathcal{L}}_{\mathcal{Z}^{(t)}}\left(f_{(\tilde{\mathbf{a}},\mathbf{W}^{(t)},\mathbf{b})}\right) - \mathcal{L}_{\mathcal{D}}(f_{(\tilde{\mathbf{a}},\mathbf{W}^{(1)},\mathbf{b})})\right| \tag{155}$$

$$\leq \left|\widetilde{\mathcal{L}}_{\mathcal{Z}^{(t)}}\left(f_{(\tilde{\mathbf{a}},\mathbf{W}^{(t)},\mathbf{b})}\right) - \mathcal{L}_{\mathcal{D}}(f_{(\tilde{\mathbf{a}},\mathbf{W}^{(t)},\mathbf{b})})\right| + \left|\mathcal{L}_{\mathcal{D}}(f_{(\tilde{\mathbf{a}},\mathbf{W}^{(t)},\mathbf{b})}) - \mathcal{L}_{\mathcal{D}}(f_{(\tilde{\mathbf{a}},\mathbf{W}^{(1)},\mathbf{b})})\right| \tag{156}$$

$$\leq O\left(\frac{\left(\|\tilde{\mathbf{a}}\|_0\|\tilde{\mathbf{a}}\|_\infty(\max_{i\in[4m]}\|\mathbf{w}_i^{(t)}\|_2 B_x + \tilde{b}) + 1\right)\sqrt{\log n}}{n^{\frac{1}{2}}}\right) \tag{157}$$

$$+ B_{x1}\|\tilde{\mathbf{a}}\|_2\sqrt{\|\tilde{\mathbf{a}}\|_0}\max_{i\in[4m]}\|\mathbf{w}_i^{(t)} - \mathbf{w}_i^{(1)}\|_2. \tag{158}$$

By Lemma D.8 and Lemma D.12, with probability at least $1 - O\left(\frac{T}{n}\right)$, for all $t \in [T]$, $i \in [4m]$ we have

$$\left|\frac{\partial\widetilde{\mathcal{L}}_{\mathcal{Z}^{(t)}}(f_{\Xi^{(t)}})}{\partial\mathbf{a}_i^{(t)}}\right| \leq B_{x1}(\|\mathbf{w}_i^{(t)} - \mathbf{w}_i^{(1)}\|_2 + \|\mathbf{w}_i^{(1)}\|_2) + \tilde{b} + O\left(\frac{\|\mathbf{w}_i^{(t)}\|_2\sqrt{B_{x2}\log n}}{n^{\frac{1}{2}}}\right). \tag{159}$$

$\square$

We are now ready to prove the main theorem.

**Theorem D.17** (Online Convex Optimization under Empirical Risk. Full Statement of Theorem 3.12). *Consider training by Algorithm 1, and any $\delta \in (0,1)$. Assume $d \geq \log m$. Set*

$$\sigma_w > 0, \quad \tilde{b} > 0, \quad \eta^{(t)} = \eta, \quad \lambda^{(t)} = 0 \text{ for all } t \in \{2,3,\ldots,T\},$$

$$\eta^{(1)} = \Theta\left(\frac{\min\{O(\eta), O(\eta\tilde{b})\}}{-\ell'(0)(B_{x1}\sigma_w\sqrt{d} + \tilde{b})}\right), \quad \lambda^{(1)} = \frac{1}{\eta^{(1)}}, \quad \sigma_a = \Theta\left(\frac{\tilde{b}(mp)^{\frac{1}{4}}}{-\ell'(0)\eta^{(1)}B_{x1}\sqrt{B_G B_b}}\right).$$

*Let* $0 < T\eta B_{x1} \leq o(1)$, $m = \Omega\left(\frac{1}{\sqrt{\delta}} + \frac{1}{p}\left(\log\left(\frac{r}{\delta}\right)\right)^2\right)$ *and* $\frac{n}{\log n} > \Omega\left(\frac{B_x^2}{B_{x2}} + \frac{1}{p} + \left(\frac{1}{B_G^2} + \frac{1}{B_{x1}^2}\right)\frac{B_{x2}}{|\ell'(0)|^2} + \frac{Tm}{\delta}\right)$. *With probability at least* $1 - \delta$ *over the initialization and training samples, there exists* $t \in [T]$ *such that*

$$\mathcal{L}_{\mathcal{D}}\left(f_{\Xi^{(t)}}\right) \tag{160}$$

$$\leq \mathrm{OPT}_{d,r,B_F,S_{p,\gamma,B_G}} + rB_{a1}\left(\frac{2\sqrt{2}B_{x1}}{(mp)^{\frac{1}{4}}}\sqrt{\frac{B_b}{B_G}} + B_{x1}\sqrt{2\gamma + O\left(\frac{\sqrt{B_{x2}\log n}}{B_G|\ell'(0)|n^{\frac{1}{2}}}\right)}\right) \tag{161}$$

$$+ \eta\left(\sqrt{r}B_{a2}B_bT\eta B_{x1}^2 + m\tilde{b}\right)O\left(\frac{\sqrt{\log m}B_{x1}(mp)^{\frac{1}{4}}}{\sqrt{B_bB_G}} + 1\right) + O\left(\frac{B_{a2}^2B_b^2}{\eta T\tilde{b}^2(mp)^{\frac{1}{2}}}\right) \tag{162}$$

$$+ \frac{\sqrt{\log n}}{n^{\frac{1}{2}}}O\left(\left(\frac{rB_{a1}B_b}{\tilde{b}} + m\left(\frac{\tilde{b}\sqrt{\log m}(mp)^{\frac{1}{4}}}{\sqrt{B_bB_G}} + \frac{\tilde{b}}{B_{x1}}\right)\right)\right) \tag{163}$$

$$\cdot\left(\left(\frac{\tilde{b}\sqrt{\log m}(mp)^{\frac{1}{4}}}{\sqrt{B_bB_G}} + T\eta^2 B_{x1}\tilde{b}\right)B_x + \tilde{b}\right) + 2\right) \tag{164}$$

$$+ \frac{\sqrt{\log n}}{n^{\frac{1}{2}}}O\left(m\eta\left(\frac{\tilde{b}\sqrt{\log m}(mp)^{\frac{1}{4}}}{\sqrt{B_bB_G}} + T\eta^2 B_{x1}\tilde{b}\right)\sqrt{B_{x2}}\right). \tag{165}$$

*Furthermore, for any* $\epsilon \in (0, 1)$, *set*

$$\tilde{b} = \Theta\left(\frac{B_G^{\frac{1}{4}}B_{a2}B_b^{\frac{3}{4}}}{\sqrt{rB_{a1}}}\right), \quad m = \Omega\left(\frac{1}{p\epsilon^4}\left(rB_{a1}B_{x1}\sqrt{\frac{B_b}{B_G}}\right)^4 + \frac{1}{\sqrt{\delta}} + \frac{1}{p}\left(\log\left(\frac{r}{\delta}\right)\right)^2\right),$$

$$\eta = \Theta\left(\frac{\epsilon}{\left(\frac{\sqrt{r}B_{a2}B_bB_{x1}}{(mp)^{\frac{1}{4}}} + m\tilde{b}\right)\left(\frac{\sqrt{\log m}B_{x1}(mp)^{\frac{1}{4}}}{\sqrt{B_bB_G}} + 1\right)}\right), \quad T = \Theta\left(\frac{1}{\eta B_{x1}(mp)^{\frac{1}{4}}}\right),$$

$$\frac{n}{\log n} = \Omega\left(\frac{m^3pB_x^2B_{a2}^4B_b(\log m)^2}{\epsilon^2 r^2 B_{a1}^2 B_G} + \frac{(mp)^{\frac{1}{2}}B_{x2}\log m}{B_bB_G} + \frac{B_x^2}{B_{x2}} + \frac{1}{p} + \left(\frac{1}{B_G^2} + \frac{1}{B_{x1}^2}\right)\frac{B_{x2}}{|\ell'(0)|^2} + \frac{Tm}{\delta}\right),$$

*we have there exists* $t \in [T]$ *with*

$$\Pr[\mathrm{sign}(f_{\Xi^{(t)}})(\mathbf{x}) \neq y] \leq \mathcal{L}_{\mathcal{D}}\left(f_{\Xi^{(t)}}\right) \tag{166}$$

$$\leq \mathrm{OPT}_{d,r,B_F,S_{p,\gamma,B_G}} + rB_{a1}B_{x1}\sqrt{2\gamma + O\left(\frac{\sqrt{B_{x2}\log n}}{B_G|\ell'(0)|n^{\frac{1}{2}}}\right)} + \epsilon. \tag{167}$$

*Proof of Theorem D.17.* We follow the proof in Theorem D.9. By $m = \Omega\left(\frac{1}{\sqrt{\delta}} + \frac{1}{p}\left(\log\left(\frac{r}{\delta}\right)\right)^2\right)$ and $\frac{n}{\log n} > \Omega\left(\frac{B_x^2}{B_{x2}} + \frac{1}{p} + \frac{B_{x2}}{B_G^2|\ell'(0)|^2} + \frac{Tm}{\delta}\right)$, we have $2re^{-\sqrt{\frac{mp}{2}}} + \frac{1}{m^2} + O\left(\frac{Tm}{n}\right) \leq \delta$. For any $B_\epsilon \in (0, B_b)$, when $\sigma_a = \Theta\left(\frac{\tilde{b}}{-\ell'(0)\eta^{(1)}B_GB_\epsilon}\right)$, by Theorem D.5, Lemma D.12, Lemma D.13,

Lemma D.16, with probability at least $1 - \delta$ over the initialization and training samples, we have

$$\frac{1}{T}\sum_{t=1}^{T}\mathcal{L}_{\mathcal{D}}\left(f_{\Xi^{(t)}}\right) \tag{168}$$

$$\leq \frac{1}{T}\sum_{t=1}^{T}|\mathcal{L}_{\mathcal{D}}\left(f_{\Xi^{(t)}}\right) - \widetilde{\mathcal{L}}_{\mathcal{Z}^{(t)}}\left(f_{\Xi^{(t)}}\right)| + \frac{1}{T}\sum_{t=1}^{T}\widetilde{\mathcal{L}}_{\mathcal{Z}^{(t)}}\left(f_{\Xi^{(t)}}\right) \tag{169}$$

$$\leq \frac{1}{T}\sum_{t=1}^{T}|\mathcal{L}_{\mathcal{D}}\left(f_{\Xi^{(t)}}\right) - \widetilde{\mathcal{L}}_{\mathcal{Z}^{(t)}}\left(f_{\Xi^{(t)}}\right)| + \frac{1}{T}\sum_{t=1}^{T}\left|\widetilde{\mathcal{L}}_{\mathcal{Z}^{(t)}}\left(f_{(\tilde{\mathbf{a}},\mathbf{W}^{(t)},\mathbf{b})}\right) - \mathcal{L}_{\mathcal{D}}(f_{(\tilde{\mathbf{a}},\mathbf{W}^{(1)},\mathbf{b})})\right| \tag{170}$$

$$+ \mathcal{L}_{\mathcal{D}}(f_{(\tilde{\mathbf{a}},\mathbf{W}^{(1)},\mathbf{b})}) + \frac{\|\tilde{\mathbf{a}}\|_2^2}{2\eta T} + (2\|\mathbf{a}^{(1)}\|_2\sqrt{m} + 4\eta m)\max_{t\in[T],i\in[4m]}\left|\frac{\partial\widetilde{\mathcal{L}}_{\mathcal{Z}^{(t)}}\left(f_{\Xi^{(t)}}\right)}{\partial\mathbf{a}_i^{(t)}}\right| \tag{171}$$

$$\leq B_{x1}\|\tilde{\mathbf{a}}\|_2\sqrt{\|\tilde{\mathbf{a}}\|_0}\max_{i\in[4m]}\|\mathbf{w}_i^{(T)} - \mathbf{w}_i^{(1)}\|_2 \tag{172}$$

$$+ O\left(\frac{\left(\|\tilde{\mathbf{a}}\|_0\|\tilde{\mathbf{a}}\|_\infty + m\|\mathbf{a}^{(T)}\|_\infty\right)(\max_{i\in[4m]}\|\mathbf{w}_i^{(T)}\|_2 B_x + \tilde{b}) + 2)\sqrt{\log n}}{n^{\frac{1}{2}}}\right) \tag{173}$$

$$+ \mathrm{OPT}_{d,r,B_F,S_{p,\gamma},B_G} + rB_{a1}\left(\frac{2B_{x1}^2 B_b}{\sqrt{mp}B_G B_\epsilon} + B_{x1}\sqrt{2\gamma + O\left(\frac{\sqrt{B_{x2}\log n}}{B_G|\ell'(0)|n^{\frac{1}{2}}}\right)} + B_\epsilon\right) \tag{174}$$

$$+ \frac{\|\tilde{\mathbf{a}}\|_2^2}{2\eta T} + 4mB_{x1}(\|\mathbf{a}^{(1)}\|_\infty + \eta) \tag{175}$$

$$\cdot\left(\max_{i\in[4m]}\|\mathbf{w}_i^{(T)} - \mathbf{w}_i^{(1)}\|_2 + \max_{i\in[4m]}\|\mathbf{w}_i^{(1)}\|_2 + \frac{\tilde{b}}{B_{x1}} + O\left(\frac{\max_{i\in[4m]}\|\mathbf{w}_i^{(T)}\|_2\sqrt{B_{x2}\log n}}{B_{x1}n^{\frac{1}{2}}}\right)\right).$$

Set $B_\epsilon = \frac{B_{x1}}{(mp)^{\frac{1}{4}}}\sqrt{\frac{2B_b}{B_G}}$, we have $\sigma_a = \Theta\left(\frac{\tilde{b}(mp)^{\frac{1}{4}}}{-\ell'(0)\eta^{(1)}B_{x1}\sqrt{B_G B_b}}\right)$ which satisfy the requirements. By Lemma D.13, Lemma D.14, Lemma D.15, $\frac{n}{\log n} > \Omega\left(\frac{B_{x2}}{B_{x1}^2|\ell'(0)|^2}\right)$, when $\eta^{(1)} = \Theta\left(\frac{\min\{O(\eta),O(\eta\tilde{b})\}}{-\ell'(0)(B_{x1}\sigma_w\sqrt{d}+\tilde{b})}\right)$, we have

$$\|\tilde{\mathbf{a}}\|_0 = O\left(r(mp)^{\frac{1}{2}}\right), \quad \|\tilde{\mathbf{a}}\|_2 = O\left(\frac{B_{a2}B_b}{\tilde{b}(mp)^{\frac{1}{4}}}\right), \quad \|\tilde{\mathbf{a}}\|_\infty = O\left(\frac{B_{a1}B_b}{\tilde{b}(mp)^{\frac{1}{2}}}\right) \tag{176}$$

$$\|\mathbf{a}^{(1)}\|_\infty = O\left(\eta^{(1)}|\ell'(0)|(B_{x1}\sigma_w\sqrt{d}+\tilde{b}) + \eta^{(1)}\frac{\sigma_w\sqrt{dB_{x2}\log n}}{n^{\frac{1}{2}}}\right) \tag{177}$$

$$= \min\{O(\eta),O(\eta\tilde{b})\} \tag{178}$$

$$\|\mathbf{a}^{(T)}\|_\infty \leq O\left(\|\mathbf{a}^{(1)}\|_\infty + \max_{i\in[4m]}\|\mathbf{w}_i^{(1)}\|_2 + \frac{\tilde{b}}{\left(B_{x1} + \frac{\sqrt{B_{x2}\log n}}{n^{\frac{1}{2}}}\right)} + \eta\tilde{b}\right) \tag{179}$$

$$\leq O\left(\max_{i\in[4m]}\|\mathbf{w}_i^{(1)}\|_2 + \frac{\tilde{b}}{B_{x1}}\right) \tag{180}$$

$$\max_{i \in [4m]} \|\mathbf{w}_i^{(1)}\|_2 = O\left( \frac{\tilde{b}\sqrt{\log m}B_{x1}}{B_G B_\epsilon} + \frac{\tilde{b}\sqrt{\log m}\sqrt{B_{x2}\log n}}{|\ell'(0)|B_G B_\epsilon n^{\frac{1}{2}}} \right) \tag{181}$$

$$= O\left( \frac{\tilde{b}\sqrt{\log m}B_{x1}}{B_G B_\epsilon} \right) \tag{182}$$

$$= O\left( \frac{\tilde{b}\sqrt{\log m}(mp)^{\frac{1}{4}}}{\sqrt{B_b B_G}} \right) \tag{183}$$

$$\max_{i \in [4m]} \|\mathbf{w}_i^{(T)} - \mathbf{w}_i^{(1)}\|_2 = O\left( T\eta \left( B_{x1} + \frac{\sqrt{B_{x2}\log n}}{n^{\frac{1}{2}}} \right) |\mathbf{a}_i^{(1)}| \right) \tag{184}$$

$$+ T\eta^2 \left( B_{x1} + \frac{\sqrt{B_{x2}\log n}}{n^{\frac{1}{2}}} \right)^2 \|\mathbf{w}_i^{(1)}\|_2 \tag{185}$$

$$+ T\eta^2 \left( B_{x1} + \frac{\sqrt{B_{x2}\log n}}{n^{\frac{1}{2}}} \right) \tilde{b} \right) \tag{186}$$

$$= O\left( T\eta^2 B_{x1}^2 \left( \max_{i \in [4m]} \|\mathbf{w}_i^{(1)}\|_2 + \frac{\tilde{b}}{B_{x1}} \right) \right). \tag{187}$$

Then, following the proof in Theorem D.9, we have

$$\frac{1}{T}\sum_{t=1}^{T} \mathcal{L}_{\mathcal{D}}\left(f_{\Xi^{(t)}}\right) \tag{188}$$

$$\leq \mathrm{OPT}_{d,r,B_F,S_{p,\gamma,B_G}} + rB_{a1}\left( \frac{2\sqrt{2}B_{x1}}{(mp)^{\frac{1}{4}}}\sqrt{\frac{B_b}{B_G}} + B_{x1}\sqrt{2\gamma + O\left( \frac{\sqrt{B_{x2}\log n}}{B_G|\ell'(0)|n^{\frac{1}{2}}} \right)} \right) \tag{189}$$

$$+ \eta\left( \sqrt{r}B_{a2}B_b T\eta B_{x1}^2 + m\tilde{b} \right)O\left( \frac{\sqrt{\log m}B_{x1}(mp)^{\frac{1}{4}}}{\sqrt{B_b B_G}} + 1 \right) + O\left( \frac{B_{a2}^2 B_b^2}{\eta T \tilde{b}^2 (mp)^{\frac{1}{2}}} \right) \tag{190}$$

$$+ O\left( \frac{\left( (\|\tilde{\mathbf{a}}\|_0 \|\tilde{\mathbf{a}}\|_\infty + m\|\mathbf{a}^{(T)}\|_\infty)(\max_{i \in [4m]}\|\mathbf{w}_i^{(T)}\|_2 B_x + \tilde{b}) + 2 \right)\sqrt{\log n}}{n^{\frac{1}{2}}} \right) \tag{191}$$

$$+ O\left( \frac{m\eta \max_{i \in [4m]}\|\mathbf{w}_i^{(T)}\|_2 \sqrt{B_{x2}\log n}}{n^{\frac{1}{2}}} \right) \tag{192}$$

$$\leq \mathrm{OPT}_{d,r,B_F,S_{p,\gamma,B_G}} + rB_{a1}\left( \frac{2\sqrt{2}B_{x1}}{(mp)^{\frac{1}{4}}}\sqrt{\frac{B_b}{B_G}} + B_{x1}\sqrt{2\gamma + O\left( \frac{\sqrt{B_{x2}\log n}}{B_G|\ell'(0)|n^{\frac{1}{2}}} \right)} \right) \tag{193}$$

$$+ \eta\left( \sqrt{r}B_{a2}B_b T\eta B_{x1}^2 + m\tilde{b} \right)O\left( \frac{\sqrt{\log m}B_{x1}(mp)^{\frac{1}{4}}}{\sqrt{B_b B_G}} + 1 \right) + O\left( \frac{B_{a2}^2 B_b^2}{\eta T \tilde{b}^2 (mp)^{\frac{1}{2}}} \right) \tag{194}$$

$$+ \frac{\sqrt{\log n}}{n^{\frac{1}{2}}}O\left( \left( \frac{rB_{a1}B_b}{\tilde{b}} + m\left( \frac{\tilde{b}\sqrt{\log m}(mp)^{\frac{1}{4}}}{\sqrt{B_b B_G}} + \frac{\tilde{b}}{B_{x1}} \right) \right) \right) \tag{195}$$

$$\cdot \left( \left( \frac{\tilde{b}\sqrt{\log m}(mp)^{\frac{1}{4}}}{\sqrt{B_b B_G}} + T\eta^2 B_{x1}\tilde{b} \right)B_x + \tilde{b} \right) + 2 \right) \tag{196}$$

$$+ \frac{\sqrt{\log n}}{n^{\frac{1}{2}}}O\left( m\eta\left( \frac{\tilde{b}\sqrt{\log m}(mp)^{\frac{1}{4}}}{\sqrt{B_b B_G}} + T\eta^2 B_{x1}\tilde{b} \right)\sqrt{B_{x2}} \right). \tag{197}$$

Furthermore, for any $\epsilon \in (0,1)$, set

$$\tilde{b} = \Theta\left(\frac{B_G^{\frac{1}{4}}B_{a2}B_b^{\frac{3}{4}}}{\sqrt{r}B_{a1}}\right), \quad m = \Omega\left(\frac{1}{p\epsilon^4}\left(rB_{a1}B_{x1}\sqrt{\frac{B_b}{B_G}}\right)^4 + \frac{1}{\sqrt{\delta}} + \frac{1}{p}\left(\log\left(\frac{r}{\delta}\right)\right)^2\right),$$

$$\eta = \Theta\left(\frac{\epsilon}{\left(\frac{\sqrt{r}B_{a2}B_bB_{x1}}{(mp)^{\frac{1}{4}}} + m\tilde{b}\right)\left(\frac{\sqrt{\log m}B_{x1}(mp)^{\frac{1}{4}}}{\sqrt{B_bB_G}} + 1\right)}\right), \quad T = \Theta\left(\frac{1}{\eta B_{x1}(mp)^{\frac{1}{4}}}\right),$$

$$\frac{n}{\log n} = \Omega\left(\frac{m^3pB_x^2B_{a2}^4B_b(\log m)^2}{\epsilon^2r^2B_{a1}^2B_G} + \frac{(mp)^{\frac{1}{2}}B_{x2}\log m}{B_bB_G} + \frac{B_x^2}{B_{x2}} + \frac{1}{p} + \left(\frac{1}{B_G^2} + \frac{1}{B_{x1}^2}\right)\frac{B_{x2}}{|\ell'(0)|^2} + \frac{Tm}{\delta}\right),$$

and note that $B_G \leq B_{x_1} \leq B_x$ and $\sqrt{B_{x_2}} \leq B_x$ naturally, we have

$$\frac{1}{T}\sum_{t=1}^{T}\mathcal{L}_\mathcal{D}\left(f_{\Xi^{(t)}}\right) \tag{198}$$

$$\leq \text{OPT}_{d,r,B_F,S_{p,\gamma,B_G}} + rB_{a1}\left(\frac{2\sqrt{2}B_{x1}}{(mp)^{\frac{1}{4}}}\sqrt{\frac{B_b}{B_G}} + B_{x1}\sqrt{2\gamma + O\left(\frac{\sqrt{B_{x2}\log n}}{B_G|\ell'(0)|n^{\frac{1}{2}}}\right)}\right) \tag{199}$$

$$+ \frac{\epsilon}{2} + O\left(\frac{B_{x1}B_{a2}^2B_b^2}{\tilde{b}^2(mp)^{\frac{1}{4}}}\right) + \frac{\sqrt{\log n}}{n^{\frac{1}{2}}}O\left(\frac{mB_xB_{a2}^2\sqrt{B_b}(mp)^{\frac{1}{2}}\log m}{rB_{a1}\sqrt{B_G}}\right) \tag{200}$$

$$+ \frac{\sqrt{\log n}}{n^{\frac{1}{2}}}O\left(\frac{\epsilon\sqrt{B_{x2}\log m}(mp)^{\frac{1}{4}}}{\sqrt{B_bB_G}}\right) \tag{201}$$

$$\leq \text{OPT}_{d,r,B_F,S_{p,\gamma,B_G}} + rB_{a1}B_{x1}\sqrt{2\gamma + O\left(\frac{\sqrt{B_{x2}\log n}}{B_G|\ell'(0)|n^{\frac{1}{2}}}\right)} + \epsilon. \tag{202}$$

We finish the proof as the 0-1 classification error is bounded by the loss function, e.g., $\mathbb{I}[\text{sign}(f(\mathbf{x})) \neq y] \leq \frac{\ell(yf(\mathbf{x}))}{\ell(0)}$, where $\ell(0) = 1$.

$\square$

# E   Applications in Special Cases

We present the case study of linear data in Appendix E.1, mixtures of Gaussians in Appendix E.2 and Appendix E.3, parity functions in Appendix E.4, Appendix E.5 and Appendix E.6, and multiple-index models in Appendix E.7.

In special case applications, we consider binary classification with hinge loss, e.g., $\ell(z) = \max\{1 - z, 0\}$. Let $\mathcal{X} = \mathbb{R}^d$ be the input space, and $\mathcal{Y} = \{\pm 1\}$ be the label space.

**Remark E.1** (Hinge Loss and Logistic Loss). *Both hinge loss and logistic loss can be used in special cases and general cases. For convenience, we use hinge loss in special cases, where we can directly get the ground-truth NN close form of the optimal solution which has zero loss. For logistic loss, there is no zero-loss solution. We can still show that the OPT value has an exponentially small upper bound at the cost of more computation.*

## E.1   Linear Data

**Data Distributions.**   Suppose two labels are equiprobable, i.e., $\mathbb{E}[y = -1] = \mathbb{E}[y = +1] = \frac{1}{2}$. The input data are linearly separable and there is a ground truth direction $\mathbf{w}^*$, where $\|\mathbf{w}^*\|_2 = 1$, such that $y\langle\mathbf{w}^*, \mathbf{x}\rangle > 0$. We also assume $\mathbb{E}[yP_{\mathbf{w}^*\perp}\mathbf{x}] = 0$, where $P_{\mathbf{w}^*\perp}$ is the projection operator on the complementary space of the ground truth, i.e., the components of input data being orthogonal with the ground truth are independent of the label $y$. We define the input data signal level as $\rho := \mathbb{E}[y\langle\mathbf{w}^*, \mathbf{x}\rangle] > 0$ and the margin as $\beta := \min_{(\mathbf{x},y)} y\langle\mathbf{w}^*, \mathbf{x}\rangle > 0$.

We call this data distribution $\mathcal{D}_{linear}$.

**Lemma E.2** (Linear Data: Gradient Feature Set). *Let $\tilde{b} = d^\tau B_{x1} \sigma_w$, where $\tau$ is any number large enough to satisfy $d^{\tau/2 - \frac{1}{4}} > \Omega\left(\frac{\sqrt{B_{x2}}}{\rho}\right)$. For $\mathcal{D}_{linear}$ setting, we have $(\mathbf{w}^*, -1) \in S_{p,\gamma,B_G}$ where*

$$p = \frac{1}{2}, \quad \gamma = \Theta\left(\frac{\sqrt{B_{x2}}}{\rho d^{\tau/2 - \frac{1}{4}}}\right), \quad B_G = \rho - \Theta\left(\frac{\sqrt{B_{x2}}}{d^{\tau/2 - \frac{1}{4}}}\right). \tag{203}$$

*Proof of Lemma E.2.* By data distribution, we have

$$\mathbb{E}_{(\mathbf{x},y)}[y\mathbf{x}] = \rho\mathbf{w}^*. \tag{204}$$

Define $S_{Sure} : \{i \in [m] : \|\mathbf{w}_i^{(0)}\|_2 \leq 2\sqrt{d}\sigma_w\}$. For all $i \in [m]$, we have

$$\Pr[i \in S_{Sure}] = \Pr[\|\mathbf{w}_i^{(0)}\|_2 \leq 2\sqrt{d}\sigma_w] \geq \frac{1}{2}. \tag{205}$$

For all $i \in S_{Sure}$, by Markov's inequality and considering neuron $i + m$, we have

$$\Pr_{\mathbf{x}}\left[\left\langle \mathbf{w}_{i+m}^{(0)}, \mathbf{x}\right\rangle - \mathbf{b}_{i+m} < 0\right] = \Pr_{\mathbf{x}}\left[\left\langle \mathbf{w}_i^{(0)}, \mathbf{x}\right\rangle + \mathbf{b}_i < 0\right] \tag{206}$$

$$\leq \Pr_{\mathbf{x}}\left[\|\mathbf{w}_i^{(0)}\|_2 \|\mathbf{x}\|_2 \geq \mathbf{b}_i\right] \tag{207}$$

$$\leq \Pr_{\mathbf{x}}\left[\|\mathbf{x}\|_2 \geq \frac{d^{\tau - \frac{1}{2}} B_{x1}}{2}\right] \tag{208}$$

$$\leq \Theta\left(\frac{1}{d^{\tau - \frac{1}{2}}}\right). \tag{209}$$

For all $i \in S_{Sure}$, by Hölder's inequality, we have

$$\left\|\mathbb{E}_{(\mathbf{x},y)}\left[y\left(1 - \sigma'\left[\left\langle \mathbf{w}_{i+m}^{(0)}, \mathbf{x}\right\rangle - \mathbf{b}_{i+m}\right]\right)\mathbf{x}\right]\right\|_2 \tag{210}$$

$$= \left\|\mathbb{E}_{(\mathbf{x},y)}\left[y\left(1 - \sigma'\left[\left\langle \mathbf{w}_i^{(0)}, \mathbf{x}\right\rangle + \mathbf{b}_i\right]\right)\mathbf{x}\right]\right\|_2 \tag{211}$$

$$\leq \sqrt{\mathbb{E}[\|\mathbf{x}\|_2^2]\mathbb{E}\left[\left(1 - \sigma'\left[\left\langle \mathbf{w}_i^{(0)}, \mathbf{x}\right\rangle + \mathbf{b}_i\right]\right)^2\right]} \tag{212}$$

$$\leq \Theta\left(\frac{\sqrt{B_{x2}}}{d^{\tau/2 - \frac{1}{4}}}\right). \tag{213}$$

We have

$$1 - \frac{\left|\left\langle G(\mathbf{w}_{i+m}^{(0)}, \mathbf{b}_{i+m}), \mathbf{w}^*\right\rangle\right|}{\|G(\mathbf{w}_{i+m}^{(0)}, \mathbf{b}_{i+m})\|_2} = 1 - \frac{\left|\left\langle G(\mathbf{w}_i^{(0)}, -\mathbf{b}_i), \mathbf{w}^*\right\rangle\right|}{\|G(\mathbf{w}_i^{(0)}, -\mathbf{b}_i)\|_2} \tag{214}$$

$$\leq 1 - \frac{\rho - \Theta\left(\frac{\sqrt{B_{x2}}}{d^{\tau/2 - \frac{1}{4}}}\right)}{\rho + \Theta\left(\frac{\sqrt{B_{x2}}}{d^{\tau/2 - \frac{1}{4}}}\right)} \tag{215}$$

$$= \Theta\left(\frac{\sqrt{B_{x2}}}{\rho d^{\tau/2 - \frac{1}{4}}}\right) = \gamma. \tag{216}$$

We finish the proof by $\frac{\mathbf{b}_{i+m}}{|\mathbf{b}_{i+m}|} = -1$. $\qquad\square$

**Lemma E.3** (Linear Data: Existence of Good Networks). *Assume the same conditions as in Lemma E.2. Define*

$$f^*(\mathbf{x}) = \frac{1}{\beta}\sigma(\langle \mathbf{w}^*, \mathbf{x}\rangle) - \frac{1}{\beta}\sigma(\langle -\mathbf{w}^*, \mathbf{x}\rangle). \tag{217}$$

*For $\mathcal{D}_{linear}$ setting, we have $f^* \in \mathcal{F}_{d,r,B_F,S_{p,\gamma,B_G}}$, where $r = 2, B_F = (B_{a1}, B_{a2}, B_b) = \left(\frac{1}{\beta}, \frac{\sqrt{2}}{\beta}, \frac{1}{B_{x1}^2}\right), p = \frac{1}{2}, \gamma = \Theta\left(\frac{\sqrt{B_{x2}}}{\rho d^{\tau/2 - \frac{1}{4}}}\right), B_G = \rho - \Theta\left(\frac{\sqrt{B_{x2}}}{d^{\tau/2 - \frac{1}{4}}}\right).$ We also have $\mathrm{OPT}_{d,r,B_F,S_{p,\gamma,B_G}} = 0.$*

*Proof of Lemma E.3.* By Lemma E.2 and Lemma F.3, we have $f^* \in \mathcal{F}_{d,r,B_F,S_{p,\gamma},B_G}$. We also have

$$\text{OPT}_{d,r,B_F,S_{p,\gamma},B_G} \leq \mathcal{L}_{\mathcal{D}_{linear}}(f^*) \tag{218}$$

$$= \mathbb{E}_{(\mathbf{x},y)\sim\mathcal{D}_{linear}} \mathcal{L}_{(\mathbf{x},y)}(f^*) \tag{219}$$

$$= 0. \tag{220}$$

$\square$

**Theorem E.4** (Linear Data: Main Result). *For $\mathcal{D}_{linear}$ setting, for any $\delta \in (0,1)$ and for any $\epsilon \in (0,1)$ when*

$$m = \text{poly}\left(\frac{1}{\delta}, \frac{1}{\epsilon}, \frac{1}{\beta}, \frac{1}{\rho}\right) \leq e^d, \quad T = \text{poly}(m, B_{x1}), \quad n = \text{poly}\left(m, B_x, \frac{1}{\delta}, \frac{1}{\epsilon}, \frac{1}{\beta}, \frac{1}{\rho}\right), \tag{221}$$

*trained by Algorithm 1 with hinge loss, with probability at least $1 - \delta$ over the initialization, with proper hyper-parameters, there exists $t \in [T]$ such that*

$$\Pr[\text{sign}(f_{\Xi^{(t)}}(\mathbf{x})) \neq y] \leq \epsilon. \tag{222}$$

*Proof of Theorem E.4.* Let $\tilde{b} = d^\tau B_{x1}\sigma_w$, where $\tau$ is a number large enough to satisfy $d^{\tau/2 - \frac{1}{4}} > \Omega\left(\frac{\sqrt{B_{x2}}}{\rho}\right)$ and $O\left(\frac{B_{x1}B_{x2}^{\frac{1}{4}}}{\beta\sqrt{\rho}d^{\tau/4 - \frac{1}{8}}}\right) \leq \frac{\epsilon}{2}$. By Lemma E.3, we have $f^* \in \mathcal{F}_{d,r,B_F,S_{p,\gamma},B_G}$, where $r = 2, B_F = (B_{a1}, B_{a2}, B_b) = \left(\frac{1}{\beta}, \frac{\sqrt{2}}{\beta}, \frac{1}{B_{x1}^2}\right), p = \frac{1}{2}, \gamma = \Theta\left(\frac{\sqrt{B_{x2}}}{\rho d^{\tau/2 - \frac{1}{4}}}\right), B_G = \rho - \Theta\left(\frac{\sqrt{B_{x2}}}{d^{\tau/2 - \frac{1}{4}}}\right)$.
We also have $\text{OPT}_{d,r,B_F,S_{p,\gamma},B_G} = 0$.

Adjust $\sigma_w$ such that $\tilde{b} = d^\tau B_{x1}\sigma_w = \Theta\left(\frac{B_G^{\frac{1}{4}}B_{a2}B_b^{\frac{3}{4}}}{\sqrt{r}B_{a1}}\right)$. Injecting above parameters into Theorem 3.12, we have with probability at least $1 - \delta$ over the initialization, with proper hyper-parameters, there exists $t \in [T]$ such that

$$\Pr[\text{sign}(f_{\Xi^{(t)}}(\mathbf{x})) \neq y] \leq O\left(\frac{B_{x1}B_{x2}^{\frac{1}{4}}}{\beta\sqrt{\rho}d^{\tau/4 - \frac{1}{8}}}\right) + O\left(\frac{B_{x1}B_{x2}^{\frac{1}{4}}(\log n)^{\frac{1}{4}}}{\beta\sqrt{\rho}n^{\frac{1}{4}}}\right) + \epsilon/2 \leq \epsilon. \tag{223}$$

$\square$

## E.2 Mixture of Gaussians

We recap the problem setup in Section 4.1 for readers' convenience.

### E.2.1 Problem Setup

**Data Distributions.** We follow the notations from [99]. The data are from a mixture of $r$ high-dimensional Gaussians, and each Gaussian is assigned to one of two possible labels in $\mathcal{Y} = \{\pm 1\}$. Let $\mathcal{S}(y) \subseteq [r]$ denote the set of indices of the Gaussians associated with the label $y$. The data distribution is then:

$$q(\mathbf{x}, y) = q(y)q(\mathbf{x}|y), \quad q(\mathbf{x}|y) = \sum_{j \in \mathcal{S}(y)} p_j \mathcal{N}_j(\mathbf{x}), \tag{224}$$

where $\mathcal{N}_j(\mathbf{x})$ is a multivariate normal distribution with mean $\mu_j$ and covariance $\Sigma_j$, and $p_j$ are chosen such that $q(\mathbf{x}, y)$ is correctly normalized.

We call this data distribution $\mathcal{D}_{mixture}$.

We will make some assumptions about the Gaussians, for which we first introduce some notations. For all $j \in [r]$, let $y_{(j)} \in \{+1, -1\}$ be the label for $\mathcal{N}_j(\mathbf{x})$.

$$D_j := \frac{\mu_j}{\|\mu_j\|_2}, \quad \tilde{\mu}_j := \mu_j/\sqrt{d}, \quad B_{\mu 1} := \min_{j \in [r]} \|\tilde{\mu}_j\|_2, \quad B_{\mu 2} := \max_{j \in [r]} \|\tilde{\mu}_j\|_2, \quad p_B := \min_{j \in [r]} p_j.$$

**Assumption E.5** (Mixture of Gaussians. Recap of Assumption 4.1). *Let $8 \leq \tau \leq d$ be a parameter that will control our final error guarantee. Assume*

- *Equiprobable labels:* $q(-1) = q(+1) = 1/2$.
- *For all $j \in [r]$, $\Sigma_j = \sigma_j I_{d \times d}$. Let $\sigma_B := \max_{j \in [r]} \sigma_j$ and $\sigma_{B+} := \max\{\sigma_B, B_{\mu 2}\}$.*
- $r \leq 2d, \quad p_B \geq \frac{1}{2d}, \quad \Omega\left(\frac{1}{d} + \sqrt{\frac{\tau \sigma_{B+}{}^2 \log d}{d}}\right) \leq B_{\mu 1} \leq B_{\mu 2} \leq d.$
- *The Gaussians are well-separated: for all $i \neq j \in [r]$, we have $-1 \leq \langle D_i, D_j \rangle \leq \theta$, where*
  $0 \leq \theta \leq \min\left\{\frac{1}{2r}, \frac{\sigma_{B+}}{B_{\mu 2}}\sqrt{\frac{\tau \log d}{d}}\right\}.$

Below, we define a sufficient condition that randomly initialized weights will fall in nice gradients set after the first gradient step update.

**Definition E.6** (Mixture of Gaussians: Subset of Nice Gradients Set). *Recall $\mathbf{w}_i^{(0)}$ is the weight for the $i$-th neuron at initialization. For all $j \in [r]$, let $S_{D_j, Sure} \subseteq [m]$ be those neurons that satisfy*

- $\left\langle \mathbf{w}_i^{(0)}, \mu_j \right\rangle \geq C_{Sure,1} \mathbf{b}_i,$
- $\left\langle \mathbf{w}_i^{(0)}, \mu_{j'} \right\rangle \leq C_{Sure,2} \mathbf{b}_i,$ *for all $j' \neq j, j' \in [r]$.*
- $\left\| \mathbf{w}_i^{(0)} \right\|_2 \leq \Theta(\sqrt{d}\sigma_w).$

### E.2.2 Mixture of Gaussians: Feature Learning

We show the important Lemma E.7 first and defer other Lemmas after it.

**Lemma E.7** (Mixture of Gaussians: Gradient Feature Set. Part statement of Lemma 4.3). *Let $C_{Sure,1} = \frac{3}{2}, C_{Sure,2} = \frac{1}{2}, \tilde{b} = C_b\sqrt{\tau d \log d}\sigma_w \sigma_{B+}$, where $C_b$ is a large enough universal constant. For $\mathcal{D}_{mixture}$ setting, we have $(D_j, +1) \in S_{p,\gamma,B_G}$ for all $j \in [r]$, where*

$$p = \Theta\left(\frac{B_{\mu 1}}{\sqrt{\tau \log d}\sigma_{B+} \cdot d^{\left(9C_b^2 \tau \sigma_{B+}{}^2/(2B_{\mu 1}^2)\right)}}\right), \quad \gamma = \frac{1}{d^{0.9\tau - 1.5}}, \tag{225}$$

$$B_G = p_B B_{\mu 1}\sqrt{d} - O\left(\frac{\sigma_{B+}}{d^{0.9\tau}}\right). \tag{226}$$

*Proof of Lemma E.7.* For all $j \in [r]$, by Lemma E.10, for all $i \in S_{D_j, Sure}$,

$$1 - \frac{\left|\left\langle G(\mathbf{w}_i^{(0)}, \mathbf{b}_i), D_j \right\rangle\right|}{\|G(\mathbf{w}_i^{(0)}, \mathbf{b}_i)\|_2} \tag{227}$$

$$\leq 1 - \frac{\left|\left\langle G(\mathbf{w}_i^{(0)}, \mathbf{b}_i), D_j \right\rangle\right|}{\sqrt{\left|\left\langle G(\mathbf{w}_i^{(0)}, \mathbf{b}_i), D_j \right\rangle\right|^2 + \max_{D_j^\top D_j^\perp = 0, \|D_j^\perp\|_2 = 1}\left|\left\langle G(\mathbf{w}_i^{(0)}, \mathbf{b}_i), D_j^\perp \right\rangle\right|^2}} \tag{228}$$

$$\leq 1 - \frac{\left|\left\langle G(\mathbf{w}_i^{(0)}, \mathbf{b}_i), D_j \right\rangle\right|}{\left|\left\langle G(\mathbf{w}_i^{(0)}, \mathbf{b}_i), D_j \right\rangle\right| + \max_{D_j^\top D_j^\perp = 0, \|D_j^\perp\|_2 = 1}\left|\left\langle G(\mathbf{w}_i^{(0)}, \mathbf{b}_i), D_j^\perp \right\rangle\right|} \tag{229}$$

$$\leq 1 - \frac{1}{1 + \frac{B_{\mu 2}O\left(\frac{1}{d^{\tau - \frac{1}{2}}}\right) + \sigma_{B+}O\left(\frac{1}{d^{0.9\tau}}\right)}{p_j B_{\mu 1}\sqrt{d}\left(1 - O\left(\frac{1}{d^\tau}\right)\right) - B_{\mu 2}O\left(\frac{1}{d^{\tau - \frac{1}{2}}}\right) - \sigma_{B+}O\left(\frac{1}{d^{0.9\tau}}\right)}} \tag{230}$$

$$\leq \frac{\sigma_{B+}O\left(\frac{1}{d^{0.9\tau}}\right)}{p_j B_{\mu 1}\sqrt{d} - \sigma_{B+}O\left(\frac{1}{d^{0.9\tau}}\right)} \tag{231}$$

$$< \frac{1}{d^{0.9\tau - 1.5}} = \gamma, \tag{232}$$

where the last inequality follows $B_{\mu 1} \geq \Omega\left(\sigma_{B+}\sqrt{\frac{\tau \log d}{d}}\right)$.

Thus, we have $G(\mathbf{w}_i^{(0)}, \mathbf{b}_i) \in \mathcal{C}_{D_j,\gamma}$ and $\left|\left\langle G(\mathbf{w}_i^{(0)}, \mathbf{b}_i), D_j \right\rangle\right| \leq \|G(\mathbf{w}_i^{(0)}, \mathbf{b}_i)\|_2 \leq B_{x1}$, $\frac{\mathbf{b}_i}{|\mathbf{b}_i|} = +1$. Thus, by Lemma E.8, we have

$$\Pr_{\mathbf{w},b}\left[G(\mathbf{w},b) \in \mathcal{C}_{D_j,\gamma} \text{ and } \|G(\mathbf{w},b)\|_2 \geq B_G \text{ and } \frac{b}{|b|} = +1\right] \tag{233}$$

$$\geq \Pr\left[i \in S_{D_j,Sure}\right] \tag{234}$$

$$\geq p. \tag{235}$$

Thus, $(D_j, +1) \in S_{p,\gamma,B_G}$. We finish the proof. $\qquad\square$

Below are Lemmas used in the proof of Lemma E.7. In Lemma E.8, we calculate $p$ used in $S_{p,\gamma,B_G}$.

**Lemma E.8** (Mixture of Gaussians: Geometry at Initialization. Lemma B.2 in [7]). *Assume the same conditions as in Lemma E.7, recall for all $i \in [m]$, $\mathbf{w}_i^{(0)} \sim \mathcal{N}(0, \sigma_w^2 I_{d\times d})$, over the random initialization, we have for all $i \in [m], j \in [r]$,*

$$\Pr\left[i \in S_{D_j,Sure}\right] \geq \Theta\left(\frac{B_{\mu 1}}{\sqrt{\tau \log d}\sigma_{B+} \cdot d^{\left(9C_b^2 \tau \sigma_{B+}^2/(2B_{\mu 1}^2)\right)}}\right). \tag{236}$$

*Proof of Lemma E.8.* Recall for all $l \in [r]$, $\tilde{\mu}_l = \mu_l/\sqrt{d}$.

WLOG, let $j = r$. For all $l \in [r-1]$. We define $Z_1 = \{l \in [r-1] : \langle D_l, D_r \rangle \geq -\theta\}$ and $Z_2 = \{l \in [r-1] : -1 < \langle D_l, D_r \rangle < -\theta\}$. WLOG, let $Z_1 = [r_1]$, $Z_2 = \{r_1 + 1, \ldots, r_2\}$, where $0 \leq r_1 \leq r_2 \leq r - 1$. We define the following events

$$\zeta_l = \left\{\left\langle \mathbf{w}_i^{(0)}, \mu_l \right\rangle \leq C_{Sure,2}\mathbf{b}_i\right\}, \hat{\zeta}_l = \left\{\left|\left\langle \mathbf{w}_i^{(0)}, \mu_l \right\rangle\right| \leq C_{Sure,2}\mathbf{b}_i\right\}. \tag{237}$$

We define space $A = \text{span}(\mu_1, \ldots, \mu_{r_1})$ and $\hat{\mu}_r = P_{A^\perp}\mu_r$, where $P_{A^\perp}$ is the projection operator on the complementary space of $A$. For $l \in Z_2$, we also define $\dot{\mu}_l = \mu_l - \frac{\langle \mu_l, \mu_r \rangle \mu_r}{\|\mu_r\|_2^2}$, and the event

$$\dot{\zeta}_l = \left\{\left\langle \mathbf{w}_i^{(0)}, \dot{\mu}_l \right\rangle \leq C_{Sure,2}\mathbf{b}_i\right\}, \hat{\dot{\zeta}}_l = \left\{\left|\left\langle \mathbf{w}_i^{(0)}, \dot{\mu}_l \right\rangle\right| \leq C_{Sure,2}\mathbf{b}_i\right\}. \tag{238}$$

For $l \in Z_2$, we have $\mu_l = \dot{\mu}_l - \rho\mu_r$, where $\rho \geq 0$. So $\langle \mathbf{w}, \mu_l \rangle = \langle \mathbf{w}, \dot{\mu}_l \rangle - \rho\langle \mathbf{w}, \mu_r \rangle \leq \langle \mathbf{w}, \dot{\mu}_l \rangle$ when $\langle \mathbf{w}, \mu_r \rangle \geq 0$. As a result, we have

$$\dot{\zeta}_l \cap \left\{\left\langle \mathbf{w}_i^{(0)}, \mu_r \right\rangle \geq C_{Sure,1}\mathbf{b}_i\right\} \subseteq \zeta_l \cap \left\{\left\langle \mathbf{w}_i^{(0)}, \mu_r \right\rangle \geq C_{Sure,1}\mathbf{b}_i\right\}. \tag{239}$$

By Assumption 4.1, we have

$$\frac{1}{2} \leq 1 - r\theta \leq 1 - r_1\theta \leq \frac{\|\hat{\mu}_r\|_2}{\|\mu_r\|_2} \leq 1. \tag{240}$$

We also have,

$$\Pr\left[\left\langle \mathbf{w}_i^{(0)}, \mu_r \right\rangle \geq C_{Sure,1}\mathbf{b}_i, \zeta_1, \ldots, \zeta_{r-1}\right] \tag{241}$$

$$= \Pr\left[\left\langle \mathbf{w}_i^{(0)}, \mu_r \right\rangle \geq C_{Sure,1}\mathbf{b}_i, \zeta_1, \ldots, \zeta_{r_2}\right] \tag{242}$$

$$\geq \Pr\left[\left\langle \mathbf{w}_i^{(0)}, \mu_r \right\rangle \geq C_{Sure,1}\mathbf{b}_i, \zeta_1, \ldots, \zeta_{r_1}, \dot{\zeta}_{r_1+1}, \ldots, \dot{\zeta}_{r_2}\right] \tag{243}$$

$$\geq \Pr\left[\left\langle \mathbf{w}_i^{(0)}, \mu_r \right\rangle \geq C_{Sure,1}\mathbf{b}_i, \hat{\zeta}_1, \ldots, \hat{\zeta}_{r_1}, \hat{\dot{\zeta}}_{r_1+1}, \ldots, \hat{\dot{\zeta}}_{r_2}\right] \tag{244}$$

$$= \underbrace{\Pr\left[\left\langle \mathbf{w}_i^{(0)}, \mu_r \right\rangle \geq C_{Sure,1}\mathbf{b}_i \middle| \hat{\zeta}_1, \ldots, \hat{\zeta}_{r_1}, \hat{\dot{\zeta}}_{r_1+1}, \ldots, \hat{\dot{\zeta}}_{r_2}\right]}_{p_r} \underbrace{\Pr\left[\hat{\zeta}_1, \ldots, \hat{\zeta}_{r_1}, \hat{\dot{\zeta}}_{r_1+1}, \ldots, \hat{\dot{\zeta}}_{r_2}\right]}_{\Pi_{l\in[r_2]}p_l}.$$

For the first condition in Definition E.6, we have,

$$p_r = \Pr\left[\left\langle \mathbf{w}_i^{(0)}, \mu_r \right\rangle \geq C_{Sure,1}\mathbf{b}_i \middle| \hat{\zeta}_1, \ldots, \hat{\zeta}_{r_1}, \hat{\hat{\zeta}}_{r_1+1}, \ldots, \hat{\hat{\zeta}}_{r_2}\right] \tag{245}$$

$$= \Pr\left[\left\langle \mathbf{w}_i^{(0)}, \hat{\mu}_r + \mu_r - \hat{\mu}_r \right\rangle \geq C_{Sure,1}\mathbf{b}_i \middle| \hat{\zeta}_1, \ldots, \hat{\zeta}_{r_1}\right] \tag{246}$$

$$\geq \Pr\left[\left\langle \mathbf{w}_i^{(0)}, \hat{\mu}_r + \mu_r - \hat{\mu}_r \right\rangle \geq C_{Sure,1}\mathbf{b}_i, \ \left\langle \mathbf{w}_i^{(0)}, \mu_r - \hat{\mu}_r \right\rangle \geq 0 \middle| \hat{\zeta}_1, \ldots, \hat{\zeta}_{r_1}\right] \tag{247}$$

$$= \Pr\left[\left\langle \mathbf{w}_i^{(0)}, \hat{\mu}_r + \mu_r - \hat{\mu}_r \right\rangle \geq C_{Sure,1}\mathbf{b}_i \middle| \left\langle \mathbf{w}_i^{(0)}, \mu_r - \hat{\mu}_r \right\rangle \geq 0, \hat{\zeta}_1, \ldots, \hat{\zeta}_{r_1}\right] \tag{248}$$

$$\cdot \Pr\left[\left\langle \mathbf{w}_i^{(0)}, \mu_r - \hat{\mu}_r \right\rangle \geq 0 \middle| \hat{\zeta}_1, \ldots, \hat{\zeta}_{r_1}\right] \tag{249}$$

$$= \frac{1}{2}\Pr\left[\left\langle \mathbf{w}_i^{(0)}, \hat{\mu}_r + \mu_r - \hat{\mu}_r \right\rangle \geq C_{Sure,1}\mathbf{b}_i \middle| \left\langle \mathbf{w}_i^{(0)}, \mu_r - \hat{\mu}_r \right\rangle \geq 0, \hat{\zeta}_1, \ldots, \hat{\zeta}_{r_1}\right] \tag{250}$$

$$\geq \frac{1}{2}\Pr\left[\left\langle \mathbf{w}_i^{(0)}, \hat{\mu}_r \right\rangle \geq C_{Sure,1}\mathbf{b}_i \middle| \left\langle \mathbf{w}_i^{(0)}, \mu_r - \hat{\mu}_r \right\rangle \geq 0, \hat{\zeta}_1, \ldots, \hat{\zeta}_{r_1}\right] \tag{251}$$

$$= \frac{1}{2}\Pr\left[\left\langle \mathbf{w}_i^{(0)}, \hat{\mu}_r \right\rangle \geq C_{Sure,1}\mathbf{b}_i\right] \tag{252}$$

$$\geq \Theta\left(\frac{\|\tilde{\mu}_r\|_2}{\sqrt{\tau \log d}\sigma_{B+} \cdot d^{\left(9C_b^2\tau\sigma_{B+}{}^2/(2\|\tilde{\mu}_r\|_2^2)\right)}}\right), \tag{253}$$

where the last equality following that $\hat{\mu}_r$ is orthogonal with $\mu_1, \ldots, \mu_{r_1}$ and the property of the standard Gaussian vector, and the last inequality follows Lemma F.6.

For the second condition in Definition E.6, by Lemma F.6, we have,

$$p_1 = \Pr\left[\hat{\zeta}_1\right] = 1 - \Theta\left(\frac{\|\tilde{\mu}_1\|_2}{\sqrt{\tau \log d}\sigma_{B+} \cdot d^{\left(C_b^2\tau\sigma_{B+}{}^2/(8\|\tilde{\mu}_1\|_2^2)\right)}}\right) \tag{254}$$

$$p_2 = \Pr\left[\hat{\zeta}_2\middle|\hat{\zeta}_1\right] \geq \Pr\left[\hat{\zeta}_2\right] \geq 1 - \Theta\left(\frac{\|\tilde{\mu}_2\|_2}{\sqrt{\tau \log d}\sigma_{B+} \cdot d^{\left(C_b^2\tau\sigma_{B+}{}^2/(8\|\tilde{\mu}_2\|_2^2)\right)}}\right) \tag{255}$$

$$\vdots \tag{256}$$

$$p_{r-1} = \Pr\left[\hat{\hat{\zeta}}_{r_2}\middle|\hat{\zeta}_1, \ldots, \hat{\zeta}_{r_1}, \hat{\hat{\zeta}}_{r_1+1}, \ldots, \hat{\hat{\zeta}}_{r_2}\right] \geq \Pr\left[\hat{\hat{\zeta}}_{r_2}\right] \geq \Pr\left[\hat{\zeta}_{r_2}\right] \tag{257}$$

$$\geq 1 - \Theta\left(\frac{\|\tilde{\mu}_{r-1}\|_2}{\sqrt{\tau \log d}\sigma_{B+} \cdot d^{\left(C_b^2\tau\sigma_{B+}{}^2/(8\|\tilde{\mu}_{r_2}\|_2^2)\right)}}\right). \tag{258}$$

On the other hand, if $X$ is a $\chi^2(k)$ random variable. Then we have

$$\Pr(X \geq k + 2\sqrt{kx} + 2x) \leq e^{-x}. \tag{259}$$

Therefore, by assumption $B_{\mu1} \geq \Omega\left(\sigma_{B+}\sqrt{\frac{\tau \log d}{d}}\right)$, we have

$$\Pr\left(\frac{1}{\sigma_w^2}\left\|\mathbf{w}_i^{(0)}\right\|_2^2 \geq d + 2\sqrt{\left(9C_b^2\tau\sigma_{B+}{}^2/(2B_{\mu1}^2) + 2\right)d\log d}\right. \tag{260}$$

$$\left. + 2\left(9C_b^2\tau\sigma_{B+}{}^2/(2B_{\mu1}^2) + 2\right)\log d\right) \tag{261}$$

$$\leq O\left(\frac{1}{d^2 \cdot d^{\left(9C_b^2\tau\sigma_{B+}{}^2/(2B_{\mu1}^2)\right)}}\right). \tag{262}$$

Recall $B_{\mu 1} = \min_{j \in [r]} \|\tilde{\mu}_j\|_2$, $B_{\mu 2} = \max_{j \in [r]} \|\tilde{\mu}_j\|_2$. Thus, by union bound, we have

$$\Pr\left[i \in S_{D_j, Sure}\right] \tag{263}$$

$$\geq \Pi_{l \in [r]} p_l - O\left(\frac{1}{d^2 \cdot d^{\left(9C_b^2 \tau \sigma_{B+}{}^2/(2B_{\mu 1}^2)\right)}}\right) \tag{264}$$

$$\geq \Theta\left(\frac{B_{\mu 1}}{\sqrt{\tau \log d}\sigma_{B+} \cdot d^{\left(9C_b^2 \tau \sigma_{B+}{}^2/(2B_{\mu 1}^2)\right)}} \cdot \left(1 - \frac{rB_{\mu 2}}{\sqrt{\tau \log d}\sigma_{B+} \cdot d^{\left(C_b^2 \tau \sigma_{B+}{}^2/(8B_{\mu 2}^2)\right)}}\right)\right) \tag{265}$$

$$- O\left(\frac{1}{d^2 \cdot d^{\left(9C_b^2 \tau \sigma_{B+}{}^2/(2B_{\mu 1}^2)\right)}}\right) \tag{266}$$

$$\geq \Theta\left(\frac{B_{\mu 1}}{\sqrt{\tau \log d}\sigma_{B+} \cdot d^{\left(9C_b^2 \tau \sigma_{B+}{}^2/(2B_{\mu 1}^2)\right)}}\right). \tag{267}$$

$\square$

In Lemma E.9, we compute the activation pattern for the neurons in $S_{D_j, Sure}$.

**Lemma E.9** (Mixture of Gaussians: Activation Pattern). *Assume the same conditions as in Lemma E.7, for all $j \in [r], i \in S_{D_j, Sure}$, we have*

*(1) When $\mathbf{x} \sim \mathcal{N}_j(\mu_j, \sigma_j I_{d \times d})$, the activation probability satisfies,*

$$\Pr_{\mathbf{x} \sim \mathcal{N}_j(\mu_j, \sigma_j I_{d \times d})}\left[\left\langle \mathbf{w}_i^{(0)}, \mathbf{x} \right\rangle - \mathbf{b}_i \geq 0\right] \geq 1 - O\left(\frac{1}{d^\tau}\right). \tag{268}$$

*(2) For all $j' \neq j, j' \in [r]$, when $\mathbf{x} \sim \mathcal{N}_{j'}(\mu_{j'}, \Sigma_{j'})$, the activation probability satisfies,*

$$\Pr_{\mathbf{x} \sim \mathcal{N}_{j'}(\mu_{j'}, \sigma_{j'} I_{d \times d})}\left[\left\langle \mathbf{w}_i^{(0)}, \mathbf{x} \right\rangle - \mathbf{b}_i \geq 0\right] \leq O\left(\frac{1}{d^\tau}\right). \tag{269}$$

*Proof of Lemma E.9.* In the proof, we need $\tilde{b} = C_b \sqrt{\tau d \log d}\sigma_w \sigma_{B+}$, where $C_b$ is a large enough universal constant. For the first statement, when $\mathbf{x} \sim \mathcal{N}_j(\mu_j, \sigma_j I_{d \times d})$, by $C_{Sure,1} \geq \frac{3}{2}$, we have

$$\Pr_{\mathbf{x} \sim \mathcal{N}_j(\mu_j, \sigma_j I_{d \times d})}\left[\left\langle \mathbf{w}_i^{(0)}, \mathbf{x} \right\rangle - \mathbf{b}_i \geq 0\right] \geq \Pr_{\mathbf{x} \sim \mathcal{N}(0, \sigma_j I_{d \times d})}\left[\left\langle \mathbf{w}_i^{(0)}, \mathbf{x} \right\rangle \geq (1 - C_{Sure,1})\mathbf{b}_i\right] \tag{270}$$

$$\geq \Pr_{\mathbf{x} \sim \mathcal{N}(0, \sigma_j I_{d \times d})}\left[\left\langle \mathbf{w}_i^{(0)}, \mathbf{x} \right\rangle \geq -\frac{\mathbf{b}_i}{2}\right] \tag{271}$$

$$= 1 - \Pr_{\mathbf{x} \sim \mathcal{N}(0, \sigma_j I_{d \times d})}\left[\left\langle \mathbf{w}_i^{(0)}, \mathbf{x} \right\rangle \leq -\frac{\mathbf{b}_i}{2}\right] \tag{272}$$

$$\geq 1 - \exp\left(-\frac{\mathbf{b}_i{}^2}{\Theta(d\sigma_w^2 \sigma_j^2)}\right) \tag{273}$$

$$\geq 1 - O\left(\frac{1}{d^\tau}\right), \tag{274}$$

where the third inequality follows the Chernoff bound and symmetricity of the Gaussian vector.

For the second statement, we prove similarly by $0 < C_{Sure,2} \leq \frac{1}{2}$. $\square$

Then, Lemma E.10 gives gradients of neurons in $S_{D_j, Sure}$. It shows that these gradients are highly aligned with $D_j$.

**Lemma E.10** (Mixture of Gaussians: Feature Emergence). *Assume the same conditions as in Lemma E.7, for all $j \in [r]$, $i \in S_{D_j, Sure}$, we have*

$$\left\langle \mathbb{E}_{(\mathbf{x}, y)}\left[y\sigma'\left(\left\langle \mathbf{w}_i^{(0)}, \mathbf{x} \right\rangle - \mathbf{b}_i\right)\mathbf{x}\right], y_{(j)} D_j \right\rangle \tag{275}$$

$$\geq p_j B_{\mu 1} \sqrt{d}\left(1 - O\left(\frac{1}{d^\tau}\right)\right) - B_{\mu 2} O\left(\frac{1}{d^{\tau - \frac{1}{2}}}\right) - \sigma_{B+} O\left(\frac{1}{d^{0.9\tau}}\right). \tag{276}$$

*For any unit vector $D_j^{\perp}$ which is orthogonal with $D_j$, we have*

$$\left| \left\langle \mathbb{E}_{(\mathbf{x},y)} \left[ y\sigma' \left( \left\langle \mathbf{w}_i^{(0)}, \mathbf{x} \right\rangle - \mathbf{b}_i \right) \mathbf{x} \right], D_j^{\perp} \right\rangle \right| \leq B_{\mu 2} O \left( \frac{1}{d^{\tau - \frac{1}{2}}} \right) + \sigma_{B+} O \left( \frac{1}{d^{0.9\tau}} \right). \tag{277}$$

*Proof of Lemma E.10.* For all $j \in [r]$, $i \in S_{D_j, Sure}$, we have

$$\mathbb{E}_{(\mathbf{x},y)} \left[ y\sigma' \left( \left\langle \mathbf{w}_i^{(0)}, \mathbf{x} \right\rangle - \mathbf{b}_i \right) \mathbf{x} \right] \tag{278}$$

$$= \sum_{l \in [r]} p_l \mathbb{E}_{\mathbf{x} \sim \mathcal{N}_l(\mathbf{x})} \left[ y\sigma' \left( \left\langle \mathbf{w}_i^{(0)}, \mathbf{x} \right\rangle - \mathbf{b}_i \right) \mathbf{x} \right] \tag{279}$$

$$= \sum_{l \in [r]} p_l y_{(l)} \mathbb{E}_{\mathbf{x} \sim \mathcal{N}(0, \sigma_l I_{d \times d})} \left[ \sigma' \left( \left\langle \mathbf{w}_i^{(0)}, \mathbf{x} + \mu_l \right\rangle - \mathbf{b}_i \right) (\mathbf{x} + \mu_l) \right]. \tag{280}$$

Thus, by Lemma F.7 and Lemma E.9,

$$\left\langle \mathbb{E}_{(\mathbf{x},y)} \left[ y\sigma' \left( \left\langle \mathbf{w}_i^{(0)}, \mathbf{x} \right\rangle - \mathbf{b}_i \right) \mathbf{x} \right], y_{(j)} D_j \right\rangle \tag{281}$$

$$= p_j \mathbb{E}_{\mathbf{x} \sim \mathcal{N}(0, \sigma_j I_{d \times d})} \left[ \sigma' \left( \left\langle \mathbf{w}_i^{(0)}, \mathbf{x} + \mu_j \right\rangle - \mathbf{b}_i \right) (\mathbf{x} + \mu_j)^{\top} D_j \right] \tag{282}$$

$$+ \sum_{l \in [r], l \neq j} p_l y_{(l)} y_{(j)} \mathbb{E}_{\mathbf{x} \sim \mathcal{N}(0, \sigma_l I_{d \times d})} \left[ \sigma' \left( \left\langle \mathbf{w}_i^{(0)}, \mathbf{x} + \mu_l \right\rangle - \mathbf{b}_i \right) (\mathbf{x} + \mu_l)^{\top} D_j \right] \tag{283}$$

$$\geq p_j \mu_j^{\top} D_j \left( 1 - O \left( \frac{1}{d^{\tau}} \right) \right) - \sum_{l \in [r], l \neq j} p_l |\mu_l^{\top} D_j| O \left( \frac{1}{d^{\tau}} \right) \tag{284}$$

$$- p_j \left| \mathbb{E}_{\mathbf{x} \sim \mathcal{N}(0, \sigma_j I)} \left[ \sigma' \left( \left\langle \mathbf{w}_i^{(0)}, \mathbf{x} + \mu_j \right\rangle - \mathbf{b}_i \right) \mathbf{x}^{\top} D_j \right] \right| \tag{285}$$

$$- \sum_{l \in [r], l \neq j} p_l \left| \mathbb{E}_{\mathbf{x} \sim \mathcal{N}(0, \sigma_l I)} \left[ \sigma' \left( \left\langle \mathbf{w}_i^{(0)}, \mathbf{x} + \mu_l \right\rangle - \mathbf{b}_i \right) \mathbf{x}^{\top} D_j \right] \right| \tag{286}$$

$$\geq p_j B_{\mu 1} \sqrt{d} \left( 1 - O \left( \frac{1}{d^{\tau}} \right) \right) - B_{\mu 2} O \left( \frac{1}{d^{\tau - \frac{1}{2}}} \right) \tag{287}$$

$$- p_j \left| \mathbb{E}_{\mathbf{x} \sim \mathcal{N}(0, \sigma_j I)} \left[ \left( 1 - \sigma' \left( \left\langle \mathbf{w}_i^{(0)}, \mathbf{x} + \mu_j \right\rangle - \mathbf{b}_i \right) - 1 \right) \mathbf{x}^{\top} D_j \right] \right| \tag{288}$$

$$- \sum_{l \in [r], l \neq j} p_l \left| \mathbb{E}_{\mathbf{x} \sim \mathcal{N}(0, \sigma_l I)} \left[ \sigma' \left( \left\langle \mathbf{w}_i^{(0)}, \mathbf{x} + \mu_l \right\rangle - \mathbf{b}_i \right) \mathbf{x}^{\top} D_j \right] \right| \tag{289}$$

$$= p_j B_{\mu 1} \sqrt{d} \left( 1 - O \left( \frac{1}{d^{\tau}} \right) \right) - B_{\mu 2} O \left( \frac{1}{d^{\tau - \frac{1}{2}}} \right) \tag{290}$$

$$- p_j \left| \mathbb{E}_{\mathbf{x} \sim \mathcal{N}(0, \sigma_j I)} \left[ \left( 1 - \sigma' \left( \left\langle \mathbf{w}_i^{(0)}, \mathbf{x} + \mu_j \right\rangle - \mathbf{b}_i \right) \right) \mathbf{x}^{\top} D_j \right] \right| \tag{291}$$

$$- \sum_{l \in [r], l \neq j} p_l \left| \mathbb{E}_{\mathbf{x} \sim \mathcal{N}(0, \sigma_l I)} \left[ \sigma' \left( \left\langle \mathbf{w}_i^{(0)}, \mathbf{x} + \mu_l \right\rangle - \mathbf{b}_i \right) \mathbf{x}^{\top} D_j \right] \right| \tag{292}$$

$$\geq p_j B_{\mu 1} \sqrt{d} \left( 1 - O \left( \frac{1}{d^{\tau}} \right) \right) - B_{\mu 2} O \left( \frac{1}{d^{\tau - \frac{1}{2}}} \right) - \sigma_{B+} O \left( \frac{1}{d^{0.9\tau}} \right). \tag{293}$$

For any unit vector $D_j^\perp$ which is orthogonal with $D_j$, similarly, we have

$$\left| \left\langle \mathbb{E}_{(\mathbf{x},y)} \left[ y\sigma' \left( \left\langle \mathbf{w}_i^{(0)}, \mathbf{x} \right\rangle - \mathbf{b}_i \right) \mathbf{x} \right], D_j^\perp \right\rangle \right| \tag{294}$$

$$\leq p_j \left| \mathbb{E}_{\mathbf{x} \sim \mathcal{N}(0,\sigma_j I)} \left[ \sigma' \left( \left\langle \mathbf{w}_i^{(0)}, \mathbf{x} + \mu_j \right\rangle - \mathbf{b}_i \right) \mathbf{x}^\top D_j^\perp \right] \right| \tag{295}$$

$$+ \sum_{l \in [r], l \neq j} p_l \left| \mathbb{E}_{\mathbf{x} \sim \mathcal{N}(0,\sigma_l I)} \left[ \sigma' \left( \left\langle \mathbf{w}_i^{(0)}, \mathbf{x} + \mu_l \right\rangle - \mathbf{b}_i \right) (\mathbf{x} + \mu_l)^\top D_j^\perp \right] \right| \tag{296}$$

$$\leq B_{\mu 2} O \left( \frac{1}{d^{\tau - \frac{1}{2}}} \right) + p_j \left| \mathbb{E}_{\mathbf{x} \sim \mathcal{N}(0,\sigma_j I)} \left[ \sigma' \left( \left\langle \mathbf{w}_i^{(0)}, \mathbf{x} + \mu_j \right\rangle - \mathbf{b}_i \right) \mathbf{x}^\top D_j^\perp \right] \right| \tag{297}$$

$$+ \sum_{l \in [r], l \neq j} p_l \left| \mathbb{E}_{\mathbf{x} \sim \mathcal{N}(0,\sigma_l I)} \left[ \sigma' \left( \left\langle \mathbf{w}_i^{(0)}, \mathbf{x} + \mu_l \right\rangle - \mathbf{b}_i \right) \mathbf{x}^\top D_j^\perp \right] \right| \tag{298}$$

$$\leq B_{\mu 2} O \left( \frac{1}{d^{\tau - \frac{1}{2}}} \right) + \sigma_{B+} O \left( \frac{1}{d^{0.9\tau}} \right). \tag{299}$$

$\square$

### E.2.3 Mixture of Gaussians: Final Guarantee

**Lemma E.11** (Mixture of Gaussians: Existence of Good Networks. Part statement of Lemma 4.3)**.**
*Assume the same conditions as in Lemma E.7. Define*

$$f^*(\mathbf{x}) = \sum_{j=1}^r \frac{y_{(j)}}{\sqrt{\tau \log d}\sigma_{B+}} \left[ \sigma \left( \langle D_j, \mathbf{x} \rangle - 2\sqrt{\tau \log d}\sigma_{B+} \right) \right]. \tag{300}$$

*For $\mathcal{D}_{mixture}$ setting, we have $f^* \in \mathcal{F}_{d,r,B_F,S_{p,\gamma,B_G}}$, where $B_F = (B_{a1}, B_{a2}, B_b) = \left( \frac{1}{\sqrt{\tau \log d}\sigma_{B+}}, \frac{\sqrt{r}}{\sqrt{\tau \log d}\sigma_{B+}}, 2\sqrt{\tau \log d}\sigma_{B+} \right)$, $p = \Theta\left( \frac{B_{\mu 1}}{\sqrt{\tau \log d}\sigma_{B+} \cdot d^{\left(9C_b^2 \tau \sigma_{B+}{}^2/(2B_{\mu 1}^2)\right)}} \right)$, $\gamma = \frac{1}{d^{0.9\tau - 1.5}}$, $B_G = p_B B_{\mu 1}\sqrt{d} - O\left(\frac{\sigma_{B+}}{d^{0.9\tau}}\right)$ and $B_{x1} = (B_{\mu 2} + \sigma_{B+})\sqrt{d}, B_{x2} = (B_{\mu 2} + \sigma_{B+})^2 d$. We also have $\mathrm{OPT}_{d,r,B_F,S_{p,\gamma,B_G}} \leq \frac{3}{d^\tau} + \frac{4}{d^{0.9\tau - 1}\sqrt{\tau \log d}}$.*

*Proof of Lemma E.11.* We can check $B_{x1} = (B_{\mu 2} + \sigma_{B+})\sqrt{d}, B_{x2} = (B_{\mu 2} + \sigma_{B+})^2 d$ by direct calculation. By Lemma E.7, we have $f^* \in \mathcal{F}_{d,r,B_F,S_{p,\gamma,B_G}}$.

For any $j \in [r]$, by $B_{\mu 1} \geq \Omega\left( \sigma_{B+}\sqrt{\frac{\tau \log d}{d}} \right) \geq 4\sigma_{B+}\sqrt{\frac{\tau \log d}{d}}$, we have

$$\Pr_{\mathbf{x} \sim \mathcal{N}_j(\mu_j, \sigma_j I_{d \times d})} \left[ \langle D_j, \mathbf{x} \rangle - 2\sqrt{\tau \log d}\sigma_{B+} \geq \sqrt{\tau \log d}\sigma_{B+} \right] \tag{301}$$

$$= \Pr_{\mathbf{x} \sim \mathcal{N}_j(0, \sigma_j I_{d \times d})} \left[ \langle D_j, \mathbf{x} \rangle + \|\mu_j\|_2 - 2\sqrt{\tau \log d}\sigma_{B+} \geq \sqrt{\tau \log d}\sigma_{B+} \right] \tag{302}$$

$$\geq \Pr_{\mathbf{x} \sim \mathcal{N}_j(0, \sigma_j I_{d \times d})} \left[ \langle D_j, \mathbf{x} \rangle + \sqrt{d}B_{\mu 1} - 2\sqrt{\tau \log d}\sigma_{B+} \geq \sqrt{\tau \log d}\sigma_{B+} \right] \tag{303}$$

$$\geq \Pr_{\mathbf{x} \sim \mathcal{N}_j(0, \sigma_j I_{d \times d})} \left[ \langle D_j, \mathbf{x} \rangle \geq -\sqrt{\tau \log d}\sigma_{B+} \right] \tag{304}$$

$$\geq 1 - \frac{1}{d^\tau}, \tag{305}$$

where the last inequality follows Chernoff bound.

For any $l \neq j, l \in [r]$, by $\theta \leq \frac{\sigma_{B+}}{B_{\mu 2}} \sqrt{\frac{\tau \log d}{d}}$, we have

$$\Pr_{\mathbf{x} \sim \mathcal{N}_j(\mu_j, \sigma_j I_{d \times d})} \left[ \langle D_l, \mathbf{x} \rangle - 2\sqrt{\tau \log d}\sigma_{B+} \geq 0 \right] \tag{306}$$

$$\leq \Pr_{\mathbf{x} \sim \mathcal{N}_j(0, \sigma_j I_{d \times d})} \left[ \langle D_l, \mathbf{x} \rangle + \theta B_{\mu 2}\sqrt{d} - 2\sqrt{\tau \log d}\sigma_{B+} \geq 0 \right] \tag{307}$$

$$\leq \Pr_{\mathbf{x} \sim \mathcal{N}_j(0, \sigma_j I_{d \times d})} \left[ \langle D_l, \mathbf{x} \rangle \geq \sqrt{\tau \log d}\sigma_{B+} \right] \tag{308}$$

$$\leq \frac{1}{d^\tau}. \tag{309}$$

Thus, we have

$$\Pr_{(\mathbf{x}, y) \sim \mathcal{D}_{mixture}} [y f^*(\mathbf{x}) > 1] \tag{310}$$

$$\geq \sum_{j \in [r]} p_j \left( \Pr_{\mathbf{x} \sim \mathcal{N}_j(\mu_j, \sigma_j I_{d \times d})} \left[ \langle D_j, \mathbf{x} \rangle - 2\sqrt{\tau \log d}\sigma_{B+} \geq \sqrt{\tau \log d}\sigma_{B+} \right] \right) \tag{311}$$

$$- \sum_{j \in [r]} p_j \left( \sum_{l \neq j, l \in [r]} \Pr_{\mathbf{x} \sim \mathcal{N}_j(\mu_j, \sigma_j I_{d \times d})} \left[ \langle D_l, \mathbf{x} \rangle - 2\sqrt{\tau \log d}\sigma_{B+} < 0 \right] \right) \tag{312}$$

$$\geq 1 - \frac{2}{d^\tau}. \tag{313}$$

We also have

$$\mathbb{E}_{(\mathbf{x}, y) \sim \mathcal{D}_{mixture}} [\mathbb{I}[y f^*(\mathbf{x}) \leq 1] |y f^*(\mathbf{x})|] \tag{314}$$

$$\leq \sum_{j \in [r]} p_j \left( \Pr_{\mathbf{x} \sim \mathcal{N}_j(\mu_j, \sigma_j I_{d \times d})} \left[ \langle D_j, \mathbf{x} \rangle - 2\sqrt{\tau \log d}\sigma_{B+} < \sqrt{\tau \log d}\sigma_{B+} \right] \frac{y_{(j)}^2 \sqrt{\tau \log d}\sigma_{B+}}{\sqrt{\tau \log d}\sigma_{B+}} \right)$$

$$+ \sum_{j \in [r]} p_j \left( \sum_{l \neq j, l \in [r]} \mathbb{E}_{\mathbf{x} \sim \mathcal{N}_j(\mu_j, \sigma_j I_{d \times d})} \left[ \sigma' \left[ \langle D_l, \mathbf{x} \rangle - 2\sqrt{\tau \log d}\sigma_{B+} > 0 \right] \frac{\langle D_l, \mathbf{x} \rangle - 2\sqrt{\tau \log d}\sigma_{B+}}{\sqrt{\tau \log d}\sigma_{B+}} \right] \right)$$

$$\leq \frac{1}{d^\tau} + \sum_{j \in [r]} p_j \left( \sum_{l \neq j, l \in [r]} \mathbb{E}_{\mathbf{x} \sim \mathcal{N}_j(0, \sigma_j I_{d \times d})} \left[ \sigma' \left[ \langle D_l, \mathbf{x} \rangle > \sqrt{\tau \log d}\sigma_{B+} \right] \frac{\langle D_l, \mathbf{x} \rangle - \sqrt{\tau \log d}\sigma_{B+}}{\sqrt{\tau \log d}\sigma_{B+}} \right] \right)$$

$$\leq \frac{1}{d^\tau} + \frac{1}{\sqrt{\tau \log d}} \sum_{j \in [r]} p_j \left( \sum_{l \neq j, l \in [r]} \mathbb{E}_{\mathbf{x} \sim \mathcal{N}_j(0, I_{d \times d})} \left[ \sigma' \left[ \langle D_l, \mathbf{x} \rangle > \sqrt{\tau \log d} \right] \langle D_l, \mathbf{x} \rangle \right] \right) \tag{315}$$

$$\leq \frac{1}{d^\tau} + \frac{4}{d^{0.9\tau - 1}\sqrt{\tau \log d}}, \tag{316}$$

where the second last inequality follows Lemma F.7 and $r \leq 2d$. Thus, we have

$$\text{OPT}_{d, r, B_F, S_{p, \gamma, B_G}} \leq \mathbb{E}_{(\mathbf{x}, y) \sim \mathcal{D}_{mixture}} [\ell(y f^*(\mathbf{x}))] \tag{317}$$

$$= \mathbb{E}_{(\mathbf{x}, y) \sim \mathcal{D}_{mixture}} [\mathbb{I}[y f^*(\mathbf{x}) \leq 1](1 - y f^*(\mathbf{x}))] \tag{318}$$

$$\leq \mathbb{E}_{(\mathbf{x}, y) \sim \mathcal{D}_{mixture}} [\mathbb{I}[y f^*(\mathbf{x}) \leq 1] |y f^*(\mathbf{x})|] + \mathbb{E}_{(\mathbf{x}, y) \sim \mathcal{D}_{mixture}} [\mathbb{I}[y f^*(\mathbf{x}) \leq 1]]$$

$$\leq \frac{3}{d^\tau} + \frac{4}{d^{0.9\tau - 1}\sqrt{\tau \log d}}. \tag{319}$$

$\square$

**Theorem 4.4** (Mixtures of Gaussians: Main Result). *Assume Assumption 4.1. For any $\epsilon, \delta \in (0, 1)$, when Algorithm 1 uses hinge loss with*

$$m = \text{poly}\left( \frac{1}{\delta}, \frac{1}{\epsilon}, d^{\Theta\left(\tau \sigma_{B+}^2 / B_{\mu 1}^2\right)}, r, \frac{1}{p_B} \right) \leq e^d, \quad T = \text{poly}(m), \quad n = \text{poly}(m)$$

*and proper hyper-parameters, then with probability at least $1 - \delta$, there exists $t \in [T]$ such that*

$$\Pr[\text{sign}(f_{\Xi^{(t)}}(\mathbf{x})) \neq y] \leq \frac{\sqrt{2}r}{d^{0.4\tau-0.8}} + \epsilon.$$

*Proof of Theorem 4.4.* Let $\tilde{b} = C_b\sqrt{\tau d \log d}\sigma_w\sigma_{B+}$, where $C_b$ is a large enough universal constant. By Lemma E.11, we have $f^* \in \mathcal{F}_{d,r,B_F,S_{p,\gamma,B_G}}$, where $B_F = (B_{a1}, B_{a2}, B_b) = \left(\frac{1}{\sqrt{\tau \log d}\sigma_{B+}}, \frac{\sqrt{r}}{\sqrt{\tau \log d}\sigma_{B+}}, 2\sqrt{\tau \log d}\sigma_{B+}\right)$, $p = \Theta\left(\frac{B_{\mu1}}{\sqrt{\tau \log d}\sigma_{B+}\cdot d^{\left(9C_b^2\tau\sigma_{B+}^2/(2B_{\mu1}^2)\right)}}\right)$, $\gamma = \frac{1}{d^{0.9\tau-1.5}}$, $B_G = p_B B_{\mu1}\sqrt{d} - O\left(\frac{\sigma_{B+}}{d^{0.9\tau}}\right)$ and $B_{x1} = (B_{\mu2} + \sigma_{B+})\sqrt{d}, B_{x2} = (B_{\mu2} + \sigma_{B+})^2 d$. We also have $\text{OPT}_{d,r,B_F,S_{p,\gamma,B_G}} \leq \frac{3}{d^\tau} + \frac{4}{d^{0.9\tau-1}\sqrt{\tau \log d}}$.

Adjust $\sigma_w$ such that $\tilde{b} = C_b\sqrt{\tau d \log d}\sigma_w\sigma_{B+} = \Theta\left(\frac{B_G^{\frac{1}{4}}B_{a2}B_b^{\frac{3}{4}}}{\sqrt{r}B_{a1}}\right)$. Injecting above parameters into Theorem 3.12, we have with probability at least $1 - \delta$ over the initialization, with proper hyper-parameters, there exists $t \in [T]$ such that

$$\Pr[\text{sign}(f_{\Xi^{(t)}}(\mathbf{x})) \neq y] \tag{320}$$

$$\leq \frac{3}{d^\tau} + \frac{4}{d^{0.9\tau-1}\sqrt{\tau \log d}} + \frac{\sqrt{2}rB_{\mu2}}{d^{(0.9\tau-1.5)/2}\sqrt{\tau \log d}\sigma_{B+}} + O\left(\frac{rB_{a1}B_{x1}B_{x2}^{\frac{1}{4}}(\log n)^{\frac{1}{4}}}{\sqrt{B_G}n^{\frac{1}{4}}}\right) + \epsilon/2$$

$$\leq \frac{\sqrt{2}r}{d^{0.4\tau-0.8}} + \epsilon. \tag{321}$$

$\square$

## E.3 Mixture of Gaussians - XOR

We consider a special Mixture of Gaussians distribution studied in [99]. Consider the same data distribution in Appendix E.2.1 and Definition E.6 with the following assumptions.

**Assumption E.12** (Mixture of Gaussians in [99]). *Assume four Gaussians cluster with XOR-like pattern, for any $\tau > 0$,*

- *$r = 4$ and $p_1 = p_2 = p_3 = p_4 = \frac{1}{4}$.*
- *$\mu_1 = -\mu_2$, $\mu_3 = -\mu_4$ and $\|\mu_1\|_2 = \|\mu_2\|_2 = \|\mu_3\|_2 = \|\mu_4\|_2 = \sqrt{d}$ and $\langle \mu_1, \mu_3 \rangle = 0$.*
- *For all $j \in [4]$, $\Sigma_j = \sigma_B I_{d\times d}$ and $1 \leq \sigma_B \leq \sqrt{\frac{d}{\tau \log \log d}}$.*
- *$y_{(1)} = y_{(2)} = 1$ and $y_{(3)} = y_{(4)} = -1$.*

We denote this data distribution as $\mathcal{D}_{mixture-xor}$ setting.

### E.3.1 Mixture of Gaussians - XOR: Feature Learning

**Lemma E.13** (Mixture of Gaussians in [99]: Gradient Feature Set). *Let $C_{Sure,1} = \frac{6}{5}$, $C_{Sure,2} = \frac{\sqrt{2}}{\sqrt{\tau \log \log d}}$, $\tilde{b} = \sqrt{\tau d \log \log d}\sigma_w\sigma_B$ and $d$ is large enough. For $\mathcal{D}_{mixture-xor}$ setting, we have $(D_j, +1) \in S_{p,\gamma,B_G}$ for all $j \in [4]$, where*

$$p = \Theta\left(\frac{1}{\sqrt{\tau \log \log d}\sigma_B \cdot (\log d)^{\frac{18\tau\sigma_B^2}{25}}}\right), \quad \gamma = \frac{\sigma_B}{\sqrt{d}}, \tag{322}$$

$$B_G = \frac{\sqrt{d}}{4}\left(1 - O\left(\frac{1}{(\log d)^{\frac{\tau}{50}}}\right)\right) - \sigma_B O\left(\frac{1}{(\log d)^{0.018\tau}}\right). \tag{323}$$

*Proof of Lemma E.13.* For all $j \in [r]$, by Lemma E.16, for all $i \in S_{D_j, Sure}$,

$$1 - \frac{\left|\left\langle G(\mathbf{w}_i^{(0)}, \mathbf{b}_i), D_j\right\rangle\right|}{\|G(\mathbf{w}_i^{(0)}, \mathbf{b}_i)\|_2} \tag{324}$$

$$\leq 1 - \frac{\left|\left\langle G(\mathbf{w}_i^{(0)}, \mathbf{b}_i), D_j\right\rangle\right|}{\sqrt{\left|\left\langle G(\mathbf{w}_i^{(0)}, \mathbf{b}_i), D_j\right\rangle\right|^2 + \max_{D_j^\top D_j^\perp = 0, \|D_j^\perp\|_2 = 1}\left|\left\langle G(\mathbf{w}_i^{(0)}, \mathbf{b}_i), D_j^\perp\right\rangle\right|^2}} \tag{325}$$

$$\leq 1 - \frac{\left|\left\langle G(\mathbf{w}_i^{(0)}, \mathbf{b}_i), D_j\right\rangle\right|}{\left|\left\langle G(\mathbf{w}_i^{(0)}, \mathbf{b}_i), D_j\right\rangle\right| + \max_{D_j^\top D_j^\perp = 0, \|D_j^\perp\|_2 = 1}\left|\left\langle G(\mathbf{w}_i^{(0)}, \mathbf{b}_i), D_j^\perp\right\rangle\right|} \tag{326}$$

$$\leq 1 - \frac{1}{1 + \frac{\sigma_B O\left(\frac{1}{(\log d)^{0.018\tau}}\right)}{\frac{1}{4}\sqrt{d}\left(1 - O\left(\frac{1}{(\log d)^{\frac{\tau}{50}}}\right)\right) - \sigma_B O\left(\frac{1}{(\log d)^{0.018\tau}}\right)}} \tag{327}$$

$$\leq \frac{\sigma_B O\left(\frac{1}{(\log d)^{0.018\tau}}\right)}{\frac{1}{4}\sqrt{d}\left(1 - O\left(\frac{1}{(\log d)^{\frac{\tau}{50}}}\right)\right) - \sigma_B O\left(\frac{1}{(\log d)^{0.018\tau}}\right)} \tag{328}$$

$$< \frac{\sigma_B}{\sqrt{d}} = \gamma. \tag{329}$$

Thus, we have $G(\mathbf{w}_i^{(0)}, \mathbf{b}_i) \in \mathcal{C}_{D_j, \gamma}$ and $\left|\left\langle G(\mathbf{w}_i^{(0)}, \mathbf{b}_i), D_j\right\rangle\right| \leq \|G(\mathbf{w}_i^{(0)}, \mathbf{b}_i)\|_2 \leq B_{x1}$, $\frac{\mathbf{b}_i}{|\mathbf{b}_i|} = +1$. Thus, by Lemma E.14, we have

$$\Pr_{\mathbf{w}, b}\left[G(\mathbf{w}, b) \in \mathcal{C}_{D_j, \gamma} \text{ and } \|G(\mathbf{w}, b)\|_2 \geq B_G \text{ and } \frac{b}{|b|} = +1\right] \tag{330}$$

$$\geq \Pr\left[i \in S_{D_j, Sure}\right] \tag{331}$$

$$\geq p. \tag{332}$$

Thus, $(D_j, +1) \in S_{p, \gamma, B_G}$. We finish the proof. $\square$

**Lemma E.14** (Mixture of Gaussians in [99]: Geometry at Initialization). *Assume the same conditions as in Lemma E.13. Recall for all $i \in [m]$, $\mathbf{w}_i^{(0)} \sim \mathcal{N}(0, \sigma_w^2 I_{d \times d})$, over the random initialization, we have for all $i \in [m], j \in [4]$,*

$$\Pr\left[i \in S_{D_j, Sure}\right] \geq \Theta\left(\frac{1}{\sqrt{\tau \log \log d}\sigma_B \cdot (\log d)^{\frac{18\tau\sigma_B^2}{25}}}\right). \tag{333}$$

*Proof of Lemma E.14.* WLOG, let $j = 1$. By Assumption E.12, for the first condition in Definition E.6, we have,

$$\Pr\left[\left\langle \mathbf{w}_i^{(0)}, \mu_1\right\rangle \geq C_{Sure,1} \mathbf{b}_i\right] \geq \Theta\left(\frac{1}{\sqrt{\tau \log \log d}\sigma_B \cdot (\log d)^{\frac{18\tau\sigma_B^2}{25}}}\right), \tag{334}$$

where the the last inequality follows Lemma F.6.

For the second condition in Definition E.6, by Lemma F.6, we have,

$$\Pr\left[\left|\left\langle \mathbf{w}_i^{(0)}, \mu_2\right\rangle\right| \leq C_{Sure,2} \mathbf{b}_i\right] \geq 1 - \frac{1}{2\sqrt{\pi}}\frac{1}{\sigma_B \cdot e^{\sigma_B^2}}, \tag{335}$$

On the other hand, if $X$ is a $\chi^2(k)$ random variable. Then we have

$$\Pr(X \geq k + 2\sqrt{kx} + 2x) \leq e^{-x}. \tag{336}$$

Therefore, we have

$$\Pr\left(\frac{1}{\sigma_w^2}\left\|\mathbf{w}_i^{(0)}\right\|_2^2 \geq d + 2\sqrt{\left(\frac{18\tau\sigma_B^2}{25}+2\right)d\log\log d} + 2\left(\frac{18\tau\sigma_B^2}{25}+2\right)\log\log d\right) \quad (337)$$

$$\leq O\left(\frac{1}{(\log d)^2 \cdot (\log d)^{\frac{18\tau\sigma_B^2}{25}}}\right). \quad (338)$$

Thus, by union bound, we have

$$\Pr\left[i \in S_{D_j,Sure}\right] \geq \Theta\left(\frac{1}{\sqrt{\tau\log\log d}\sigma_B \cdot (\log d)^{\frac{18\tau\sigma_B^2}{25}}}\right). \quad (339)$$

$\square$

**Lemma E.15** (Mixture of Gaussians in [99]: Activation Pattern). *Assume the same conditions as in Lemma E.13, for all $j \in [4], i \in S_{D_j,Sure}$, we have*

*(1) When $\mathbf{x} \sim \mathcal{N}_j(\mu_j, \sigma_B I_{d\times d})$, the activation probability satisfies,*

$$\Pr_{\mathbf{x}\sim\mathcal{N}_j(\mu_j,\sigma_B I_{d\times d})}\left[\left\langle\mathbf{w}_i^{(0)},\mathbf{x}\right\rangle - \mathbf{b}_i \geq 0\right] \geq 1 - \frac{1}{(\log d)^{\frac{\tau}{50}}}. \quad (340)$$

*(2) For all $j' \neq j, j' \in [4]$, when $\mathbf{x} \sim \mathcal{N}_{j'}(\mu_{j'}, \sigma_B I_{d\times d})$, the activation probability satisfies,*

$$\Pr_{\mathbf{x}\sim\mathcal{N}_{j'}(\mu_{j'},\sigma_B I_{d\times d})}\left[\left\langle\mathbf{w}_i^{(0)},\mathbf{x}\right\rangle - \mathbf{b}_i \geq 0\right] \leq O\left(\frac{1}{(\log d)^{\frac{\tau}{2}}}\right). \quad (341)$$

*Proof of Lemma E.15.* In the proof, we need $\tilde{b} = \sqrt{\tau d\log\log d}\sigma_w\sigma_B$. For the first statement, when $\mathbf{x} \sim \mathcal{N}_j(\mu_j, \sigma_B I_{d\times d})$, by $C_{Sure,1} \geq \frac{6}{5}$, we have

$$\Pr_{\mathbf{x}\sim\mathcal{N}_j(\mu_j,\sigma_B I_{d\times d})}\left[\left\langle\mathbf{w}_i^{(0)},\mathbf{x}\right\rangle - \mathbf{b}_i \geq 0\right] \geq \Pr_{\mathbf{x}\sim\mathcal{N}(0,\sigma_B I_{d\times d})}\left[\left\langle\mathbf{w}_i^{(0)},\mathbf{x}\right\rangle \geq (1 - C_{Sure,1})\mathbf{b}_i\right] \quad (342)$$

$$\geq \Pr_{\mathbf{x}\sim\mathcal{N}(0,\sigma_B I_{d\times d})}\left[\left\langle\mathbf{w}_i^{(0)},\mathbf{x}\right\rangle \geq -\frac{\mathbf{b}_i}{5}\right] \quad (343)$$

$$= 1 - \Pr_{\mathbf{x}\sim\mathcal{N}(0,\sigma_B I_{d\times d})}\left[\left\langle\mathbf{w}_i^{(0)},\mathbf{x}\right\rangle \leq -\frac{\mathbf{b}_i}{5}\right] \quad (344)$$

$$\geq 1 - \exp\left(-\frac{\mathbf{b}_i^2}{50d\sigma_w^2\sigma_B^2}\right) \quad (345)$$

$$\geq 1 - \frac{1}{(\log d)^{\frac{\tau}{50}}}, \quad (346)$$

where the third inequality follows the Chernoff bound and symmetricity of the Gaussian vector.

For the second statement, we prove similarly by $0 < C_{Sure,2} \leq \frac{\sqrt{2}}{\sqrt{\tau\log\log d}}$. $\square$

Then, Lemma E.16 gives gradients of neurons in $S_{D_j,Sure}$. It shows that these gradients are highly aligned with $D_j$.

**Lemma E.16** (Mixture of Gaussians in [99]: Feature Emergence). *Assume the same conditions as in Lemma E.13, for all $j \in [4]$, $i \in S_{D_j,Sure}$, we have*

$$\left\langle\mathbb{E}_{(\mathbf{x},y)}\left[y\sigma'\left(\left\langle\mathbf{w}_i^{(0)},\mathbf{x}\right\rangle - \mathbf{b}_i\right)\mathbf{x}\right], y_{(j)}D_j\right\rangle \quad (347)$$

$$\geq \frac{1}{4}\sqrt{d}\left(1 - O\left(\frac{1}{(\log d)^{\frac{\tau}{50}}}\right)\right) - \sigma_B O\left(\frac{1}{(\log d)^{0.018\tau}}\right). \quad (348)$$

*For any unit vector $D_j^\perp$ which is orthogonal with $D_j$, we have*

$$\left|\left\langle\mathbb{E}_{(\mathbf{x},y)}\left[y\sigma'\left(\left\langle\mathbf{w}_i^{(0)},\mathbf{x}\right\rangle - \mathbf{b}_i\right)\mathbf{x}\right], D_j^\perp\right\rangle\right| \leq \sigma_B O\left(\frac{1}{(\log d)^{0.018\tau}}\right). \quad (349)$$

*Proof of Lemma E.16.* For all $j \in [4]$, $i \in S_{D_j, Sure}$, we have

$$\mathbb{E}_{(\mathbf{x},y)} \left[ y\sigma' \left( \left\langle \mathbf{w}_i^{(0)}, \mathbf{x} \right\rangle - \mathbf{b}_i \right) \mathbf{x} \right] \tag{350}$$

$$= \sum_{l \in [4]} \frac{1}{4} \mathbb{E}_{\mathbf{x} \sim \mathcal{N}_l(\mathbf{x})} \left[ y\sigma' \left( \left\langle \mathbf{w}_i^{(0)}, \mathbf{x} \right\rangle - \mathbf{b}_i \right) \mathbf{x} \right] \tag{351}$$

$$= \sum_{l \in [4]} \frac{1}{4} y_{(l)} \mathbb{E}_{\mathbf{x} \sim \mathcal{N}(0, \sigma_l I_{d \times d})} \left[ \sigma' \left( \left\langle \mathbf{w}_i^{(0)}, \mathbf{x} + \mu_l \right\rangle - \mathbf{b}_i \right) (\mathbf{x} + \mu_l) \right]. \tag{352}$$

Thus, by Lemma F.7 and Lemma E.15,

$$\left\langle \mathbb{E}_{(\mathbf{x},y)} \left[ y\sigma' \left( \left\langle \mathbf{w}_i^{(0)}, \mathbf{x} \right\rangle - \mathbf{b}_i \right) \mathbf{x} \right], y_{(j)} D_j \right\rangle \tag{353}$$

$$= \frac{1}{4} \mathbb{E}_{\mathbf{x} \sim \mathcal{N}(0, \sigma_B I_{d \times d})} \left[ \sigma' \left( \left\langle \mathbf{w}_i^{(0)}, \mathbf{x} + \mu_j \right\rangle - \mathbf{b}_i \right) (\mathbf{x} + \mu_j)^\top D_j \right] \tag{354}$$

$$+ \sum_{l \in [4], l \neq j} \frac{1}{4} y_{(l)} y_{(j)} \mathbb{E}_{\mathbf{x} \sim \mathcal{N}(0, \sigma_l I_{d \times d})} \left[ \sigma' \left( \left\langle \mathbf{w}_i^{(0)}, \mathbf{x} + \mu_l \right\rangle - \mathbf{b}_i \right) (\mathbf{x} + \mu_l)^\top D_j \right] \tag{355}$$

$$\geq \frac{1}{4} \mu_j^\top D_j \left( 1 - O \left( \frac{1}{(\log d)^{\frac{\tau}{50}}} \right) \right) - \sum_{l \in [4], l \neq j} \frac{1}{4} |\mu_l^\top D_j| O \left( \frac{1}{d^{\frac{\tau}{2}}} \right) \tag{356}$$

$$- \frac{1}{4} \left| \mathbb{E}_{\mathbf{x} \sim \mathcal{N}(0, \sigma_B I)} \left[ \sigma' \left( \left\langle \mathbf{w}_i^{(0)}, \mathbf{x} + \mu_j \right\rangle - \mathbf{b}_i \right) \mathbf{x}^\top D_j \right] \right| \tag{357}$$

$$- \sum_{l \in [4], l \neq j} \frac{1}{4} \left| \mathbb{E}_{\mathbf{x} \sim \mathcal{N}(0, \sigma_l I)} \left[ \sigma' \left( \left\langle \mathbf{w}_i^{(0)}, \mathbf{x} + \mu_l \right\rangle - \mathbf{b}_i \right) \mathbf{x}^\top D_j \right] \right| \tag{358}$$

$$\geq \frac{1}{4} \sqrt{d} \left( 1 - O \left( \frac{1}{(\log d)^{\frac{\tau}{50}}} \right) \right) - \frac{1}{4} \left| \mathbb{E}_{\mathbf{x} \sim \mathcal{N}(0, \sigma_B I)} \left[ \left( 1 - \sigma' \left( \left\langle \mathbf{w}_i^{(0)}, \mathbf{x} + \mu_j \right\rangle - \mathbf{b}_i \right) - 1 \right) \mathbf{x}^\top D_j \right] \right|$$

$$- \sum_{l \in [4], l \neq j} \frac{1}{4} \left| \mathbb{E}_{\mathbf{x} \sim \mathcal{N}(0, \sigma_l I)} \left[ \sigma' \left( \left\langle \mathbf{w}_i^{(0)}, \mathbf{x} + \mu_l \right\rangle - \mathbf{b}_i \right) \mathbf{x}^\top D_j \right] \right| \tag{359}$$

$$= \frac{1}{4} \sqrt{d} \left( 1 - O \left( \frac{1}{(\log d)^{\frac{\tau}{50}}} \right) \right) - \frac{1}{4} \left| \mathbb{E}_{\mathbf{x} \sim \mathcal{N}(0, \sigma_B I)} \left[ \left( 1 - \sigma' \left( \left\langle \mathbf{w}_i^{(0)}, \mathbf{x} + \mu_j \right\rangle - \mathbf{b}_i \right) \right) \mathbf{x}^\top D_j \right] \right|$$

$$- \sum_{l \in [4], l \neq j} \frac{1}{4} \left| \mathbb{E}_{\mathbf{x} \sim \mathcal{N}(0, \sigma_l I)} \left[ \sigma' \left( \left\langle \mathbf{w}_i^{(0)}, \mathbf{x} + \mu_l \right\rangle - \mathbf{b}_i \right) \mathbf{x}^\top D_j \right] \right| \tag{360}$$

$$\geq \frac{1}{4} \sqrt{d} \left( 1 - O \left( \frac{1}{(\log d)^{\frac{\tau}{50}}} \right) \right) - \sigma_B O \left( \frac{1}{(\log d)^{0.018\tau}} \right). \tag{361}$$

For any unit vector $D_j^\perp$ which is orthogonal with $D_j$, similarly, we have

$$\left| \left\langle \mathbb{E}_{(\mathbf{x},y)} \left[ y\sigma' \left( \left\langle \mathbf{w}_i^{(0)}, \mathbf{x} \right\rangle - \mathbf{b}_i \right) \mathbf{x} \right], D_j^\perp \right\rangle \right| \tag{362}$$

$$\leq \frac{1}{4} \left| \mathbb{E}_{\mathbf{x} \sim \mathcal{N}(0, \sigma_B I)} \left[ \sigma' \left( \left\langle \mathbf{w}_i^{(0)}, \mathbf{x} + \mu_j \right\rangle - \mathbf{b}_i \right) \mathbf{x}^\top D_j^\perp \right] \right| \tag{363}$$

$$+ \sum_{l \in [4], l \neq j} \frac{1}{4} \left| \mathbb{E}_{\mathbf{x} \sim \mathcal{N}(0, \sigma_l I)} \left[ \sigma' \left( \left\langle \mathbf{w}_i^{(0)}, \mathbf{x} + \mu_l \right\rangle - \mathbf{b}_i \right) (\mathbf{x} + \mu_l)^\top D_j^\perp \right] \right| \tag{364}$$

$$\leq \frac{1}{4} \left| \mathbb{E}_{\mathbf{x} \sim \mathcal{N}(0, \sigma_B I)} \left[ \sigma' \left( \left\langle \mathbf{w}_i^{(0)}, \mathbf{x} + \mu_j \right\rangle - \mathbf{b}_i \right) \mathbf{x}^\top D_j^\perp \right] \right| \tag{365}$$

$$+ \sum_{l \in [4], l \neq j} \frac{1}{4} \left| \mathbb{E}_{\mathbf{x} \sim \mathcal{N}(0, \sigma_l I)} \left[ \sigma' \left( \left\langle \mathbf{w}_i^{(0)}, \mathbf{x} + \mu_l \right\rangle - \mathbf{b}_i \right) \mathbf{x}^\top D_j^\perp \right] \right| \tag{366}$$

$$\leq \sigma_B O \left( \frac{1}{(\log d)^{0.018\tau}} \right). \tag{367}$$

$\square$

### E.3.2 Mixture of Gaussians - XOR: Final Guarantee

**Lemma E.17** (Mixture of Gaussians in [99]: Existence of Good Networks)**.** *Assume the same conditions as in Lemma E.13 and let $\tau = 1$ and when $0 < \tilde{\tau} \leq O\left(\frac{d}{\sigma_B^2 \log d}\right)$. Define*

$$f^*(\mathbf{x}) = \sum_{j=1}^{4} \frac{y_{(j)}}{\sqrt{\tilde{\tau} \log d} \sigma_B} \left[ \sigma \left( \langle D_j, \mathbf{x} \rangle - 2\sqrt{\tilde{\tau} \log d} \sigma_B \right) \right]. \tag{368}$$

*For $\mathcal{D}_{mixture-xor}$ setting, we have $f^* \in \mathcal{F}_{d,r,B_F,S_p,\gamma,B_G}$, where $B_F = (B_{a1}, B_{a2}, B_b) = \left( \frac{1}{\sqrt{\tilde{\tau} \log d} \sigma_B}, \frac{2}{\sqrt{\tilde{\tau} \log d} \sigma_B}, 2\sqrt{\tilde{\tau} \log d} \sigma_B \right)$, $p = \Omega\left( \frac{1}{\sigma_B \cdot (\log d)^{\sigma_B^2}} \right)$, $\gamma = \frac{\sigma_B}{\sqrt{d}}$, $r = 4$, $B_G = \frac{1}{5}\sqrt{d}$ and $B_{x1} = (1 + \sigma_B)\sqrt{d}, B_{x2} = (1 + \sigma_B)^2 d$. We also have $\mathrm{OPT}_{d,r,B_F,S_p,\gamma,B_G} \leq \frac{3}{d^{\tilde{\tau}}} + \frac{4}{d^{0.9\tilde{\tau}-1}\sqrt{\tilde{\tau} \log d}}$.*

*Proof of Lemma E.17.* We finish the proof by following the proof of Lemma E.11 □

**Theorem E.18** (Mixture of Gaussians in [99]: Main Result)**.** *For $\mathcal{D}_{mixture-xor}$ setting with Assumption E.12, when $d$ is large enough, for any $\delta \in (0, 1)$ and for any $\epsilon \in (0, 1)$ when*

$$m = \Omega\left( \sigma_B (\log d)^{\sigma_B^2} \left( \left( \log\left(\frac{1}{\delta}\right) \right)^2 + \frac{1 + \sigma_B}{\epsilon^4} \right) + \frac{1}{\sqrt{\delta}} \right) \leq e^d, \tag{369}$$

$$T = \mathrm{poly}(\sigma_B, 1/\epsilon, 1/\delta, \log d), \tag{370}$$

$$n = \tilde{\Omega}\left( \frac{m^3(1 + \sigma_B^2)}{\epsilon^2 \max\left\{ \sigma_B \cdot (\log d)^{\sigma_B^2}, 1 \right\}} + \sigma_B \cdot (\log d)^{\sigma_B^2} + \frac{Tm}{\delta} \right), \tag{371}$$

*trained by Algorithm 1 with hinge loss, with probability at least $1 - \delta$ over the initialization and training samples, with proper hyper-parameters, there exists $t \in [T]$ such that*

$$\Pr[\mathrm{sign}(f_{\Xi^{(t)}}(\mathbf{x})) \neq y] \leq O\left( \left(1 + \sigma_B^{\frac{3}{2}}\right) \left( \frac{1}{d^{\frac{1}{4}}} + \frac{(\log n)^{\frac{1}{4}}}{n^{\frac{1}{4}}} \right) \right) + \epsilon. \tag{372}$$

*Proof of Theorem E.18.* Let $\tilde{b} = \sqrt{d \log \log d} \sigma_w \sigma_B$. By Lemma E.17, let $\tau = 1$ and when $\tilde{\tau} = O\left(\frac{d}{\sigma_B^2 \log d}\right)$, we have $f^* \in \mathcal{F}_{d,r,B_F,S_p,\gamma,B_G}$, where $B_F = (B_{a1}, B_{a2}, B_b) = \left( \frac{1}{\sqrt{\tilde{\tau} \log d} \sigma_B}, \frac{2}{\sqrt{\tilde{\tau} \log d} \sigma_B}, 2\sqrt{\tilde{\tau} \log d} \sigma_B \right)$, $p = \Omega\left( \frac{1}{\sigma_B \cdot (\log d)^{\sigma_B^2}} \right)$, $\gamma = \frac{\sigma_B}{\sqrt{d}}$, $r = 4$, $B_G = \frac{1}{5}\sqrt{d}$ and $B_{x1} = (1 + \sigma_B)\sqrt{d}, B_{x2} = (1 + \sigma_B)^2 d$. We also have $\mathrm{OPT}_{d,r,B_F,S_p,\gamma,B_G} \leq \frac{3}{d^{\tilde{\tau}}} + \frac{4}{d^{0.9\tilde{\tau}-1}\sqrt{\tilde{\tau} \log d}}$.

Adjust $\sigma_w$ such that $\tilde{b} = \sqrt{d \log \log d} \sigma_w \sigma_B = \Theta\left( \frac{B_G^{\frac{1}{4}} B_{a2} B_b^{\frac{3}{4}}}{\sqrt{r} B_{a1}} \right)$. Injecting above parameters into Theorem 3.12, we have with probability at least $1 - \delta$ over the initialization, with proper hyper-parameters, there exists $t \in [T]$ such that

$$\Pr[\mathrm{sign}(f_{\Xi^{(t)}}(\mathbf{x})) \neq y] \leq O\left( \left(1 + \sigma_B^{\frac{3}{2}}\right) \left( \frac{1}{d^{\frac{1}{4}}} + \frac{(\log n)^{\frac{1}{4}}}{n^{\frac{1}{4}}} \right) \right) + \epsilon. \tag{373}$$

□

### E.4 Parity Functions

We recap the problem setup in Section 4.2 for readers' convenience.

#### E.4.1 Problem Setup

**Data Distributions.** Suppose $\mathbf{M} \in \mathbb{R}^{d \times D}$ is an unknown dictionary with $D$ columns that can be regarded as patterns. For simplicity, assume $d = D$ and $\mathbf{M}$ is orthonormal. Let $\phi \in \mathbb{R}^d$ be a hidden representation vector. Let $A \subseteq [D]$ be a subset of size $rk$ corresponding to the class relevant patterns

and $r$ is an odd number. Then the input is generated by $\mathbf{M}\phi$, and some function on $\phi_A$ generates the label. WLOG, let $A = \{1, \ldots, rk\}$, $A^\perp = \{rk + 1, \ldots, d\}$. Also, we split $A$ such that for all $j \in [r]$, $A_j = \{(j-1)k + 1, \ldots, jk\}$. Then the input $\mathbf{x}$ and the class label $y$ are given by:

$$\mathbf{x} = \mathbf{M}\phi, \; y = g^*(\phi_A) = \text{sign}\left(\sum_{j=1}^{r} \text{XOR}(\phi_{A_j})\right), \tag{374}$$

where $g^*$ is the ground-truth labeling function mapping from $\mathbb{R}^{rk}$ to $\mathcal{Y} = \{\pm 1\}$, $\phi_A$ is the sub-vector of $\phi$ with indices in $A$, and $\text{XOR}(\phi_{A_j}) = \prod_{l \in A_j} \phi_l$ is the parity function.

We still need to specify the distribution of $\phi$, which determines the structure of the input distribution:

$$\mathcal{X} := (1 - 2rp_A)\mathcal{X}_U + \sum_{j \in [r]} p_A(\mathcal{X}_{j,+} + \mathcal{X}_{j,-}). \tag{375}$$

For all corresponding $\phi_{A^\perp}$ in $\mathcal{X}$, we have $\forall l \in A^\perp$, independently:

$$\phi_l = \begin{cases} +1, & \text{w.p. } p_o \\ -1, & \text{w.p. } p_o \\ 0, & \text{w.p. } 1 - 2p_o \end{cases}$$

where $p_o$ controls the signal noise ratio: if $p_o$ is large, then there are many nonzero entries in $A^\perp$ which are noise interfering with the learning of the ground-truth labeling function on $A$.

For corresponding $\phi_A$, any $j \in [r]$, we have

- In $\mathcal{X}_{j,+}$, $\phi_{A_j} = [+1, +1, \ldots, +1]^\top$ and $\phi_{A \setminus A_j}$ only have zero elements.
- In $\mathcal{X}_{j,-}$, $\phi_{A_j} = [-1, -1, \ldots, -1]^\top$ and $\phi_{A \setminus A_j}$ only have zero elements.
- In $\mathcal{X}_U$, we have $\phi_A$ draw from $\{+1, -1\}^{rk}$ uniformly.

We call this data distribution $\mathcal{D}_{parity}$.

**Assumption E.19** (Parity Functions. Recap of Assumption 4.5). *Let $8 \leq \tau \leq d$ be a parameter that will control our final error guarantee. Assume $k$ is an odd number and:*

$$k \geq \Omega(\tau \log d), \quad d \geq rk + \Omega(\tau r \log d), \quad p_o = O\left(\frac{rk}{d - rk}\right), \quad p_A \geq \frac{1}{d}. \tag{376}$$

**Remark E.20.** *The assumptions require $k, d$, and $p_A$ to be sufficiently large so as to provide enough large signals for learning. When $p_o = \Theta(\frac{rk}{d-rk})$ means that the signal noise ratio is constant: the expected norm of $\phi_A$ and that of $\phi_{A^\perp}$ are comparable.*

To apply our framework, again we only need to compute the parameters in the Gradient Feature set and the corresponding optimal approximation loss. To this end, we first define the gradient features: For all $j \in [r]$, let

$$D_j = \frac{\sum_{l \in A_j} \mathbf{M}_l}{\|\sum_{l \in A_j} \mathbf{M}_l\|_2}. \tag{377}$$

**Remark E.21.** *Our data distribution is symmetric, which means for any $\phi \in \mathbb{R}^d$:*

- $-y = g^*(-\phi_A)$ and $-x = \mathbf{M}(-\phi)$,
- $\mathbb{P}(\phi) = \mathbb{P}(-\phi)$,
- $\mathbb{E}_{(\mathbf{x},y)}[y\mathbf{x}] = \mathbf{0}$.

Below, we define a sufficient condition that randomly initialized weights will fall in nice gradients set after the first gradient step update.

**Definition E.22** (Parity Functions: Subset of Nice Gradients Set). *Recall $\mathbf{w}_i^{(0)}$ is the weight for the $i$-th neuron at initialization. For all $j \in [r]$, let $S_{D_j, Sure} \subseteq [m]$ be those neurons that satisfy*

- $\left\langle \mathbf{w}_i^{(0)}, D_j \right\rangle \geq \frac{C_{Sure,1}}{\sqrt{k}} \mathbf{b}_i,$

- $\left|\left\langle \mathbf{w}_i^{(0)}, D_{j'} \right\rangle\right| \leq \frac{C_{Sure,2}}{\sqrt{k}} \mathbf{b}_i$, *for all* $j' \neq j, j' \in [r]$,
- $\left\|P_A \mathbf{w}_i^{(0)}\right\|_2 \leq \Theta(\sqrt{rk}\sigma_w)$,
- $\left\|P_{A^\perp} \mathbf{w}_i^{(0)}\right\|_2 \leq \Theta(\sqrt{d - rk}\sigma_w)$,

*where* $P_A, P_{A^\perp}$ *are the projection operator on the space* $\mathbf{M}_A$ *and* $\mathbf{M}_{A^\perp}$.

### E.4.2 Parity Functions: Feature Learning

We show the important Lemma E.23 first and defer other Lemmas after it.

**Lemma E.23** (Parity Functions: Gradient Feature Set. Part statement of Lemma 4.7). *Let* $C_{Sure,1} = \frac{3}{2}$, $C_{Sure,2} = \frac{1}{2}$, $\tilde{b} = C_b\sqrt{\tau rk \log d}\sigma_w$, *where* $C_b$ *is a large enough universal constant. For* $\mathcal{D}_{parity}$ *setting, we have* $(D_j, +1), (D_j, -1) \in S_{p,\gamma,B_G}$ *for all* $j \in [r]$, *where*

$$p = \Theta\left(\frac{1}{\sqrt{\tau r \log d} \cdot d^{(9C_b^2 \tau r/8)}}\right), \quad \gamma = \frac{1}{d^{\tau-2}}, \quad B_G = \sqrt{k}p_A - O\left(\frac{\sqrt{k}}{d^\tau}\right). \tag{378}$$

*Proof of Lemma E.23.* Note that for all $l \in [d]$, we have $\mathbf{M}_l^\top \mathbf{x} = \phi_l$. For all $j \in [r]$, by Lemma E.26, for all $i \in S_{D_j, Sure}$, when $\gamma = \frac{1}{d^{\tau-2}}$,

$$\left|\left\langle G(\mathbf{w}_i^{(0)}, \mathbf{b}_i), D_j \right\rangle\right| - (1-\gamma)\|G(\mathbf{w}_i^{(0)}, \mathbf{b}_i)\|_2 \tag{379}$$

$$= \left|\left\langle G(\mathbf{w}_i^{(0)}, \mathbf{b}_i), \frac{\sum_{l \in A_j} \mathbf{M}_l}{\sqrt{k}} \right\rangle\right| - (1-\gamma)\|G(\mathbf{w}_i^{(0)}, \mathbf{b}_i)\|_2 \tag{380}$$

$$\geq \sqrt{k}p_A - O\left(\frac{\sqrt{k}}{d^\tau}\right) - \left(1 - \frac{1}{d^{\tau-2}}\right)\sqrt{kp_A^2 + \sum_{l \in [d]} O\left(\frac{1}{d^\tau}\right)^2} \tag{381}$$

$$\geq \sqrt{k}p_A - O\left(\frac{\sqrt{k}}{d^\tau}\right) - \left(1 - \frac{1}{d^{\tau-2}}\right)\left(\sqrt{k}p_A + O\left(\frac{1}{d^{\tau-\frac{1}{2}}}\right)\right) \tag{382}$$

$$\geq \frac{\sqrt{k}p_A}{d^{\tau-2}} - O\left(\frac{\sqrt{k}}{d^\tau}\right) - O\left(\frac{1}{d^{\tau-\frac{1}{2}}}\right) \tag{383}$$

$$> 0. \tag{384}$$

Thus, we have $G(\mathbf{w}_i^{(0)}, \mathbf{b}_i) \in \mathcal{C}_{D_j,\gamma}$ and $\sqrt{k}p_A - O\left(\frac{\sqrt{k}}{d^\tau}\right) \leq \|G(\mathbf{w}_i^{(0)}, \mathbf{b}_i)\|_2 \leq \sqrt{k}p_A + O\left(\frac{1}{d^{\tau-\frac{1}{2}}}\right)$, $\frac{\mathbf{b}_i}{|\mathbf{b}_i|} = +1$. Thus, by Lemma E.24, we have

$$\Pr_{\mathbf{w},b}\left[G(\mathbf{w}, b) \in \mathcal{C}_{D_j,\gamma} \text{ and } \|G(\mathbf{w}, b)\|_2 \geq B_G \text{ and } \frac{b}{|b|} = +1\right] \tag{385}$$

$$\geq \Pr\left[i \in S_{D_j, Sure}\right] \tag{386}$$

$$\geq p. \tag{387}$$

Thus, $(D_j, +1) \in S_{p,\gamma,B_G}$. Since $\mathbb{E}_{(\mathbf{x},y)}[y\mathbf{x}] = \mathbf{0}$, by Lemma F.2 and considering $i \in [2m] \setminus [m]$, we have $(D_j, -1) \in S_{p,\gamma,B_G}$. We finish the proof. $\square$

Below are Lemmas used in the proof of Lemma E.23. In Lemma E.24, we calculate $p$ used in $S_{p,\gamma,B_G}$.

**Lemma E.24** (Parity Functions: Geometry at Initialization. Lemma B.2 in [7]). *Assume the same conditions as in Lemma E.23, recall for all* $i \in [m]$, $\mathbf{w}_i^{(0)} \sim \mathcal{N}(0, \sigma_w^2 I_{d \times d})$, *over the random initialization, we have for all* $i \in [m], j \in [r]$,

$$\Pr\left[i \in S_{D_j, Sure}\right] \geq \Theta\left(\frac{1}{\sqrt{\tau r \log d} \cdot d^{(9C_b^2 \tau r/8)}}\right). \tag{388}$$

*Proof of Lemma E.24.* For every $i \in [m]$, $j, j' \in [r]$, $j \neq j'$, by Lemma F.6,

$$p_1 = \Pr\left[\left\langle \mathbf{w}_i^{(0)}, D_j \right\rangle \geq \frac{C_{Sure,1}}{\sqrt{k}} \mathbf{b}_i \right] = \Theta\left(\frac{1}{\sqrt{\tau r \log d} \cdot d^{\left(9C_b^2 \tau r/8\right)}}\right) \tag{389}$$

$$p_2 = \Pr\left[\left|\left\langle \mathbf{w}_i^{(0)}, D_{j'} \right\rangle\right| \geq \frac{C_{Sure,2}}{\sqrt{k}} \mathbf{b}_i \right] = \Theta\left(\frac{1}{\sqrt{\tau r \log d} \cdot d^{\left(C_b^2 \tau r/8\right)}}\right). \tag{390}$$

On the other hand, if $X$ is a $\chi^2(k)$ random variable, by Lemma F.5, we have

$$\Pr(X \geq k + 2\sqrt{kx} + 2x) \leq e^{-x}. \tag{391}$$

Therefore, by assumption $rk \geq \Omega(\tau r \log d), d - rk \geq \Omega(\tau r \log d)$, we have

$$\Pr\left(\frac{1}{\sigma_w^2}\left\|P_A \mathbf{w}_i^{(0)}\right\|_2^2 \geq rk + 2\sqrt{(9C_b^2 \tau r/8 + 2)rk\log d} + 2\left(9C_b^2 \tau r/8 + 2\right)\log d\right) \tag{392}$$

$$\leq O\left(\frac{1}{d^2 \cdot d^{\left(9C_b^2 \tau r/8\right)}}\right), \tag{393}$$

$$\Pr\left(\frac{1}{\sigma_w^2}\left\|P_A \mathbf{w}_i^{(0)}\right\|_2^2 \geq (d - rk) + 2\sqrt{(9C_b^2 \tau r/8 + 2)(d-rk)\log d} + 2\left(9C_b^2 \tau r/8 + 2\right)\log d\right)$$

$$\leq O\left(\frac{1}{d^2 \cdot d^{\left(9C_b^2 \tau r/8\right)}}\right). \tag{394}$$

Thus, by union bound, and $D_1, \ldots, D_r$ being orthogonal with each other, we have

$$\Pr\left[i \in S_{D_j, Sure}\right] \geq p_1(1-p_2)^{r-1} - O\left(\frac{1}{d^2 \cdot d^{\left(9C_b^2 \tau r/8\right)}}\right) \tag{395}$$

$$= \Theta\left(\frac{1}{\sqrt{\tau r \log d} \cdot d^{\left(9C_b^2 \tau r/8\right)}} \cdot \left(1 - \frac{r}{\sqrt{\tau r \log d} \cdot d^{\left(C_b^2 \tau r/8\right)}}\right)\right) \tag{396}$$

$$\quad - O\left(\frac{1}{d^2 \cdot d^{\left(9C_b^2 \tau r/8\right)}}\right) \tag{397}$$

$$= \Theta\left(\frac{1}{\sqrt{\tau r \log d} \cdot d^{\left(9C_b^2 \tau r/8\right)}}\right). \tag{398}$$

$$\square$$

In Lemma E.25, we compute the activation pattern for the neurons in $S_{D_j, Sure}$.

**Lemma E.25** (Parity Functions: Activation Pattern). *Assume the same conditions as in Lemma E.23, for all $j \in [r], i \in S_{D_j, Sure}$, we have*

*(1) When $\mathbf{x} \sim \mathcal{X}$, we have*

$$\Pr_{\mathbf{x} \sim \mathcal{X}}\left[\left|\sum_{l \in A^\perp} \left\langle \mathbf{w}_i^{(0)}, \mathbf{M}_l \phi_l \right\rangle\right| \geq t\right] \leq \exp\left(-\frac{t^2}{\Theta\left(rk\sigma_w^2\right)}\right). \tag{399}$$

*(2) When $\mathbf{x} \sim \mathcal{X}_U$, we have*

$$\Pr_{\mathbf{x} \sim \mathcal{X}_U}\left[\left|\sum_{l \in A} \left\langle \mathbf{w}_i^{(0)}, \mathbf{M}_l \phi_l \right\rangle\right| \geq t\right] \leq \exp\left(-\frac{t^2}{\Theta(rk\sigma_w^2)}\right). \tag{400}$$

*(3) When $\mathbf{x} \sim \mathcal{X}_U$, the activation probability satisfies,*

$$\Pr_{\mathbf{x} \sim \mathcal{X}_U}\left[\sum_{l \in [d]} \left\langle \mathbf{w}_i^{(0)}, \mathbf{M}_l \phi_l \right\rangle - \mathbf{b}_i \geq 0\right] \leq O\left(\frac{1}{d^\tau}\right). \tag{401}$$

*(4) When $\mathbf{x} \sim \mathcal{X}_{j,+}$, the activation probability satisfies,*

$$\Pr_{\mathbf{x} \sim \mathcal{X}_{j,+}} \left[ \sum_{l \in [d]} \left\langle \mathbf{w}_i^{(0)}, \mathbf{M}_l \phi_l \right\rangle - \mathbf{b}_i \geq 0 \right] \geq 1 - O\left(\frac{1}{d^\tau}\right). \tag{402}$$

*(5) For all $j' \neq j, j' \in [r]$, $s \in \{+, -\}$, when $\mathbf{x} \sim \mathcal{X}_{j',s}$, or $\mathbf{x} \sim \mathcal{X}_{j,-}$, the activation probability satisfies,*

$$\Pr \left[ \sum_{l \in [d]} \left\langle \mathbf{w}_i^{(0)}, \mathbf{M}_l \phi_l \right\rangle - \mathbf{b}_i \geq 0 \right] \leq O\left(\frac{1}{d^\tau}\right). \tag{403}$$

*Proof of Lemma E.25.* For the first statement, when $\mathbf{x} \sim \mathcal{X}$, note that $\left\langle \mathbf{w}_i^{(0)}, \mathbf{M}_l \right\rangle \phi_l$ is a mean-zero sub-Gaussian random variable with sub-Gaussion norm $\Theta\left( \left| \left\langle \mathbf{w}_i^{(0)}, \mathbf{M}_l \right\rangle \right| \sqrt{p_o} \right)$.

$$\Pr_{\mathbf{x} \sim \mathcal{X}} \left[ \left| \sum_{l \in A^\perp} \left\langle \mathbf{w}_i^{(0)}, \mathbf{M}_l \phi_l \right\rangle \right| \geq t \right] = \Pr_{\mathbf{x} \sim \mathcal{X}} \left[ \left| \sum_{l \in A^\perp} \left\langle \mathbf{w}_i^{(0)}, \mathbf{M}_l \right\rangle \phi_l \right| \geq t \right] \tag{404}$$

$$\leq \exp\left( -\frac{t^2}{\sum_{l \in A^\perp} \Theta\left( \left\langle \mathbf{w}_i^{(0)}, \mathbf{M}_l \right\rangle^2 p_o \right)} \right) \tag{405}$$

$$\leq \exp\left( -\frac{t^2}{\Theta\left( (d - rk)\sigma_w^2 p_o \right)} \right) \tag{406}$$

$$\leq \exp\left( -\frac{t^2}{\Theta(rk\sigma_w^2)} \right), \tag{407}$$

where the inequality follows general Hoeffding's inequality.

For the second statement, when $\mathbf{x} \sim \mathcal{X}_U$, by Hoeffding's inequality,

$$\Pr_{\mathbf{x} \sim \mathcal{X}_U} \left[ \left| \sum_{l \in A} \left\langle \mathbf{w}_i^{(0)}, \mathbf{M}_l \phi_l \right\rangle \right| \geq t \right] = \Pr_{\mathbf{x} \sim \mathcal{X}_U} \left[ \left| \sum_{l \in A} \left\langle \mathbf{w}_i^{(0)}, \mathbf{M}_l \right\rangle \phi_l \right| \geq t \right] \tag{408}$$

$$\leq 2 \exp\left( -\frac{t^2}{2 \sum_{l \in A} \left\langle \mathbf{w}_i^{(0)}, \mathbf{M}_l \right\rangle^2} \right) \tag{409}$$

$$\leq \exp\left( -\frac{t^2}{\Theta(rk\sigma_w^2)} \right). \tag{410}$$

In the proof of the third to the last statement, we need $\tilde{b} = C_b\sqrt{\tau rk \log d}\sigma_w$, where $C_b$ is a large enough universal constant.

For the third statement, when $\mathbf{x} \sim \mathcal{X}_U$, by union bound and previous statements,

$$\Pr_{\mathbf{x} \sim \mathcal{X}_U} \left[ \sum_{l \in [d]} \left\langle \mathbf{w}_i^{(0)}, \mathbf{M}_l \phi_l \right\rangle - \mathbf{b}_i \geq 0 \right] \tag{411}$$

$$\leq \Pr_{\mathbf{x} \sim \mathcal{X}_U} \left[ \sum_{l \in A} \left\langle \mathbf{w}_i^{(0)}, \mathbf{M}_l \phi_l \right\rangle \geq \frac{\mathbf{b}_i}{2} \right] + \Pr_{\mathbf{x} \sim \mathcal{X}_U} \left[ \sum_{l \in A^\perp} \left\langle \mathbf{w}_i^{(0)}, \mathbf{M}_l \phi_l \right\rangle \geq \frac{\mathbf{b}_i}{2} \right] \tag{412}$$

$$\leq O\left(\frac{1}{d^\tau}\right). \tag{413}$$

For the forth statement, when $\mathbf{x} \sim \mathcal{X}_{j,+}$, by $C_{Sure,1} \geq \frac{3}{2}$ and previous statements,

$$\Pr_{\mathbf{x}\sim\mathcal{X}_{j,+}}\left[\sum_{l\in[d]}\left\langle\mathbf{w}_i^{(0)},\mathbf{M}_l\phi_l\right\rangle - \mathbf{b}_i \geq 0\right] \tag{414}$$

$$= \Pr_{\mathbf{x}\sim\mathcal{X}_{j,+}}\left[\sum_{l\in A_j}\left\langle\mathbf{w}_i^{(0)},\mathbf{M}_l\phi_l\right\rangle + \sum_{l\in A\setminus A_j}\left\langle\mathbf{w}_i^{(0)},\mathbf{M}_l\phi_l\right\rangle + \sum_{l\in A^\perp}\left\langle\mathbf{w}_i^{(0)},\mathbf{M}_l\phi_l\right\rangle \geq \mathbf{b}_i\right] \tag{415}$$

$$\geq \Pr_{\mathbf{x}\sim\mathcal{X}_{j,+}}\left[\sum_{l\in A^\perp}\left\langle\mathbf{w}_i^{(0)},\mathbf{M}_l\phi_l\right\rangle \geq (1-C_{Sure,1})\mathbf{b}_i\right] \tag{416}$$

$$\geq \Pr_{\mathbf{x}\sim\mathcal{X}_{j,+}}\left[\sum_{l\in A^\perp}\left\langle\mathbf{w}_i^{(0)},\mathbf{M}_l\phi_l\right\rangle \geq -\frac{\mathbf{b}_i}{2}\right] \tag{417}$$

$$\geq 1 - O\left(\frac{1}{d^\tau}\right). \tag{418}$$

For the last statement, we prove similarly by $0 < C_{Sure,2} \leq \frac{1}{2}$. $\qquad\square$

Then, Lemma E.26 gives gradients of neurons in $S_{D_j,Sure}$. It shows that these gradients are highly aligned with $D_j$.

**Lemma E.26** (Parity Functions: Feature Emergence). *Assume the same conditions as in Lemma E.23, for all $j \in [r]$, $i \in S_{D_j,Sure}$, we have the following holds:*

*(1) For all $l \in A_j$, we have*

$$p_A - O\left(\frac{1}{d^\tau}\right) \leq \mathbb{E}_{(\mathbf{x},y)}\left[y\sigma'\left(\left\langle\mathbf{w}_i^{(0)},\mathbf{x}\right\rangle - \mathbf{b}_i\right)\phi_l\right] \leq p_A + O\left(\frac{1}{d^\tau}\right). \tag{419}$$

*(2) For all $l \in A_{j'}$, any $j' \neq j, j' \in [r]$, we have*

$$\left|\mathbb{E}_{(\mathbf{x},y)}\left[y\sigma'\left(\left\langle\mathbf{w}_i^{(0)},\mathbf{x}\right\rangle - \mathbf{b}_i\right)\phi_l\right]\right| \leq O\left(\frac{1}{d^\tau}\right). \tag{420}$$

*(3) For all $l \in A^\perp$, we have*

$$\left|\mathbb{E}_{(\mathbf{x},y)}\left[y\sigma'\left(\left\langle\mathbf{w}_i^{(0)},\mathbf{x}\right\rangle - \mathbf{b}_i\right)\phi_l\right]\right| \leq O\left(\frac{1}{d^\tau}\right). \tag{421}$$

*Proof of Lemma E.26.* For all $l \in [d]$, we have

$$\mathbb{E}_{(\mathbf{x},y)}\left[y\sigma'\left(\left\langle\mathbf{w}_i^{(0)},\mathbf{x}\right\rangle - \mathbf{b}_i\right)\phi_l\right] \tag{422}$$

$$= p_A\sum_{l\in[r]}\left(\mathbb{E}_{\mathbf{x}\sim\mathcal{X}_{l,+}}\left[\sigma'\left(\left\langle\mathbf{w}_i^{(0)},\mathbf{x}\right\rangle - \mathbf{b}_i\right)\phi_l\right] - \mathbb{E}_{\mathbf{x}\sim\mathcal{X}_{l,-}}\left[\sigma'\left(\left\langle\mathbf{w}_i^{(0)},\mathbf{x}\right\rangle - \mathbf{b}_i\right)\phi_l\right]\right) \tag{423}$$

$$+ (1-2rp_A)\mathbb{E}_{\mathbf{x}\sim\mathcal{X}_U}\left[y\sigma'\left(\left\langle\mathbf{w}_i^{(0)},\mathbf{x}\right\rangle - \mathbf{b}_i\right)\phi_l\right]. \tag{424}$$

For the first statement, for all $l \in A_j$, by Lemma E.25 (3) and (4), we have

$$\mathbb{E}_{(\mathbf{x},y)}\left[y\sigma'\left(\left\langle\mathbf{w}_i^{(0)},\mathbf{x}\right\rangle - \mathbf{b}_i\right)\phi_l\right] \tag{425}$$

$$= p_A\left(\mathbb{E}_{\mathbf{x}\sim\mathcal{X}_{j,+}}\left[\sigma'\left(\left\langle\mathbf{w}_i^{(0)},\mathbf{x}\right\rangle - \mathbf{b}_i\right)\right] + \mathbb{E}_{\mathbf{x}\sim\mathcal{X}_{j,-}}\left[\sigma'\left(\left\langle\mathbf{w}_i^{(0)},\mathbf{x}\right\rangle - \mathbf{b}_i\right)\right]\right) \tag{426}$$

$$+ (1-2rp_A)\mathbb{E}_{\mathbf{x}\sim\mathcal{X}_U}\left[y\sigma'\left(\left\langle\mathbf{w}_i^{(0)},\mathbf{x}\right\rangle - \mathbf{b}_i\right)\phi_l\right] \tag{427}$$

$$\geq p_A\left(1 - O\left(\frac{1}{d^\tau}\right)\right) - O\left(\frac{1}{d^\tau}\right) \tag{428}$$

$$\geq p_A - O\left(\frac{1}{d^\tau}\right), \tag{429}$$

and we also have

$$\mathbb{E}_{(\mathbf{x},y)}\left[y\sigma'\left(\left\langle \mathbf{w}_i^{(0)},\mathbf{x}\right\rangle - \mathbf{b}_i\right)\phi_l\right] \tag{430}$$

$$=p_A\left(\mathbb{E}_{\mathbf{x}\sim\mathcal{X}_{j,+}}\left[\sigma'\left(\left\langle \mathbf{w}_i^{(0)},\mathbf{x}\right\rangle - \mathbf{b}_i\right)\right] + \mathbb{E}_{\mathbf{x}\sim\mathcal{X}_{j,-}}\left[\sigma'\left(\left\langle \mathbf{w}_i^{(0)},\mathbf{x}\right\rangle - \mathbf{b}_i\right)\right]\right) \tag{431}$$

$$+ (1-2rp_A)\mathbb{E}_{\mathbf{x}\sim\mathcal{X}_U}\left[y\sigma'\left(\left\langle \mathbf{w}_i^{(0)},\mathbf{x}\right\rangle - \mathbf{b}_i\right)\phi_l\right] \tag{432}$$

$$\leq p_A + O\left(\frac{1}{d^\tau}\right). \tag{433}$$

Similarly, for the second statement, for all $l \in A_{j'}$, any $j' \neq j, j' \in [r]$, by Lemma E.25 (3) and (5), we have

$$\left|\mathbb{E}_{(\mathbf{x},y)}\left[y\sigma'\left(\left\langle \mathbf{w}_i^{(0)},\mathbf{x}\right\rangle - \mathbf{b}_i\right)\phi_l\right]\right| \tag{434}$$

$$\leq \left|p_A\left(\mathbb{E}_{\mathbf{x}\sim\mathcal{X}_{j',+}}\left[\sigma'\left(\left\langle \mathbf{w}_i^{(0)},\mathbf{x}\right\rangle - \mathbf{b}_i\right)\right] + \mathbb{E}_{\mathbf{x}\sim\mathcal{X}_{j',-}}\left[\sigma'\left(\left\langle \mathbf{w}_i^{(0)},\mathbf{x}\right\rangle - \mathbf{b}_i\right)\right]\right)\right| + O\left(\frac{1}{d^\tau}\right)$$

$$\leq O\left(\frac{1}{d^\tau}\right). \tag{435}$$

For the third statement, for all $l \in A^\perp$, by Lemma E.25 (3), (4), (5), we have

$$\left|\mathbb{E}_{(\mathbf{x},y)}\left[y\sigma'\left(\left\langle \mathbf{w}_i^{(0)},\mathbf{x}\right\rangle - \mathbf{b}_i\right)\phi_l\right]\right| \tag{436}$$

$$\leq p_A \sum_{l\in[r]}\left|\mathbb{E}_{\mathbf{x}\sim\mathcal{X}_{l,+}}\left[\sigma'\left(\left\langle \mathbf{w}_i^{(0)},\mathbf{x}\right\rangle - \mathbf{b}_i\right)\phi_l\right] - \mathbb{E}_{\mathbf{x}\sim\mathcal{X}_{l,-}}\left[\sigma'\left(\left\langle \mathbf{w}_i^{(0)},\mathbf{x}\right\rangle - \mathbf{b}_i\right)\phi_l\right]\right| + O\left(\frac{1}{d^\tau}\right)$$

$$\leq p_A \left|\mathbb{E}_{\mathbf{x}\sim\mathcal{X}_{j,+}}\left[\sigma'\left(\left\langle \mathbf{w}_i^{(0)},\mathbf{x}\right\rangle - \mathbf{b}_i\right)\phi_l\right] - \mathbb{E}_{\mathbf{x}\sim\mathcal{X}_{j,-}}\left[\sigma'\left(\left\langle \mathbf{w}_i^{(0)},\mathbf{x}\right\rangle - \mathbf{b}_i\right)\phi_l\right]\right| + O\left(\frac{1}{d^\tau}\right)$$

$$\leq p_A \left|\mathbb{E}_{\mathbf{x}\sim\mathcal{X}_{j,+}}\left[\left(1 - \sigma'\left(\left\langle \mathbf{w}_i^{(0)},\mathbf{x}\right\rangle - \mathbf{b}_i\right)\right)\phi_l\right] - \mathbb{E}_{\mathbf{x}\sim\mathcal{X}_{j,-}}\left[\left(1 - \sigma'\left(\left\langle \mathbf{w}_i^{(0)},\mathbf{x}\right\rangle - \mathbf{b}_i\right)\right)\phi_l\right]\right|$$

$$+ p_A\left|\mathbb{E}_{\mathbf{x}\sim\mathcal{X}_{j,+}}[\phi_l] - \mathbb{E}_{\mathbf{x}\sim\mathcal{X}_{j,-}}[\phi_l]\right| + O\left(\frac{1}{d^\tau}\right) \tag{437}$$

$$=p_A\left|\mathbb{E}_{\mathbf{x}\sim\mathcal{X}_{j,+}}\left[\left(1 - \sigma'\left(\left\langle \mathbf{w}_i^{(0)},\mathbf{x}\right\rangle - \mathbf{b}_i\right)\right)\phi_l\right] - \mathbb{E}_{\mathbf{x}\sim\mathcal{X}_{j,-}}\left[\left(1 - \sigma'\left(\left\langle \mathbf{w}_i^{(0)},\mathbf{x}\right\rangle - \mathbf{b}_i\right)\right)\phi_l\right]\right|$$

$$+ O\left(\frac{1}{d^\tau}\right) \tag{438}$$

$$\leq O\left(\frac{1}{d^\tau}\right), \tag{439}$$

where the second inequality follows $2rp_A \leq 1$ and the third inequality follows the triangle inequality.
□

### E.4.3 Parity Functions: Final Guarantee

**Lemma E.27** (Parity Functions: Existence of Good Networks. Part statement of Lemma 4.7). *Assume the same conditions as in Lemma E.23. Define*

$$f^*(\mathbf{x}) = \sum_{j=1}^{r}\sum_{i=0}^{k}(-1)^{i+1}\sqrt{k} \tag{440}$$

$$\cdot \left[\sigma\left(\langle D_j,\mathbf{x}\rangle - \frac{2i-k-1}{\sqrt{k}}\right) - 2\sigma\left(\langle D_j,\mathbf{x}\rangle - \frac{2i-k}{\sqrt{k}}\right) + \sigma\left(\langle D_j,\mathbf{x}\rangle - \frac{2i-k+1}{\sqrt{k}}\right)\right].$$

*For $\mathcal{D}_{parity}$ setting, we have $f^* \in \mathcal{F}_{d,3r(k+1),B_F,S_{p,\gamma},B_G}$, where $B_F = (B_{a1}, B_{a2}, B_b) = \left(2\sqrt{k}, 2\sqrt{rk(k+1)}, \frac{k+1}{\sqrt{k}}\right)$, $p = \Theta\left(\frac{1}{\sqrt{\tau r\log d}\cdot d^{\left(9C_b^2\tau r/8\right)}}\right)$, $\gamma = \frac{1}{d^{\tau-2}}$, $B_G = \sqrt{k}p_A - O\left(\frac{\sqrt{k}}{d^\tau}\right)$ and $B_{x1} = \sqrt{d}, B_{x2} = d$. We also have $\mathrm{OPT}_{d,3r(k+1),B_F,S_{p,\gamma},B_G} = 0$.*

*Proof of Lemma E.27.* We can check $B_{x1} = \sqrt{d}, B_{x2} = d$ by direct calculation. By Lemma E.23, we have $f^* \in \mathcal{F}_{d,3r(k+1),B_F,S_{p,\gamma,B_G}}$. We note that

$$\sigma\left(\langle D_j, \mathbf{x}\rangle - \frac{2i-k-1}{\sqrt{k}}\right) - 2\sigma\left(\langle D_j, \mathbf{x}\rangle - \frac{2i-k}{\sqrt{k}}\right) + \sigma\left(\langle D_j, \mathbf{x}\rangle - \frac{2i-k+1}{\sqrt{k}}\right) \quad (441)$$

is a bump function for $\langle D_j, \mathbf{x}\rangle$ at $\frac{2i-k}{\sqrt{k}}$. We can check that $yf^*(\mathbf{x}) \geq 1$. Thus, we have

$$\mathrm{OPT}_{d,3r(k+1),B_F,S_{p,\gamma,B_G}} \leq \mathcal{L}_{\mathcal{D}_{parity}}(f^*) \quad (442)$$

$$= \mathbb{E}_{(\mathbf{x},y)\sim\mathcal{D}_{parity}}\mathcal{L}_{(\mathbf{x},y)}(f^*) \quad (443)$$

$$= 0. \quad (444)$$

$\square$

**Theorem 4.8** (Parity Functions: Main Result). *Assume Assumption 4.5. For any $\epsilon, \delta \in (0,1)$, when Algorithm 1 uses hinge loss with*

$$m = \mathrm{poly}\left(\frac{1}{\delta}, \frac{1}{\epsilon}, d^{\Theta(\tau r)}, k, \frac{1}{p_A}\right) \leq e^d, \quad T = \mathrm{poly}(m), \quad n = \mathrm{poly}(m)$$

*and proper hyper-parameters, then with probability at least $1-\delta$, there exists $t \in [T]$ such that*

$$\Pr[\mathrm{sign}(f_{\Xi^{(t)}}(\mathbf{x})) \neq y] \leq \frac{3r\sqrt{k}}{d^{(\tau-3)/2}} + \epsilon.$$

*Proof of Theorem 4.8.* Let $\tilde{b} = C_b\sqrt{\tau rk \log d}\sigma_w$, where $C_b$ is a large enough universal constant. By Lemma E.27, we have $f^* \in \mathcal{F}_{d,3r(k+1),B_F,S_{p,\gamma,B_G}}$, where $B_F = (B_{a1}, B_{a2}, B_b) = \left(2\sqrt{k}, 2\sqrt{rk(k+1)}, \frac{k+1}{\sqrt{k}}\right)$, $p = \Theta\left(\frac{1}{\sqrt{\tau r \log d} \cdot d^{\left(9C_b^2 \tau r/8\right)}}\right)$, $\gamma = \frac{1}{d^{\tau-2}}$, $B_G = \sqrt{k}p_A - O\left(\frac{\sqrt{k}}{d^{\tau}}\right)$ and $B_{x1} = \sqrt{d}, B_{x2} = d$. We also have $\mathrm{OPT}_{d,3r(k+1),B_F,S_{p,\gamma,B_G}} = 0$.

Adjust $\sigma_w$ such that $\tilde{b} = C_b\sqrt{\tau rk \log d}\sigma_w = \Theta\left(\frac{B_G^{\frac{1}{4}}B_{a2}B_b^{\frac{3}{4}}}{\sqrt{r}B_{a1}}\right)$. Injecting above parameters into Theorem 3.12, we have with probability at least $1-\delta$ over the initialization, with proper hyper-parameters, there exists $t \in [T]$ such that

$$\Pr[\mathrm{sign}(f_{\Xi^{(t)}}(\mathbf{x})) \neq y] \leq \frac{2\sqrt{2}r\sqrt{k}}{d^{(\tau-3)/2}} + O\left(\frac{rB_{a1}B_{x1}B_{x2}^{\frac{1}{4}}(\log n)^{\frac{1}{4}}}{\sqrt{B_G}n^{\frac{1}{4}}}\right) + \epsilon/2 \leq \frac{3r\sqrt{k}}{d^{(\tau-3)/2}} + \epsilon.$$

$\square$

## E.5 Uniform Parity Functions

We consider the sparse parity problem over the uniform data distribution studied in [15]. We use the properties of the problem to prove the key lemma (i.e., the existence of good networks) in our framework and then derive the final guarantee from our theorem of the simple setting (Theorem 3.4). We provide Theorem E.31 as (1) use it as a warm-up and (2) follow the original analysis in [15] to give a comparison. We will provide Theorem E.40 as an alternative version that trains both layers.

Consider the same data distribution in Appendix E.4.1 and Definition E.22 with the following assumptions.

**Assumption E.28** (Uniform Parity Functions). *We follow the data distribution in Appendix E.4.1. Let $r = 1, p_A = 0, p_o = \frac{1}{2}, \mathbf{M} = I_{d\times d}$ and $d \geq 2k^2$, and $k$ is an even number.*

We denote this data distribution as $\mathcal{D}_{parity-uniform}$ setting.

To apply our framework, again we only need to compute the parameters in the Gradient Feature set and the corresponding optimal approximation loss. To this end, we first define the gradient features: let

$$D = \frac{\sum_{l\in A}\mathbf{M}_l}{\|\sum_{l\in A}\mathbf{M}_l\|_2}. \quad (445)$$

We follow the initialization and training dynamic in [15].

**Initialization and Loss.** We use hinge loss and we have unbiased initialization, for all $i \in [m]$,

$$\mathbf{a}_i^{(0)} \sim \text{Unif}(\{\pm 1\}), \mathbf{w}_i^{(0)} \sim \text{Unif}(\{\pm 1\}^d), \mathbf{b}_i = \text{Unif}(\{-1 + 1/k, \ldots, 1 - 1/k\}). \qquad (446)$$

**Training Process.** We use the following one-step training algorithm for this specific data distribution.

---

**Algorithm 4** Network Training via Gradient Descent [15]. Special case of Algorithm 2

---

Initialize $(\mathbf{a}^{(0)}, \mathbf{W}^{(0)}, \mathbf{b})$ as in Equation (8) and Equation (446); Sample $\mathcal{Z} \sim \mathcal{D}_{parity-uniform}^n$
$\mathbf{W}^{(1)} = \mathbf{W}^{(0)} - \eta^{(1)}(\nabla_\mathbf{W}\widetilde{\mathcal{L}}_\mathcal{Z}(f_{\Xi^{(0)}}) + \lambda^{(1)}\mathbf{W}^{(0)})$
$\mathbf{a}^{(1)} = \mathbf{a}^{(0)} - \eta^{(1)}(\nabla_\mathbf{a}\widetilde{\mathcal{L}}_\mathcal{Z}(f_{\Xi^{(0)}}) + \lambda^{(1)}\mathbf{a}^{(0)})$
**for** $t = 2$ **to** $T$ **do**
$\quad \mathbf{a}^{(t)} = \mathbf{a}^{(t-1)} - \eta^{(t)}\nabla_\mathbf{a}\widetilde{\mathcal{L}}_\mathcal{Z}(f_{\Xi^{(t-1)}})$
**end for**

---

Use the notation in Section 5.3 of [92], for every $S \in [n]$, s.t. $|S| = k$, we define

$$\xi_k := \widehat{\text{Maj}}(S) = (-1)^{\frac{k-1}{2}} \frac{\binom{\frac{d-1}{2}}{\frac{d-1}{2}}}{\binom{d-1}{k-1}} \cdot 2^{-(d-1)} \binom{d-1}{\frac{d-1}{2}}. \qquad (447)$$

**Lemma E.29** (Uniform Parity Functions: Existence of Good Networks. Rephrase of Lemma 5 in [15]). *For every $\epsilon, \delta \in (0, 1/2)$, denoting $\tau = \frac{|\xi_{k-1}|}{16k\sqrt{2d\log(32k^3d/\epsilon)}}$, let $\eta^{(1)} = \frac{1}{k|\xi_{k-1}|}$, $\lambda^{(1)} = \frac{1}{\eta^{(1)}}$, $m \geq k \cdot 2^k \log(k/\delta)$, $n \geq \frac{2}{\tau^2}\log(4dm/\delta)$ and $d \geq \Omega\left(k^4\log(kd/\epsilon)\right)$, w.p. at least $1 - 2\delta$ over the initialization and the training samples, there exists $\tilde{\mathbf{a}} \in \mathbb{R}^m$ with $\|\tilde{\mathbf{a}}\|_\infty \leq 8k$ and $\|\tilde{\mathbf{a}}\|_2 \leq 8k\sqrt{k}$ such that $f_{(\tilde{\mathbf{a}}, \mathbf{W}^{(1)}, \mathbf{b})}$ satisfies*

$$\mathcal{L}_{\mathcal{D}_{parity-uniform}}\left(f_{(\tilde{\mathbf{a}}, \mathbf{W}^{(1)}, \mathbf{b})}\right) \leq \epsilon. \qquad (448)$$

*Additionally, it holds that $\|\sigma(\mathbf{W}^{(1)^\top}\mathbf{x} - \mathbf{b})\|_\infty \leq d + 1$.*

**Remark E.30.** *In [15], they update the bias term in the first gradient step. However, if we check the proof carefully, we can see that the fixed bias still goes through all their analysis.*

### E.5.1 Uniform Parity Functions: Final Guarantee

Considering training by Algorithm 4, we have the following results.

**Theorem E.31** (Uniform Parity Functions: Main Result). *Fix $\epsilon \in (0, 1/2)$ and let $m \geq \Omega\left(k \cdot 2^k \log(k/\epsilon)\right)$, $n \geq \Omega\left(k^{7/6}d\binom{d}{k-1}\log(kd/\epsilon)\log(dm/\epsilon) + \frac{k^3md^2}{\epsilon^2}\right)$, $d \geq \Omega\left(k^4\log(kd/\epsilon)\right)$. Let $\eta^{(1)} = \frac{1}{k|\xi_{k-1}|}$, $\lambda^{(1)} = \frac{1}{\eta^{(1)}}$, and $\eta = \eta^{(t)} = \Theta\left(\frac{1}{d^2m}\right)$, for all $t \in \{2, 3, \ldots, T\}$. If $T \geq \Omega\left(\frac{k^3md^2}{\epsilon}\right)$, then training by Algorithm 4 with hinge loss, w.h.p. over the initialization and the training samples, there exists $t \in [T]$ such that*

$$\Pr[\text{sign}(f_{\Xi^{(t)}})(\mathbf{x}) \neq y] \leq \mathcal{L}_{\mathcal{D}_{parity-uniform}}f_{(\mathbf{a}^{(t)}, \mathbf{W}^{(1)}, \mathbf{b})} \leq \epsilon. \qquad (449)$$

*Proof of Theorem E.31.* By Lemma E.29, w.h.p., we have for properly chosen hyper-parameters,

$$\text{OPT}_{\mathbf{W}^{(1)}, \mathbf{b}, B_{a2}} \leq \mathcal{L}_{\mathcal{D}_{parity-uniform}}\left(f_{(\tilde{\mathbf{a}}, \mathbf{W}^{(1)}, \mathbf{b})}\right) \leq \frac{\epsilon}{3}. \qquad (450)$$

We compute the $L$-smooth constant of $\widetilde{\mathcal{L}}_{\mathcal{Z}}\left(f_{(\mathbf{a},\mathbf{W}^{(1)},\mathbf{b})}\right)$ to $\mathbf{a}$.

$$\left\|\nabla_{\mathbf{a}}\widetilde{\mathcal{L}}_{\mathcal{Z}}\left(f_{(\mathbf{a}_1,\mathbf{W}^{(1)},\mathbf{b})}\right) - \nabla_{\mathbf{a}}\widetilde{\mathcal{L}}_{\mathcal{Z}}\left(f_{(\mathbf{a}_2,\mathbf{W}^{(1)},\mathbf{b})}\right)\right\|_2 \tag{451}$$

$$=\left\|\frac{1}{n}\sum_{\mathbf{x}\in\mathcal{Z}}\left[\left(\ell'\left(yf_{(\mathbf{a}_1,\mathbf{W}^{(1)},\mathbf{b})}(\mathbf{x})\right) - \ell'\left(yf_{(\mathbf{a}_2,\mathbf{W}^{(1)},\mathbf{b})}(\mathbf{x})\right)\right)\sigma(\mathbf{W}^{(1)\top}\mathbf{x}-\mathbf{b})\right]\right\|_2 \tag{452}$$

$$\leq\left\|\frac{1}{n}\sum_{\mathbf{x}\in\mathcal{Z}}\left[\left|f_{(\mathbf{a}_1,\mathbf{W}^{(1)},\mathbf{b})}(\mathbf{x}) - f_{(\mathbf{a}_2,\mathbf{W}^{(1)},\mathbf{b})}(\mathbf{x})\right|\sigma(\mathbf{W}^{(1)\top}\mathbf{x}-\mathbf{b})\right]\right\|_2 \tag{453}$$

$$\leq\frac{1}{n}\sum_{\mathbf{x}\in\mathcal{Z}}\left[\|\mathbf{a}_1-\mathbf{a}_2\|_2\left\|\sigma(\mathbf{W}^{(1)\top}\mathbf{x}-\mathbf{b})\right\|_2^2\right]. \tag{454}$$

By the Lemma E.29, we have $\|\sigma(\mathbf{W}^{(1)^\top}\mathbf{x}-\mathbf{b})\|_\infty \leq d+1$. Thus, we have,

$$L = O\left(\frac{1}{n}\sum_{\mathbf{x}\in\mathcal{Z}}\left\|\sigma(\mathbf{W}^{(1)\top}\mathbf{x}-\mathbf{b})\right\|_2^2\right) \tag{455}$$

$$\leq O(d^2 m). \tag{456}$$

This means that we can let $\eta = \Theta\left(\frac{1}{d^2 m}\right)$ and we will get our convergence result. Note that we have $\mathbf{a}^{(1)} = \mathbf{0}$ and $\|\tilde{\mathbf{a}}\|_2 = O\left(k\sqrt{k}\right)$. So, if we choose $T \geq \Omega\left(\frac{k^3}{\epsilon\eta}\right)$, there exists $t \in [T]$ such that

$$\widetilde{\mathcal{L}}_{\mathcal{Z}}\left(f_{(\mathbf{a}^{(t)},\mathbf{W}^{(1)},\mathbf{b})}\right) - \widetilde{\mathcal{L}}_{\mathcal{Z}}\left(f_{(\tilde{\mathbf{a}},\mathbf{W}^{(1)},\mathbf{b})}\right) \leq O\left(\frac{L\|\mathbf{a}^{(1)}-\tilde{\mathbf{a}}\|_2^2}{T}\right) \leq \epsilon/3.$$

We also have $\sqrt{\frac{\|\tilde{\mathbf{a}}\|_2^2(\|\mathbf{W}^{(1)}\|_F^2 B_x^2 + \|\mathbf{b}\|_2^2)}{n}} \leq \frac{\epsilon}{3}$. Then our theorem gets proved by Theorem 3.4. $\qquad\square$

### E.6 Uniform Parity Functions: Alternative Analysis

It is also possible to unify [15] into our general Gradient Feature Learning Framework by mildly modifying the framework in Theorem 3.12. In order to do that, we first need to use a different metric in the definition of gradient features.

#### E.6.1 Modified General Feature Learning Framework for Uniform Parity Functions

**Definition E.32** (Gradient Feature with Infinity Norm). *For a unit vector $D \in \mathbb{R}^d$ with $\|D\|_2 = 1$, and a $\gamma_\infty \in (0,1)$, a direction neighborhood (cone) $\mathcal{C}_{D,\gamma_\infty}^\infty$ is defined as: $\mathcal{C}_{D,\gamma_\infty}^\infty := \left\{\mathbf{w} \,\Big|\, \left\|\frac{\mathbf{w}}{\|\mathbf{w}\|} - D\right\|_\infty < \gamma_\infty\right\}$. Let $\mathbf{w} \in \mathbb{R}^d$, $b \in \mathbb{R}$ be random variables drawn from some distribution $\mathcal{W},\mathcal{B}$. A Gradient Feature set with parameters $p, \gamma_\infty, B_G, B_{G1}$ is defined as:*

$$S_{p,\gamma_\infty,B_G,B_{G1}}^\infty(\mathcal{W},\mathcal{B}) := \left\{(D,s) \,\Big|\, \Pr_{\mathbf{w},b}\left[G(\mathbf{w},b) \in \mathcal{C}_{D,\gamma_\infty}^\infty, B_{G1} \geq \|G(\mathbf{w},b)\|_2 \geq B_G, s = \frac{b}{|b|}\right] \geq p\right\}.$$

*When clear from context, write it as $S_{p,\gamma_\infty,B_G,B_{G1}}^\infty$.*

**Definition E.33** (Optimal Approximation via Gradient Features with Infinity Norm). *The Optimal Approximation network and loss using gradient feature induced networks $\mathcal{F}_{d,r,B_F,S_{p,\gamma_\infty,B_G,B_{G1}}^\infty}$ are defined as:*

$$f^* := \operatorname*{argmin}_{f\in\mathcal{F}_{d,r,B_F,S_{p,\gamma_\infty,B_G,B_{G1}}^\infty}} \mathcal{L}_{\mathcal{D}}(f), \tag{457}$$

$$\mathrm{OPT}_{d,r,B_F,S_{p,\gamma_\infty,B_G,B_{G1}}^\infty} := \min_{f\in\mathcal{F}_{d,r,B_F,S_{p,\gamma_\infty,B_G,B_{G1}}^\infty}} \mathcal{L}_{\mathcal{D}}(f). \tag{458}$$

We consider the data distribution in Appendix E.4.1 with Assumption E.28, i.e., $\mathcal{D}_{parity-uniform}$ in Appendix E.5. Note that with this dataset, we have $\|\mathbf{x}\|_\infty \leq B_{x\infty} = 1$. We use the following unbiased initialization:

$$\text{for } i \in \{1,\ldots,m\}: \quad \mathbf{a}_i^{(0)} \sim \mathcal{N}(0,\sigma_a^2), \mathbf{w}_i^{(0)} \sim \{\pm 1\}^d, \mathbf{b}_i = \tilde{b} \leq 1,$$

$$\text{for } i \in \{m+1,\ldots,2m\}: \quad \mathbf{a}_i^{(0)} = -\mathbf{a}_{i-m}^{(0)}, \mathbf{w}_i^{(0)} = -\mathbf{w}_{i-m}^{(0)}, \mathbf{b}_i = -\mathbf{b}_{i-m},$$

$$\text{for } i \in \{2m+1,\ldots,4m\}: \quad \mathbf{a}_i^{(0)} = -\mathbf{a}_{i-2m}^{(0)}, \mathbf{w}_i^{(0)} = \mathbf{w}_{i-2m}^{(0)}, \mathbf{b}_i = \mathbf{b}_{i-2m} \tag{459}$$

Let $\nabla_i$ denote the gradient of the $i$-th neuron $\nabla_{\mathbf{w}_i}\mathcal{L}_{\mathcal{D}}(f_{\Xi^{(0)}})$. Denote the subset of neurons with nice gradients approximating feature $(D, s)$ as:

$$G^{\infty}_{(D,s),Nice} := \left\{ i \in [2m] : s = \frac{\mathbf{b}_i}{|\mathbf{b}_i|}, \left\| \frac{\nabla_i}{\|\nabla_i\|} - D \right\|_{\infty} \leq \gamma_{\infty}, \left| \mathbf{a}_i^{(0)} \right| B_{G1} \geq \|\nabla_i\|_2 \geq \left| \mathbf{a}_i^{(0)} \right| B_G \right\}.$$

**Lemma E.34** (Existence of Good Networks. Modified Version of Lemma 3.14 Under Uniform Parity Setting). *Let $\lambda^{(1)} = \frac{1}{\eta^{(1)}}$. For any $B_{\epsilon} \in (0, B_b)$, let $\sigma_a = \Theta\left( \frac{\tilde{b}}{-\ell'(0)\eta^{(1)}B_G B_{\epsilon}} \right)$ and $\delta = 2re^{-\sqrt{mp}} + \frac{1}{d^2}$. Then, with probability at least $1 - \delta$ over the initialization, there exists $\tilde{\mathbf{a}}_i$'s such that $f_{(\tilde{\mathbf{a}}, \mathbf{W}^{(1)}, \mathbf{b})}(\mathbf{x}) = \sum_{i=1}^{4m} \tilde{\mathbf{a}}_i \sigma\left( \left\langle \mathbf{w}_i^{(1)}, \mathbf{x} \right\rangle - \mathbf{b}_i \right)$ satisfies*

$$\mathcal{L}_{\mathcal{D}}(f_{(\tilde{\mathbf{a}}, \mathbf{W}^{(1)}, \mathbf{b})}) \leq rB_{a1}\left( \frac{B_{x1}B_{G1}B_b}{\sqrt{mp}B_G B_{\epsilon}} + \sqrt{2\log(d)}d\gamma_{\infty} + B_{\epsilon} \right) + \text{OPT}_{d,r,B_F,S^{\infty}_{p,\gamma_{\infty},B_G,B_{G1}}},$$

*and $\|\tilde{\mathbf{a}}\|_0 = O\left( r(mp)^{\frac{1}{2}} \right), \|\tilde{\mathbf{a}}\|_2 = O\left( \frac{B_{a2}B_b}{\tilde{b}(mp)^{\frac{1}{4}}} \right), \|\tilde{\mathbf{a}}\|_{\infty} = O\left( \frac{B_{a1}B_b}{\tilde{b}(mp)^{\frac{1}{2}}} \right)$.*

*Proof of Lemma E.34.* Recall $f^*(\mathbf{x}) = \sum_{j=1}^r \mathbf{a}_j^* \sigma(\langle \mathbf{w}_j^*, \mathbf{x} \rangle - \mathbf{b}_j^*)$, where $f^* \in \mathcal{F}_{d,r,B_F,S^{\infty}_{p,\gamma_{\infty},B_G,B_{G1}}}$ is defined in Definition E.33 and let $s_j^* = \frac{\mathbf{b}_j^*}{|\mathbf{b}_j^*|}$. By Lemma D.3, with probability at least $1 - \delta_1$, $\delta_1 = 2re^{-cmp}$, for all $j \in [r]$, we have $|G^{\infty}_{(\mathbf{w}_j^*, s_j^*),Nice}| \geq \frac{mp}{4}$. Then for all $i \in G^{\infty}_{(\mathbf{w}_j^*, s_j^*),Nice} \subseteq [2m]$, we have $-\ell'(0)\eta^{(1)}G(\mathbf{w}_i^{(0)}, \mathbf{b}_i)\frac{\mathbf{b}_j^*}{\tilde{b}}$ only depend on $\mathbf{w}_i^{(0)}$ and $\mathbf{b}_i$, which is independent of $\mathbf{a}_i^{(0)}$. Given Definition E.32, we have

$$-\ell'(0)\eta^{(1)}\|G(\mathbf{w}_i^{(0)}, \mathbf{b}_i)\|_2 \frac{\mathbf{b}_j^*}{\tilde{b}} \in \left[ \ell'(0)\eta^{(1)}B_{x1}\frac{B_b}{\tilde{b}}, -\ell'(0)\eta^{(1)}B_{x1}\frac{B_b}{\tilde{b}} \right]. \tag{460}$$

We split $[r]$ into $\Gamma = \{j \in [r] : |\mathbf{b}_j^*| < B_{\epsilon}\}$, $\Gamma_- = \{j \in [r] : \mathbf{b}_j^* \leq -B_{\epsilon}\}$ and $\Gamma_+ = \{j \in [r] : \mathbf{b}_j^* \geq B_{\epsilon}\}$. Let $\epsilon_a = \frac{B_{G1}B_b}{\sqrt{mp}B_G B_{\epsilon}}$. Then we know that for all $j \in \Gamma_+ \cup \Gamma_-$, for all $i \in G^{\infty}_{(\mathbf{w}_j^*, s_j^*),Nice}$, we have

$$\Pr_{\mathbf{a}_i^{(0)} \sim \mathcal{N}(0, \sigma_a^2)}\left[ \left| -\mathbf{a}_i^{(0)}\ell'(0)\eta^{(1)}\|G(\mathbf{w}_i^{(0)}, \mathbf{b}_i)\|_2 \frac{|\mathbf{b}_j^*|}{\tilde{b}} - 1 \right| \leq \epsilon_a \right] \tag{461}$$

$$= \Pr_{\mathbf{a}_i^{(0)} \sim \mathcal{N}(0, \sigma_a^2)}\left[ 1 - \epsilon_a \leq -\mathbf{a}_i^{(0)}\ell'(0)\eta^{(1)}\|G(\mathbf{w}_i^{(0)}, \mathbf{b}_i)\|_2 \frac{|\mathbf{b}_j^*|}{\tilde{b}} \leq 1 + \epsilon_a \right] \tag{462}$$

$$= \Pr_{g \sim \mathcal{N}(0,1)}\left[ 1 - \epsilon_a \leq g\Theta\left( \frac{\|G(\mathbf{w}_i^{(0)}, \mathbf{b}_i)\|_2 |\mathbf{b}_j^*|}{B_G B_{\epsilon}} \right) \leq 1 + \epsilon_a \right] \tag{463}$$

$$= \Pr_{g \sim \mathcal{N}(0,1)}\left[ (1 - \epsilon_a)\Theta\left( \frac{B_G B_{\epsilon}}{\|G(\mathbf{w}_i^{(0)}, \mathbf{b}_i)\|_2 |\mathbf{b}_j^*|} \right) \leq g \leq (1 + \epsilon_a)\Theta\left( \frac{B_G B_{\epsilon}}{\|G(\mathbf{w}_i^{(0)}, \mathbf{b}_i)\|_2 |\mathbf{b}_j^*|} \right) \right]$$

$$= \Theta\left( \frac{\epsilon_a B_G B_{\epsilon}}{\|G(\mathbf{w}_i^{(0)}, \mathbf{b}_i)\|_2 |\mathbf{b}_j^*|} \right) \tag{464}$$

$$\geq \Omega\left( \frac{\epsilon_a B_G B_{\epsilon}}{B_{G1}B_b} \right) \tag{465}$$

$$= \Omega\left( \frac{1}{\sqrt{mp}} \right). \tag{466}$$

Thus, with probability $\Omega\left( \frac{1}{\sqrt{mp}} \right)$ over $\mathbf{a}_i^{(0)}$, we have

$$\left| -\mathbf{a}_i^{(0)}\ell'(0)\eta^{(1)}\|G(\mathbf{w}_i^{(0)}, \mathbf{b}_i)\|_2 \frac{|\mathbf{b}_j^*|}{\tilde{b}} - 1 \right| \leq \epsilon_a, \quad \left| \mathbf{a}_i^{(0)} \right| = O\left( \frac{\tilde{b}}{-\ell'(0)\eta^{(1)}B_G B_{\epsilon}} \right). \tag{467}$$

Similarly, for $j \in \Gamma$, for all $i \in G^\infty_{(\mathbf{w}^*_j, s^*_j), Nice}$, with probability $\Omega\left(\frac{1}{\sqrt{mp}}\right)$ over $\mathbf{a}^{(0)}_i$, we have

$$\left| -\mathbf{a}^{(0)}_i \ell'(0)\eta^{(1)} \|G(\mathbf{w}^{(0)}_i, \mathbf{b}_i)\|_2 \frac{B_\epsilon}{\tilde{b}} - 1 \right| \leq \epsilon_a, \quad \left|\mathbf{a}^{(0)}_i\right| = O\left(\frac{\tilde{b}}{-\ell'(0)\eta^{(1)} B_G B_\epsilon}\right). \tag{468}$$

For all $j \in [r]$, let $\Lambda_j \subseteq G^\infty_{(\mathbf{w}^*_j, s^*_j), Nice}$ be the set of $i$'s such that condition Equation (467) or Equation (468) are satisfied. By Chernoff bound and union bound, with probability at least $1 - \delta_2$, $\delta_2 = re^{-\sqrt{mp}}$, for all $j \in [r]$ we have $|\Lambda_j| \geq \Omega(\sqrt{mp})$. We have for $\forall j \in \Gamma_+ \cup \Gamma_-, \forall i \in \Lambda_j$,

$$\left| \frac{|\mathbf{b}^*_j|}{\tilde{b}} \left\langle \mathbf{w}^{(1)}_i, \mathbf{x} \right\rangle - \left\langle \mathbf{w}^*_j, \mathbf{x} \right\rangle \right| \tag{469}$$

$$\leq \left| \left\langle -\mathbf{a}^{(0)}_i \ell'(0)\eta^{(1)} \|G(\mathbf{w}^{(0)}_i, \mathbf{b}_i)\|_2 \frac{|\mathbf{b}^*_j|}{\tilde{b}} \frac{\mathbf{w}^{(1)}_i}{\|\mathbf{w}^{(1)}_i\|_2} - \frac{\mathbf{w}^{(1)}_i}{\|\mathbf{w}^{(1)}_i\|_2}, \mathbf{x} \right\rangle + \left\langle \frac{\mathbf{w}^{(1)}_i}{\|\mathbf{w}^{(1)}_i\|_2} - \mathbf{w}^*_j, \mathbf{x} \right\rangle \right|$$

$$\leq \epsilon_a \|\mathbf{x}\|_2 + \sqrt{2\log(d)d}\gamma_\infty. \tag{470}$$

With probability $1 - \frac{1}{d^2}$ by Hoeffding's inequality. Similarly, for $\forall j \in \Gamma, \forall i \in \Lambda_j$,

$$\left| \frac{B_\epsilon}{\tilde{b}} \left\langle \mathbf{w}^{(1)}_i, \mathbf{x} \right\rangle - \left\langle \mathbf{w}^*_j, \mathbf{x} \right\rangle \right| \leq \epsilon_a \|\mathbf{x}\|_2 + \sqrt{2\log(d)d}\gamma_\infty. \tag{471}$$

If $i \in \Lambda_j, j \in \Gamma_+ \cup \Gamma_-$, set $\tilde{\mathbf{a}}_i = \mathbf{a}^*_j \frac{|\mathbf{b}^*_j|}{|\Lambda_j|\tilde{b}}$, if $i \in \Lambda_j, j \in \Gamma$, set $\tilde{\mathbf{a}}_i = \mathbf{a}^*_j \frac{B_\epsilon}{|\Lambda_j|\tilde{b}}$, otherwise set $\tilde{\mathbf{a}}_i = 0$, we have $\|\tilde{\mathbf{a}}\|_0 = O\left(r(mp)^{\frac{1}{2}}\right)$, $\|\tilde{\mathbf{a}}\|_2 = O\left(\frac{B_{a2}B_b}{\tilde{b}(mp)^{\frac{1}{4}}}\right)$, $\|\tilde{\mathbf{a}}\|_\infty = O\left(\frac{B_{a1}B_b}{\tilde{b}(mp)^{\frac{1}{2}}}\right)$.

Finally, we have

$$\mathcal{L}_\mathcal{D}(f_{(\tilde{\mathbf{a}}, \mathbf{W}^{(1)}, \mathbf{b})}) \tag{472}$$

$$=\mathcal{L}_\mathcal{D}(f_{(\tilde{\mathbf{a}}, \mathbf{W}^{(1)}, \mathbf{b})}) - \mathcal{L}_\mathcal{D}(f^*) + \mathcal{L}_\mathcal{D}(f^*) \tag{473}$$

$$\leq \mathbb{E}_{(\mathbf{x},y)} \left[ \left| f_{(\tilde{\mathbf{a}}, \mathbf{W}^{(1)}, \mathbf{b})}(\mathbf{x}) - f^*(\mathbf{x}) \right| \right] + \mathcal{L}_\mathcal{D}(f^*) \tag{474}$$

$$\leq \mathbb{E}_{(\mathbf{x},y)} \left[ \left| \sum_{i=1}^m \tilde{\mathbf{a}}_i \sigma\left( \left\langle \mathbf{w}_i^{(1)}, \mathbf{x} \right\rangle - \tilde{b} \right) + \sum_{i=m+1}^{2m} \tilde{\mathbf{a}}_i \sigma\left( \left\langle \mathbf{w}_i^{(1)}, \mathbf{x} \right\rangle + \tilde{b} \right) - \sum_{j=1}^r \mathbf{a}_j^* \sigma(\langle \mathbf{w}_j^*, \mathbf{x} \rangle - \mathbf{b}_j^*) \right| \right]$$
$$+ \mathcal{L}_\mathcal{D}(f^*) \tag{475}$$

$$\leq \mathbb{E}_{(\mathbf{x},y)} \left[ \left| \sum_{j \in \Gamma_+} \sum_{i \in \Lambda_j} \mathbf{a}_j^* \frac{1}{|\Lambda_j|} \left| \frac{|\mathbf{b}_j^*|}{\tilde{b}} \right| \sigma\left( \left\langle \mathbf{w}_i^{(1)}, \mathbf{x} \right\rangle - \tilde{b} \right) - \sigma(\langle \mathbf{w}_j^*, \mathbf{x} \rangle - \mathbf{b}_j^*) \right| \right] \tag{476}$$

$$+ \mathbb{E}_{(\mathbf{x},y)} \left[ \left| \sum_{j \in \Gamma_-} \sum_{i \in \Lambda_j} \mathbf{a}_j^* \frac{1}{|\Lambda_j|} \left| \frac{|\mathbf{b}_j^*|}{\tilde{b}} \right| \sigma\left( \left\langle \mathbf{w}_i^{(1)}, \mathbf{x} \right\rangle + \tilde{b} \right) - \sigma(\langle \mathbf{w}_j^*, \mathbf{x} \rangle - \mathbf{b}_j^*) \right| \right] \tag{477}$$

$$+ \mathbb{E}_{(\mathbf{x},y)} \left[ \left| \sum_{j \in \Gamma} \sum_{i \in \Lambda_j} \mathbf{a}_j^* \frac{1}{|\Lambda_j|} \left| \frac{B_\epsilon}{\tilde{b}} \right| \sigma\left( \left\langle \mathbf{w}_i^{(1)}, \mathbf{x} \right\rangle - \tilde{b} \right) - \sigma(\langle \mathbf{w}_j^*, \mathbf{x} \rangle - \mathbf{b}_j^*) \right| \right] + \mathcal{L}_\mathcal{D}(f^*) \tag{478}$$

$$\leq \mathbb{E}_{(\mathbf{x},y)} \left[ \left| \sum_{j \in \Gamma_+} \sum_{i \in \Lambda_j} \mathbf{a}_j^* \frac{1}{|\Lambda_j|} \left| \frac{|\mathbf{b}_j^*|}{\tilde{b}} \left\langle \mathbf{w}_i^{(1)}, \mathbf{x} \right\rangle - \langle \mathbf{w}_j^*, \mathbf{x} \rangle \right| \right| \right] \tag{479}$$

$$+ \mathbb{E}_{(\mathbf{x},y)} \left[ \left| \sum_{j \in \Gamma_-} \sum_{i \in \Lambda_j} \mathbf{a}_j^* \frac{1}{|\Lambda_j|} \left| \frac{|\mathbf{b}_j^*|}{\tilde{b}} \left\langle \mathbf{w}_i^{(1)}, \mathbf{x} \right\rangle - \langle \mathbf{w}_j^*, \mathbf{x} \rangle \right| \right| \right] \tag{480}$$

$$+ \mathbb{E}_{(\mathbf{x},y)} \left[ \left| \sum_{j \in \Gamma} \sum_{i \in \Lambda_j} \mathbf{a}_j^* \frac{1}{|\Lambda_j|} \left| \frac{B_\epsilon}{\tilde{b}} \left\langle \mathbf{w}_i^{(1)}, \mathbf{x} \right\rangle + B_\epsilon - \langle \mathbf{w}_j^*, \mathbf{x} \rangle \right| \right| \right] + \mathcal{L}_\mathcal{D}(f^*) \tag{481}$$

$$\leq r\|\mathbf{a}^*\|_\infty (\epsilon_a \mathbb{E}_{(\mathbf{x},y)} \|\mathbf{x}\|_2 + \sqrt{2\log(d)d}\gamma_\infty) + |\Gamma|\|\mathbf{a}^*\|_\infty B_\epsilon + \mathcal{L}_\mathcal{D}(f^*) \tag{482}$$

$$\leq rB_{a1}(\epsilon_a B_{x1} + \sqrt{2\log(d)d}\gamma_\infty) + |\Gamma|B_{a1}B_\epsilon + \mathrm{OPT}_{d,r,B_F,S_{p,\gamma,B_G,B_{G1}}^\infty}. \tag{483}$$

We finish the proof by union bound and $\delta \geq \delta_1 + \delta_2 + \frac{1}{d^2}$. $\qquad\square$

**Lemma E.35** (Empirical Gradient Concentration Bound for Single Coordinate)**.** *For $i \in [m]$, when $n \geq (\log(d))^6$, with probability at least $1 - O\left( \exp\left( -n^{\frac{1}{3}} \right) \right)$ over training samples, we have*

$$\left| \frac{\partial \widetilde{\mathcal{L}}_\mathcal{Z}(f_\Xi)}{\partial \mathbf{w}_{i,j}} - \frac{\partial \mathcal{L}_\mathcal{D}(f_\Xi)}{\partial \mathbf{w}_{i,j}} \right| \leq O\left( \frac{|\mathbf{a}_i| B_{x\infty}}{n^{\frac{1}{3}}} \right), \quad \forall j \in [d]. \tag{484}$$

*Proof of Lemma E.35.* First, we define,

$$z_{i,j}^{(l)} = \ell'(y^{(l)} f_\Xi(\mathbf{x}^{(l)})) y^{(l)} \left[ \sigma'\left( \left\langle \mathbf{w}_i, \mathbf{x}^{(l)} \right\rangle - \mathbf{b}_i \right) \mathbf{x}_j^{(l)} \right] \tag{485}$$
$$- \mathbb{E}_{(\mathbf{x},y)} \left[ \ell'(y f_\Xi(\mathbf{x})) y \left[ \sigma'\left( \langle \mathbf{w}_i, \mathbf{x} \rangle - \mathbf{b}_i \right) \mathbf{x}_j \right] \right]. \tag{486}$$

As $|\ell'(z)| \leq 1, |y| \leq 1, |\sigma'(z)| \leq 1$, we have $z_{i,j}^{(l)}$ is zero-mean random variable with $\left| z_{i,j}^{(l)} \right| \leq 2B_{x\infty}$ as well as $\mathbb{E}\left[ \left| z_{i,j}^{(l)} \right|_2^2 \right] \leq 4B_{x\infty}^2$. Then by Bernstein Inequality, for $0 < z < 2B_{x\infty}$, we have

$$\Pr\left( \left| \frac{\partial \widetilde{\mathcal{L}}_\mathcal{Z}(f_\Xi)}{\partial \mathbf{w}_{i,j}} - \frac{\partial \mathcal{L}_\mathcal{D}(f_\Xi)}{\partial \mathbf{w}_{i,j}} \right| \geq |\mathbf{a}_i| z \right) = \Pr\left( \left| \frac{1}{n} \sum_{l \in [n]} z_{i,j}^{(l)} \right| \geq z \right) \tag{487}$$

$$\leq \exp\left( -n \cdot \frac{z^2}{8B_{x\infty}} \right). \tag{488}$$

Thus, for some $i \in [m]$, when $n \geq (\log(d))^6$, with probability at least $1 - O\left(\exp\Theta\left(-n^{\frac{1}{3}}\right)\right)$, from a union bound over $j \in [d]$, we have, for $\forall j \in [d]$,

$$\left|\frac{\partial\widetilde{\mathcal{L}}_{\mathcal{Z}}(f_\Xi)}{\partial\mathbf{w}_{i,j}} - \frac{\partial\mathcal{L}_{\mathcal{D}}(f_\Xi)}{\partial\mathbf{w}_{i,j}}\right| \leq O\left(\frac{|\mathbf{a}_i|B_{x\infty}}{n^{\frac{1}{3}}}\right). \tag{489}$$

$\square$

**Lemma E.36** (Existence of Good Networks under Empirical Risk. Modified version of Lemma D.13 Under Uniform Parity Setting). *Suppose* $n > \Omega\left(\left(\frac{B_x}{\sqrt{B_{x2}}} + \log\frac{1}{p} + \frac{B_{x\infty}}{B_G|\ell'(0)|} + \frac{B_{x\infty}}{B_{G1}|\ell'(0)|}\right)^3 + (\log(d))^6\right)$. *Let* $\lambda^{(1)} = \frac{1}{\eta^{(1)}}$. *For any* $B_\epsilon \in (0, B_b)$, *let* $\sigma_a = \Theta\left(\frac{\tilde{b}}{-|\ell'(0)|\eta^{(1)}B_G B_\epsilon}\right)$ *and* $\delta = 2re^{-\sqrt{\frac{mp}{2}}} + \frac{1}{d^2}$. *Then, with probability at least* $1 - \delta$ *over the initialization and training samples, there exists* $\tilde{\mathbf{a}}_i$'s *such that* $f_{(\tilde{\mathbf{a}},\mathbf{W}^{(1)},\mathbf{b})}(\mathbf{x}) = \sum_{i=1}^{4m}\tilde{\mathbf{a}}_i\sigma\left(\left\langle\mathbf{w}_i^{(1)},\mathbf{x}\right\rangle - \mathbf{b}_i\right)$ *satisfies*

$$\mathcal{L}_{\mathcal{D}}(f_{(\tilde{\mathbf{a}},\mathbf{W}^{(1)},\mathbf{b})}) \tag{490}$$
$$\leq rB_{a1}\left(\frac{2B_{x1}B_{G1}B_b}{\sqrt{mp}B_G B_\epsilon} + \sqrt{2\log(d)d}\left(\gamma_\infty + O\left(\frac{B_{x\infty}}{B_G|\ell'(0)|n^{\frac{1}{3}}}\right)\right) + B_\epsilon\right) + \mathrm{OPT}_{d,r,B_F,S^\infty_{p,\gamma,B_G,B_{G1}}},$$

*and* $\|\tilde{\mathbf{a}}\|_0 = O\left(r(mp)^{\frac{1}{2}}\right)$, $\|\tilde{\mathbf{a}}\|_2 = O\left(\frac{B_{a2}B_b}{\tilde{b}(mp)^{\frac{1}{4}}}\right)$, $\|\tilde{\mathbf{a}}\|_\infty = O\left(\frac{B_{a1}B_b}{\tilde{b}(mp)^{\frac{1}{2}}}\right)$.

*Proof of Lemma E.36.* Denote $\rho = O\left(\exp\Theta\left(-n^{\frac{1}{3}}\right)\right)$ and $\beta = O\left(\frac{B_{x\infty}}{n^{\frac{1}{3}}}\right)$. Note that by symmetric initialization, we have $\ell'(yf_{\Xi^{(0)}}(\mathbf{x})) = |\ell'(0)|$ for any $\mathbf{x} \in \mathcal{X}$, so that, by Lemma E.35, we have $\left|\widetilde{G}(\mathbf{w}_i^{(0)},\mathbf{b}_i)_j - G(\mathbf{w}_i^{(0)},\mathbf{b}_i)_j\right| \leq \frac{\beta}{|\ell'(0)|}$ with probability at least $1 - \rho$. Thus, by union bound, we can see that $S^\infty_{p,\gamma_\infty,B_G,B_{G1}} \subseteq \widetilde{S}^\infty_{p-\rho,\gamma_\infty+\frac{\beta}{B_G|\ell'(0)|},B_G-\frac{\beta}{|\ell'(0)|},B_{G1}+\frac{\beta}{|\ell'(0)|}}$. Consequently, we have $\mathrm{OPT}_{d,r,B_F,\widetilde{S}^\infty_{p-\rho,\gamma_\infty+\frac{\beta}{B_G|\ell'(0)|},B_G-\frac{\beta}{|\ell'(0)|},B_{G1}+\frac{\beta}{|\ell'(0)|}}} \leq \mathrm{OPT}_{d,r,B_F,S^\infty_{p,\gamma_\infty,B_G,B_{G1}}}$. Exactly follow the proof in Lemma D.4 by replacing $S^\infty_{p,\gamma_\infty,B_G,B_{G1}}$ to $\widetilde{S}^\infty_{p-\rho,\gamma_\infty+\frac{\beta}{B_G|\ell'(0)|},B_G-\frac{\beta}{|\ell'(0)|},B_{G1}+\frac{\beta}{|\ell'(0)|}}$. Then, we finish the proof by $\rho \leq \frac{p}{2}$, $\frac{\beta}{|\ell'(0)|} \leq (1 - 1/\sqrt{2})B_G$, $\frac{\beta}{|\ell'(0)|} \leq (\sqrt{2} - 1)B_{G1}$. $\square$

**Theorem E.37** (Online Convex Optimization under Empirical Risk. Modified version of Theorem D.17 Under Uniform Parity Setting ). *Consider training by Algorithm 1, and any* $\delta \in (0, 1)$. *Assume* $d \geq \log m, \delta \leq O(\frac{1}{d^2})$. *Set*

$$\sigma_w > 0, \quad \tilde{b} > 0, \quad \eta^{(t)} = \eta, \quad \lambda^{(t)} = 0 \text{ for all } t \in \{2, 3, \dots, T\},$$
$$\eta^{(1)} = \Theta\left(\frac{\min\{O(\eta), O(\eta\tilde{b})\}}{-\ell'(0)(B_{x1}\sigma_w\sqrt{d} + \tilde{b})}\right), \quad \lambda^{(1)} = \frac{1}{\eta^{(1)}}, \quad \sigma_a = \Theta\left(\frac{\tilde{b}(mp)^{\frac{1}{4}}}{-\ell'(0)\eta^{(1)}B_{x1}\sqrt{B_G B_b}}\right).$$

*Let* $0 < T\eta B_{x1} \leq o(1)$, $m = \Omega\left(\frac{1}{\sqrt{\delta}} + \frac{1}{p}\left(\log\left(\frac{r}{\delta}\right)\right)^2\right)$ *and* $n > \Omega\left(\left(\frac{B_x}{\sqrt{B_{x2}}} + \log\frac{Tm}{p\delta} + (1 + \frac{1}{B_G} + \frac{1}{B_{G1}})\frac{B_{x\infty}}{|\ell'(0)|}\right)^3\right)$. *With probability at least* $1 - \delta$ *over the*

*initialization and training samples, there exists $t \in [T]$ such that*

$$\mathcal{L}_{\mathcal{D}}\left(f_{\Xi^{(t)}}\right) \tag{491}$$

$$\leq \text{OPT}_{d,r,B_F,S_{p,\gamma,B_G}} + rB_{a1}\left(\frac{2\sqrt{2}\sqrt{B_{x1}B_{G1}}}{(mp)^{\frac{1}{4}}}\sqrt{\frac{B_b}{B_G}} + \sqrt{2\log(d)d}\left(\gamma_\infty + O\left(\frac{B_{x\infty}}{B_G|\ell'(0)|n^{\frac{1}{3}}}\right)\right)\right)$$

$$+ \eta\left(\sqrt{r}B_{a2}B_b T\eta B_{x1}^2 + m\tilde{b}\right)O\left(\frac{\sqrt{\log m}B_{x1}(mp)^{\frac{1}{4}}}{\sqrt{B_b B_G}} + 1\right) + O\left(\frac{B_{a2}^2 B_b^2}{\eta T\tilde{b}^2(mp)^{\frac{1}{2}}}\right) \tag{492}$$

$$+ \frac{1}{n^{\frac{1}{3}}}O\left(\left(\frac{rB_{a1}B_b}{\tilde{b}} + m\left(\frac{\tilde{b}\sqrt{\log m}(mp)^{\frac{1}{4}}}{\sqrt{B_b B_G}} + \frac{\tilde{b}}{B_{x1}}\right)\right) \tag{493}$$

$$\cdot\left(\left(\frac{\tilde{b}\sqrt{\log m}(mp)^{\frac{1}{4}}}{\sqrt{B_b B_G}} + T\eta^2 B_{x1}\tilde{b}\right)B_x + \tilde{b}\right) + 2\right) \tag{494}$$

$$+ \frac{1}{n^{\frac{1}{3}}}O\left(m\eta\left(\frac{\tilde{b}\sqrt{\log m}(mp)^{\frac{1}{4}}}{\sqrt{B_b B_G}} + T\eta^2 B_{x1}\tilde{b}\right)\sqrt{B_{x2}}\right). \tag{495}$$

*Furthermore, for any $\epsilon \in (0,1)$, set*

$$\tilde{b} = \Theta\left(\frac{B_G^{\frac{1}{4}}B_{a2}B_b^{\frac{3}{4}}}{\sqrt{r}B_{a1}}\right), \quad m = \Omega\left(\frac{1}{p\epsilon^4}\left(rB_{a1}\sqrt{B_{x1}B_{G1}}\sqrt{\frac{B_b}{B_G}}\right)^4 + \frac{1}{\sqrt{\delta}} + \frac{1}{p}\left(\log\left(\frac{r}{\delta}\right)\right)^2\right),$$

$$\eta = \Theta\left(\frac{\epsilon}{\left(\frac{\sqrt{r}B_{a2}B_b B_{x1}}{(mp)^{\frac{1}{4}}} + m\tilde{b}\right)\left(\frac{\sqrt{\log m}B_{x1}(mp)^{\frac{1}{4}}}{\sqrt{B_b B_G}} + 1\right)}\right), \quad T = \Theta\left(\frac{1}{\eta B_{x1}(mp)^{\frac{1}{4}}}\right),$$

$$n = \Omega\left(\left(\frac{mB_x B_{a2}^2\sqrt{B_b}(mp)^{\frac{1}{2}}\log m}{\epsilon r B_{a1}\sqrt{B_G}}\right)^3 + \left(\frac{B_x}{\sqrt{B_{x2}}} + \log\frac{Tm}{p\delta} + (1 + \frac{1}{B_G} + \frac{1}{B_{G1}})\frac{B_{x\infty}}{|\ell'(0)|}\right)^3\right),$$

*we have there exists $t \in [T]$ with*

$$\Pr[\text{sign}(f_{\Xi^{(t)}})(\mathbf{x}) \neq y] \leq \mathcal{L}_{\mathcal{D}}\left(f_{\Xi^{(t)}}\right) \tag{496}$$

$$\leq \text{OPT}_{d,r,B_F,S_{p,\gamma_\infty,B_G,B_{G1}}^\infty} + rB_{a1}\sqrt{2\log(d)d}\left(\gamma_\infty + O\left(\frac{B_{x\infty}}{B_G|\ell'(0)|n^{\frac{1}{3}}}\right)\right) + \epsilon. \tag{497}$$

*Proof of Theorem E.37.* Proof of the theorem and parameter choices remain the same as Theorem D.17 except for setting $B_\epsilon = \frac{\sqrt{B_{x1}B_{G1}}}{(mp)^{\frac{1}{4}}}\sqrt{\frac{B_b}{B_G}}$ and apply Lemma E.36. $\qquad\square$

### E.6.2 Feature Learning of Uniform Parity Functions

We denote

$$g_{i,j} = \mathbb{E}_{(\mathbf{x},y)}\left[y\sigma'\left[\left\langle\mathbf{w}_i^{(0)},\mathbf{x}\right\rangle - \mathbf{b}_i\right]\mathbf{x}_j\right] \tag{498}$$

$$\xi_k = (-1)^{\frac{k-1}{2}}\frac{\binom{\frac{n-1}{2}-1}{\frac{k-1}{2}}}{\binom{n-1}{k-1}}\cdot 2^{-(n-1)}\binom{n-1}{\frac{n-1}{2}}. \tag{499}$$

**Lemma E.38** (Uniform Parity Functions: Gradient Feature Learning. Corollary of Lemma 3 in [15])**.** *Assume that $n \geq 2(k+1)^2$. Then, the following holds:*

*If $j \in A$, then*

$$g_{i,j} = \xi_{k-1}\prod_{l \in A\setminus\{j\}}(\mathbf{w}_{i,l}^{(0)}). \tag{500}$$

*If $i \notin A$, then*

$$g_{i,j} = \xi_{k-1} \prod_{l \in A \cup \{j\}} (\mathbf{w}_{i,l}^{(0)}). \tag{501}$$

**Lemma E.39** (Uniform Parity Functions: Existence of Good Networks (Alternative)). *Assume the same condition as in Lemma E.38. Define*

$$D = \frac{\sum_{l \in A} \mathbf{M}_l}{\| \sum_{l \in A} \mathbf{M}_l \|_2} \tag{502}$$

*and*

$$f^*(\mathbf{x}) = \sum_{i=0}^{k} (-1)^i \sqrt{k} \tag{503}$$

$$\cdot \left[ \sigma \left( \langle D, \mathbf{x} \rangle - \frac{2i - k - 1}{\sqrt{k}} \right) - 2\sigma \left( \langle D, \mathbf{x} \rangle - \frac{2i - k}{\sqrt{k}} \right) + \sigma \left( \langle D, \mathbf{x} \rangle - \frac{2i - k + 1}{\sqrt{k}} \right) \right].$$

*For $\mathcal{D}_{parity-uniform}$ setting, we have $f^* \in \mathcal{F}_{d,3(k+1),B_F,S^\infty_{p,\gamma_\infty,B_G,B_{G1}}}$ where $B_F = (B_{a1}, B_{a2}, B_b) = \left( 2\sqrt{k}, 2\sqrt{(k(k+1))}, \frac{k+1}{\sqrt{k}} \right)$, $p = \Theta\left( \frac{1}{2^{k-1}} \right)$, $\gamma_\infty = O\left( \frac{\sqrt{k}}{d-k} \right)$, $B_G = \Theta(B_{G1}) = \Theta(d^{-k})$ and $B_{x_1} = \sqrt{d}$, $B_{x_2} = d$. We also have $\mathrm{OPT}_{d,3(k+1),B_F,S^\infty_{p,\gamma_\infty,B_G,B_{G1}}} = 0$.*

*Proof of Lemma E.39.* Fix index $i$, with probability $p_1 = \Theta(2^{-k})$, we will have $\mathbf{w}_{i,j}^{(0)} = \mathrm{sign}(\mathbf{a}_i^{(0)}) \cdot \mathrm{sign}(\xi_{k-1})$, for $\forall j$. For $\mathbf{w}_i^{(0)}$ that satisfy these conditions, we will have:

$$\mathrm{sign}(\mathbf{a}_i^{(0)})g_{i,j} = |\xi_{k-1}|, \quad \forall j \in A \tag{504}$$

$$\mathrm{sign}(\mathbf{a}_i^{(0)})g_{i,j} = |\xi_{k+1}|, \quad \forall j \notin A. \tag{505}$$

Then by Lemma 4 in [15], we have

$$\left\| \frac{\mathrm{sign}(\mathbf{a}_i^{(0)})G(\mathbf{w}_i^{(0)}, \tilde{b})}{\|G(\mathbf{w}_i^{(0)}, \tilde{b})\|} - D \right\|_\infty \leq \max \left\{ \left| \frac{1}{k\sqrt{\frac{1}{k} + \frac{1}{d-k}}} - \frac{1}{\sqrt{k}} \right|, \left| \frac{1}{(d-k)\sqrt{\frac{1}{k} + \frac{1}{d-k}}} \right| \right\} \tag{506}$$

$$\leq \frac{\sqrt{k}}{d-k} \tag{507}$$

*and*

$$\|\mathrm{sign}(\mathbf{a}_i^{(0)})G(\mathbf{w}_i^{(0)}, \tilde{b})\|_2 = \sqrt{k|\xi_{k-1}|^2 + (d-k)|\xi_{k+1}|^2} = \Theta(d^{\Theta(k)}). \tag{508}$$

From here, we can see that if we set $\gamma_\infty = \frac{\sqrt{k}}{d-k}$, $B_G = B_{G1} = \sqrt{k|\xi_{k-1}|^2 + (d-k)|\xi_{k+1}|^2}$, $p = p_1$, we will have $(D, +1), (D, -1) \in S^\infty_{p,\gamma_\infty,B_G,B_{G1}}$ by our symmetric initialization. As a result, we have $f^* \in \mathcal{F}_{d,3(k+1),B_F,S^\infty_{p,\gamma_\infty,B_G,B_{G1}}}$. Finally, it is easy to verify that $f^*(\mathbf{x}) = \mathrm{XOR}(\mathbf{x}_A)$, thus $\mathrm{OPT}_{d,3(k+1),B_F,S^\infty_{p,\gamma_\infty,B_G,B_{G1}}} = 0$. $\square$

**Theorem E.40** (Uniform Parity Functions: Main Result (Alternative)). *For $\mathcal{D}_{parity-uniform}$ setting, for any $\delta \in (0, 1)$ satisfying $\delta \leq O(\frac{1}{d^2})$ and for any $\epsilon \in (0, 1)$ when*

$$m = \mathrm{poly}\left( \log\left( \frac{1}{\delta} \right), \frac{1}{\epsilon}, 2^{\Theta(k)}, d \right), T = \Theta\left( d^{\Theta(k)} \right), n = \Theta\left( d^{\Theta(k)} \right) \tag{509}$$

*trained by Algorithm 1 with hinge loss, with probability at least $1 - \delta$ over the initialization, with proper hyper-parameters, there exists $t \in [T]$ such that*

$$\mathrm{Pr}[\mathrm{sign}(f_{\Xi^{(t)}}(\mathbf{x})) \neq y] \leq \frac{k^2 \sqrt{d \log(d)}}{d - k} + \epsilon. \tag{510}$$

*Proof of Theorem E.40.* Plug the values of parameters into Theorem E.37 and directly get the result. $\square$

### E.7 Multiple Index Model with Low Degree Polynomial

#### E.7.1 Problem Setup

The multiple-index data problem has been used for studying network learning [18, 32]. We consider proving guarantees for the setting in [32], following our framework. We use the properties of the problem to prove the key lemma (i.e., the existence of good networks) in our framework and then derive the final guarantee from our theorem of the simple setting (Theorem 3.4).

**Data Distributions.** We draw input from the distribution $\mathcal{D}_{\mathcal{X}} = \mathcal{N}(0, I_{d \times d})$, and we assume the target function is $g^*(\mathbf{x}) : \mathbb{R}^d \to \mathbb{R}$, where $g^*$ is a degree $\tau$ polynomial normalized so that $\mathbb{E}_{\mathbf{x} \sim \mathcal{D}_{\mathcal{X}}}[g^*(\mathbf{x})^2] = 1$.

**Assumption E.41.** *There exists linearly independent vectors $u_1, \ldots, u_r$ such that $g^*(\mathbf{x}) = g(\langle \mathbf{x}, u_1 \rangle, \ldots, \langle \mathbf{x}, u_r \rangle)$. $H := \mathbb{E}_{\mathbf{x} \sim \mathcal{D}_{\mathcal{X}}}[\nabla^2 g^*(\mathbf{x})]$ has rank $r$, where $H$ is a Hessian matrix.*

**Definition E.42.** *Denote the normalized condition number of $H$ by*

$$\kappa := \frac{\|H^\dagger\|}{\sqrt{r}}. \tag{511}$$

**Initialization and Loss.** For $\forall i \in [m]$, we use the following initialization:

$$\mathbf{a}_i^{(0)} \sim \{-1, 1\}, \quad \mathbf{w}_i^{(0)} \sim \mathcal{N}\left(0, \frac{1}{d} I_{d \times d}\right) \quad \text{and} \quad \mathbf{b}_i = 0. \tag{512}$$

For this regression problem, we use mean square loss:

$$\mathcal{L}_{\mathcal{D}_{\mathcal{X}}}(f_\Xi) = \mathbb{E}_{\mathbf{x} \sim \mathcal{D}_{\mathcal{X}}}\left[(f_\Xi(\mathbf{x}) - g^*(\mathbf{x}))^2\right]. \tag{513}$$

**Training Process.** We use the following one-step training algorithm for this specific data distribution.

---

**Algorithm 5** Network Training via Gradient Descent [32]. Special case of Algorithm 2

---

Initialize $(\mathbf{a}^{(0)}, \mathbf{W}^{(0)}, \mathbf{b})$ as in Equation (8) and Equation (512); Sample $\mathcal{Z} \sim \mathcal{D}_{\mathcal{X}}^n$
$\rho = \frac{1}{n} \sum_{\mathbf{x} \in \mathcal{Z}} g^*(\mathbf{x})$, $\beta = \frac{1}{n} \sum_{\mathbf{x} \in \mathcal{Z}} g^*(\mathbf{x})\mathbf{x}$
$y = g^*(\mathbf{x}) - \rho - \beta \cdot \mathbf{x}$
$\mathbf{W}^{(1)} = \mathbf{W}^{(0)} - \eta^{(1)}(\nabla_{\mathbf{W}} \widetilde{\mathcal{L}}_{\mathcal{Z}}(f_{\Xi^{(0)}}) + \lambda^{(1)} \mathbf{W}^{(0)})$
Re-initialize $\mathbf{b}_i \sim \mathcal{N}(0, 1)$
**for** $t = 2$ **to** $T$ **do**
    $\mathbf{a}^{(t)} = \mathbf{a}^{(t-1)} - \eta^{(t)} \nabla_{\mathbf{a}} \widetilde{\mathcal{L}}_{\mathcal{Z}}(f_{\Xi^{(t-1)}})$
**end for**

---

**Lemma E.43** (Multiple Index Model with Low Degree Polynomial: Existence of Good Networks. Rephrase of Lemma 25 in [32])**.** *Assume $n \geq d^2 r \kappa^2 (C_l \log(nmd))^{\tau+1}$, $d \geq C_d \kappa r^{3/2}$, and $m \geq r^\tau \kappa^{2\tau}(C_l \log(nmd))^{6\tau+1}$ for sufficiently large constants $C_d, C_l$, and let $\eta^{(1)} = \sqrt{\frac{d}{(C_l \log(nmd))^3}}$ and $\lambda^{(1)} = \frac{1}{\eta^{(1)}}$. Then with probability $1 - \frac{1}{\text{poly}(m,d)}$, there exists $\tilde{\mathbf{a}} \in \mathbb{R}^m$ such that $f_{(\tilde{\mathbf{a}}, \mathbf{W}^{(1)}, \mathbf{b})}$ satisfies*

$$\mathcal{L}_{\mathcal{D}_{\mathcal{X}}}\left(f_{(\tilde{\mathbf{a}}, \mathbf{W}^{(1)}, \mathbf{b})}\right) \leq O\left(\frac{1}{n} + \frac{r^\tau \kappa^{2\tau}(C_l \log(nmd))^{6\tau+1}}{m}\right) \tag{514}$$

*and*

$$\|\tilde{\mathbf{a}}\|_2^2 \leq O\left(\frac{r^\tau \kappa^{2\tau}(C_l \log(nmd))^{6\tau}}{m}\right). \tag{515}$$

#### E.7.2 Multiple Index Model: Final Guarantee

Considering training by Algorithm 5, we have the following results.

**Theorem E.44** (Multiple Index Model with Low Degree Polynomial: Main Result). *Assume $n \geq \Omega\left(d^2 r \kappa^2 (C_l \log(nmd))^{\tau+1} + m\right)$, $d \geq C_d \kappa r^{3/2}$, and $m \geq \Omega\left(\frac{1}{\epsilon} r^\tau \kappa^{2\tau} (C_l \log(nmd))^{6\tau+1}\right)$ for sufficiently large constants $C_d, C_l$. Let $\eta^{(1)} = \sqrt{\frac{d}{(C_l \log(nmd))^3}}$ and $\lambda^{(1)} = \frac{1}{\eta^{(1)}}$, and $\eta = \eta^{(t)} = \Theta(m^{-1})$, for all $t \in \{2, 3, \ldots, T\}$. For any $\epsilon \in (0, 1)$, if $T \geq \Omega\left(\frac{m^2}{\epsilon}\right)$, then with properly set parameters and Algorithm 5, with high probability that there exists $t \in [T]$ such that*

$$\mathcal{L}_{\mathcal{D}_\mathcal{X}} f_{(\mathbf{a}^{(t)}, \mathbf{W}^{(1)}, \mathbf{b})} \leq \epsilon. \tag{516}$$

*Proof of Theorem E.44.* By Lemma E.43, we have for properly chosen hyper-parameters,

$$\mathrm{OPT}_{\mathbf{W}^{(1)}, \mathbf{b}, B_{a2}} \leq \mathcal{L}_{\mathcal{D}_\mathcal{X}}\left(f_{(\tilde{\mathbf{a}}, \mathbf{W}^{(1)}, \mathbf{b})}\right) \leq O\left(\frac{1}{n} + \frac{r^\tau \kappa^{2\tau} (C_l \log(nmd))^{6\tau+1}}{m}\right) \tag{517}$$

$$\leq \frac{\epsilon}{3}. \tag{518}$$

We compute the $L$-smooth constant of $\widetilde{\mathcal{L}}_\mathcal{Z}\left(f_{(\mathbf{a}, \mathbf{W}^{(1)}, \mathbf{b})}\right)$ to $\mathbf{a}$.

$$\left\|\nabla_\mathbf{a} \widetilde{\mathcal{L}}_\mathcal{Z}\left(f_{(\mathbf{a}_1, \mathbf{W}^{(1)}, \mathbf{b})}\right) - \nabla_\mathbf{a} \widetilde{\mathcal{L}}_\mathcal{Z}\left(f_{(\mathbf{a}_2, \mathbf{W}^{(1)}, \mathbf{b})}\right)\right\|_2 \tag{519}$$

$$= \left\|\frac{1}{n} \sum_{\mathbf{x} \in \mathcal{Z}} \left[2\left(f_{(\mathbf{a}_1, \mathbf{W}^{(1)}, \mathbf{b})}(\mathbf{x}) - g^* - f_{(\mathbf{a}_2, \mathbf{W}^{(1)}, \mathbf{b})}(\mathbf{x}) + g^*\right) \sigma(\mathbf{W}^{(1)\top}\mathbf{x} - \mathbf{b})\right]\right\|_2 \tag{520}$$

$$\leq \left\|\frac{1}{n} \sum_{\mathbf{x} \in \mathcal{Z}} \left[2\left(\mathbf{a}_1^\top \sigma(\mathbf{W}^{(1)\top}\mathbf{x} - \mathbf{b}) - \mathbf{a}_2^\top \sigma(\mathbf{W}^{(1)\top}\mathbf{x} - \mathbf{b})\right) \sigma(\mathbf{W}^{(1)\top}\mathbf{x} - \mathbf{b})\right]\right\|_2 \tag{521}$$

$$\leq \frac{1}{n} \sum_{\mathbf{x} \in \mathcal{Z}} \left[2\|\mathbf{a}_1 - \mathbf{a}_2\|_2 \left\|\sigma(\mathbf{W}^{(1)\top}\mathbf{x} - \mathbf{b})\right\|_2^2\right]. \tag{522}$$

By the proof of Lemma 25 in [32], we have for $\forall i \in [4m]$, with probability at least $1 - \frac{1}{\mathrm{poly}(m,d)}$, $|\langle \mathbf{w}_i, \mathbf{x}\rangle| \leq 1$, with some large polynomial $\mathrm{poly}(m, d)$. As a result, we have

$$\frac{1}{n} \sum_{\mathbf{x} \in \mathcal{Z}} \left\|\mathbf{W}^{(1)\top}\mathbf{x}\right\|_2^2 \leq m + \frac{1}{\mathrm{poly}(m, d)} \leq O(m). \tag{523}$$

Thus, we have,

$$L = O\left(\frac{1}{n} \sum_{\mathbf{x} \in \mathcal{Z}} \left\|\sigma(\mathbf{W}^{(1)\top}\mathbf{x} - \mathbf{b})\right\|_2^2\right) \tag{524}$$

$$\leq O\left(\frac{1}{n} \sum_{\mathbf{x} \in \mathcal{Z}} \left\|\mathbf{W}^{(1)\top}\mathbf{x} - \mathbf{b}\right\|_2^2\right) \tag{525}$$

$$\leq O\left(\frac{1}{n} \sum_{\mathbf{x} \in \mathcal{Z}} \left\|\mathbf{W}^{(1)\top}\mathbf{x}\right\|_2^2 + \|\mathbf{b}\|_2^2\right) \tag{526}$$

$$\leq O(m). \tag{527}$$

This means that we can let $\eta = \Theta\left(m^{-1}\right)$ and we will get our convergence result. We can bound $\|\mathbf{a}^{(1)}\|_2$ and $\|\tilde{\mathbf{a}}\|_2$ by $\|\mathbf{a}^{(1)}\|_2 = O(\sqrt{m})$ and $\|\tilde{\mathbf{a}}\|_2 = O\left(\frac{r^\tau \kappa^{2\tau} (C_l \log(nmd))^{6\tau}}{m}\right) = O(\epsilon)$. So, if we choose $T \geq \Omega\left(\frac{m}{\epsilon\eta}\right)$, there exists $t \in [T]$ such that $\widetilde{\mathcal{L}}_\mathcal{Z}\left(f_{(\mathbf{a}^{(t)}, \mathbf{W}^{(1)}, \mathbf{b})}\right) - \widetilde{\mathcal{L}}_\mathcal{Z}\left(f_{(\tilde{\mathbf{a}}, \mathbf{W}^{(1)}, \mathbf{b})}\right) \leq O\left(\frac{L\|\mathbf{a}^{(1)} - \tilde{\mathbf{a}}\|_2^2}{T}\right) \leq \epsilon/3$.

We also have $\sqrt{\frac{\|\tilde{\mathbf{a}}\|_2^2(\|\mathbf{W}^{(1)}\|_F^2 B_x^2 + \|\mathbf{b}\|_2^2)}{n}} \leq \frac{\epsilon}{3}$. Then our theorem gets proved by Theorem 3.4. $\qquad\square$

**Discussion.** We would like to unify [32], whcih are very closely related to our framework: their analysis for multiple index data follows the same principle and analysis approach as our general framework, although it does not completely fit into our Theorem 3.12 due to some technical differences. We can cover it with our Theorem 3.4.

Our work and [32] share the same principle and analysis approach. [32] shows that the first layer learns good features by one gradient step update, which can approximate the true labels by a low-degree polynomial function. Then, a classifier (the second layer) is trained on top of the learned first layer which leads to the final guarantees. This is consistent with our framework: we first show that the first layer learns good features by one gradient step update, which can approximate the true labels, and then show a good classifier can be learned on the first layer.

Our work and [32] have technical differences. First, in the second stage, [32] fix the first layer and only update the top layer which is a convex optimization. Our framework allows updates in the first layer and uses online convex learning techniques for the analysis. Second, they consider the square loss (this is used to calculate Hermite coefficients explicitly for gradients, which are useful in the low-degree polynomial function approximation). While in our online convex learning analysis, we need boundedness of the derivative of the loss to show that the first layer weights' changes are bounded in the second stage. Given the above two technicalities, we analyze their training algorithm (Algorithm 2) which fixes the first layer weights and fits into our Theorem 3.4.

# F  Auxiliary Lemmas

In this section, we present some Lemmas used frequently.

**Lemma F.1** (Lemmas on Gradients).

$$\nabla_{\mathbf{W}} \mathcal{L}_{(\mathbf{x},y)}(f_\Xi) = \left[ \frac{\partial \mathcal{L}_{(\mathbf{x},y)}(f_\Xi)}{\partial \mathbf{w}_1}, \dots, \frac{\partial \mathcal{L}_{(\mathbf{x},y)}(f_\Xi)}{\partial \mathbf{w}_i}, \dots, \frac{\partial \mathcal{L}_{(\mathbf{x},y)}(f_\Xi)}{\partial \mathbf{w}_{4m}} \right], \tag{528}$$

$$\frac{\partial \mathcal{L}_{(\mathbf{x},y)}(f_\Xi)}{\partial \mathbf{w}_i} = \mathbf{a}_i \ell'(y f_\Xi(\mathbf{x})) y \left[ \sigma' \left( \langle \mathbf{w}_i, \mathbf{x} \rangle - \mathbf{b}_i \right) \right] \mathbf{x}, \tag{529}$$

$$\nabla_{\mathbf{W}} \mathcal{L}_{\mathcal{D}}(f_\Xi) = \left[ \frac{\partial \mathcal{L}_{\mathcal{D}}(f_\Xi)}{\partial \mathbf{w}_1}, \dots, \frac{\partial \mathcal{L}_{\mathcal{D}}(f_\Xi)}{\partial \mathbf{w}_i}, \dots, \frac{\partial \mathcal{L}_{\mathcal{D}}(f_\Xi)}{\partial \mathbf{w}_{4m}} \right], \tag{530}$$

$$\frac{\partial \mathcal{L}_{\mathcal{D}}(f_\Xi)}{\partial \mathbf{w}_i} = \mathbf{a}_i \mathbb{E}_{(\mathbf{x},y)} \left[ \ell'(y f_\Xi(\mathbf{x})) y \left[ \sigma' \left( \langle \mathbf{w}_i, \mathbf{x} \rangle - \mathbf{b}_i \right) \right] \mathbf{x} \right], \tag{531}$$

$$\frac{\partial \mathcal{L}_{\mathcal{D}}(f_\Xi)}{\partial \mathbf{a}_i} = \mathbb{E}_{(\mathbf{x},y)} \left[ \ell'(y f_\Xi(\mathbf{x})) y \left[ \sigma \left( \langle \mathbf{w}_i, \mathbf{x} \rangle - \mathbf{b}_i \right) \right] \right]. \tag{532}$$

*Proof.* These can be verified by direct calculation. $\square$

**Lemma F.2** (Property of Symmetric Initialization). *For any* $\mathbf{x} \in \mathbb{R}^d$*, we have* $f_{\Xi^{(0)}}(\mathbf{x}) = 0$ *. For all* $i \in [2m]$*, we have* $\mathbf{w}_i^{(1)} = -\mathbf{w}_{i+2m}^{(1)}$*. When input data is symmetric, i.e,* $\mathbb{E}_{(\mathbf{x},y)}[y\mathbf{x}] = \mathbf{0}$*, for all* $i \in [m]$*, we have* $\mathbf{w}_i^{(1)} = \mathbf{w}_{i+m}^{(1)}$*.*

*Proof of Lemma F.2.* By symmetric initialization, we have $f_{\Xi^{(0)}}(\mathbf{x}) = 0$. For all $i \in [2m]$, we have

$$\mathbf{w}_i^{(1)} = -\eta^{(1)} \ell'(0) \mathbf{a}_i^{(0)} \mathbb{E}_{(\mathbf{x},y)} \left[ y \sigma' \left[ \left\langle \mathbf{w}_i^{(0)}, \mathbf{x} \right\rangle - \mathbf{b}_i \right] \mathbf{x} \right] \tag{533}$$

$$= \eta^{(1)} \ell'(0) \mathbf{a}_{i+2m}^{(0)} \mathbb{E}_{(\mathbf{x},y)} \left[ y \sigma' \left[ \left\langle \mathbf{w}_{i+2m}^{(0)}, \mathbf{x} \right\rangle - \mathbf{b}_{i+2m} \right] \mathbf{x} \right] \tag{534}$$

$$= -\mathbf{w}_{i+2m}^{(1)}. \tag{535}$$

When $\mathbb{E}_{(\mathbf{x},y)}[y\mathbf{x}] = \mathbf{0}$, for all $i \in [m]$, we have

$$\mathbf{w}_i^{(1)} = -\eta^{(1)}\ell'(0)\mathbf{a}_i^{(0)}\mathbb{E}_{(\mathbf{x},y)}\left[y\sigma'\left[\left\langle\mathbf{w}_i^{(0)},\mathbf{x}\right\rangle - \mathbf{b}_i\right]\mathbf{x}\right] \tag{536}$$

$$=\eta^{(1)}\ell'(0)\mathbf{a}_{i+m}^{(0)}\mathbb{E}_{(\mathbf{x},y)}\left[y\sigma'\left[\left\langle-\mathbf{w}_{i+m}^{(0)},\mathbf{x}\right\rangle + \mathbf{b}_{i+m}\right]\mathbf{x}\right] \tag{537}$$

$$=\eta^{(1)}\ell'(0)\mathbf{a}_{i+m}^{(0)}\mathbb{E}_{(\mathbf{x},y)}\left[y\sigma'\left[\left\langle-\mathbf{w}_{i+m}^{(0)},\mathbf{x}\right\rangle + \mathbf{b}_{i+m}\right]\mathbf{x} - y\mathbf{x}\right] \tag{538}$$

$$=\eta^{(1)}\ell'(0)\mathbf{a}_{i+m}^{(0)}\mathbb{E}_{(\mathbf{x},y)}\left[-y\sigma'\left[\left\langle\mathbf{w}_{i+m}^{(0)},\mathbf{x}\right\rangle - \mathbf{b}_{i+m}\right]\mathbf{x}\right] \tag{539}$$

$$=\mathbf{w}_{i+m}^{(1)}. \tag{540}$$

$\square$

**Lemma F.3** (Property of Direction Neighborhood). *If $\mathbf{w} \in \mathcal{C}_{D,\gamma}$, we have $\rho\mathbf{w} \in \mathcal{C}_{D,\gamma}$ for any $\rho \neq 0$. We also have $\mathbf{0} \notin \mathcal{C}_{D,\gamma}$. Also, if $(D,s) \in S_{p,\gamma,B_G}$, we have $(-D,s) \in S_{p,\gamma,B_G}$.*

*Proof.* These can be verified by direct calculation. $\square$

**Lemma F.4** (Maximum Gaussian Tail Bound). *$M_n$ is the maximum of $n$ i.i.d. standard normal Gaussian. Then*

$$\Pr\left(M_n \geq \sqrt{2\log n} + \frac{z}{\sqrt{2\log n}}\right) \leq e^{-z}. \tag{541}$$

*Proof.* These can be verified by direct calculation. $\square$

**Lemma F.5** (Chi-squared Tail Bound). *If $X$ is a $\chi^2(k)$ random variable. Then, $\forall z \in \mathbb{R}$, we have*

$$\Pr(X \geq k + 2\sqrt{kz} + 2z) \leq e^{-z}. \tag{542}$$

*Proof.* These can be verified by direct calculation. $\square$

**Lemma F.6** (Gaussian Tail Bound). *If $g$ is standard Gaussian and $z > 0$, we have*

$$\frac{1}{\sqrt{2\pi}}\frac{z}{z^2+1}e^{-z^2/2} < \Pr_{g\sim\mathcal{N}(0,1)}[g > z] < \frac{1}{\sqrt{2\pi}}\frac{1}{z}e^{-z^2/2}. \tag{543}$$

*Proof.* These can be verified by direct calculation. $\square$

**Lemma F.7** (Gaussian Tail Expectation Bound). *If $g$ is standard Gaussian and $z \in \mathbb{R}$, we have*

$$|\mathbb{E}_{g\sim\mathcal{N}(0,1)}[\mathbb{I}[g > z]g]| < 2\Pr_{g\sim\mathcal{N}(0,1)}[g > z]^{0.9}. \tag{544}$$

*Proof of Lemma F.7.* For any $p \in (0,1)$, we have

$$\left|\int_{-\infty}^{\sqrt{2}\text{erf}^{-1}(2p-1)}\frac{e^{-\frac{x^2}{2}}x}{\sqrt{2\pi}}dx\right| < 2p^{0.9}, \tag{545}$$

where $\sqrt{2}\text{erf}^{-1}(2p-1)$ is the quantile function of the standard Gaussian. We finish the proof by replacing $p$ to be $\Pr_{g\sim\mathcal{N}(0,1)}[g > z]$. $\square$

**Lemma F.8.** *If a function $g$ satisfy $h(n+2) = 2h(n+1) - (1-\rho^2)h(n) + \beta$ for $n \in \mathbb{N}_+$ where $\rho, \beta > 0$, then $h(n) = -\frac{\beta}{\rho^2} + c_1(1-\rho)^n + c_2(1+\rho)^n$, where $c_1, c_2$ only depends on $h(1)$ and $h(2)$.*

*Proof.* These can be verified by direct calculation. $\square$

**Lemma F.9** (Rademacher Complexity Bounds. Rephrase of Lemma 48 in [32]). *For fixed $\mathbf{W}, \mathbf{b}$, let $\mathcal{F} = \{f_{(\mathbf{a},\mathbf{W},\mathbf{b})} : \|\mathbf{a}\| \leq B_{a2}\}$. Then,*

$$\Re(\mathcal{F}) \leq \sqrt{\frac{B_{a2}^2(\|\mathbf{W}\|_F^2 B_x^2 + \|\mathbf{b}\|_2^2)}{n}}. \tag{546}$$

