# OpenReview forum: "Provable Guarantees for Neural Networks via Gradient Feature Learning"
_NeurIPS.cc/2023/Conference — NeurIPS 2023 poster_

### Official Review · Reviewer_Sj27 · 2023-07-01

**Soundness:** 4 excellent
**Presentation:** 4 excellent
**Contribution:** 4 excellent
**Rating:** 7
**Confidence:** 3

**Summary:**

This paper proposes a general framework for analyzing feature learning in two-layer ReLU neural networks. The idea is to consider the class of two-layer ReLU networks with “gradient features”, i.e. features aligned with the gradients of the loss induced by the distributions of the data and initial model parameters. The main result (Theorem 3.12) is that gradient descent on the two-layer ReLU network achieves generalization error close to the that of the optimal model in this class under weak assumptions. Instantiations of this result are provided for the special cases of the data being generated by a mixture of gaussians and parity functions. In these settings, the optimal loss among models in the gradient feature class is computed, with final generalization rates strictly improving over kernel methods and matching or surpassing the best known results in the feature learning literature.

**Strengths:**

1. The paper analyzes a highly relevant topic.

2. The idea is simple, yet powerful and novel, to my knowledge. The proposed notion of gradient features can likely be used to show feature learning guarantees for models beyond two-layer ReLUs. At present, it allows for competitive feature learning guarantees with very general boundedness assumptions on the data distribution and smoothness of the loss function.

3. The two instantiations of the main result are very helpful to concretize the significance of the main result. Both are very well-studied settings, and the provided guarantees strictly improve upon kernel methods in both cases, and match the rates for learning parities albeit with more general assumptions.

4. The presentation is clear and rigorous.

5. The related works are well-covered.


**Weaknesses:**

1. On its own, it is difficult to gauge the significance of the main result, since in general we do not know whether gradient features can lead to a good model. Indeed, it would be helpful to have more discussion of when gradient features can lead to a good model beyond the two special cases.

2. Related to above, a result showing that if the optimal model in the gradient feature class has large risk, then the ground-truth mapping from inputs to labels is not learnable by gradient descent would strengthen the paper.

3. Like other feature learning studies, the analysis leverages feature learning happening in the early steps of gradient descent, while subsequent updates are shown only to not corrupt the learned features, although it is unclear how well this aligns with practice.

**Questions:**

Minor note: the restrictions on $\tau$ should appear in the statements of Theorems 4.4 and 4.8.

**Limitations:**

Yes

---

> ### Author Rebuttal · Authors · 2023-08-09
>
> We thank the reviewer for providing thorough suggestions! For feature learning happening in the early steps of gradient descent, please refer to the global response above. Below we address the other comments. In short, we provide some failure cases that our framework cannot cover.
>
> ### More discussion
> In the Appendix, we also provide a linear data model and multiple-index data models. We admit that for general data distributions without detailed information, we may not know whether the gradient feature is good. In our case study, we can build a “ground-truth” network on gradient features. However, for arbitrary data distribution or labeling functions, the “ground-truth” networks may not exist. See examples in the Failure case below.
>
> On the other hand, one cannot hope for non-vacuous bounds for general problems, given the various hardness results of network learning on general problems, e.g., [1,2]. We agree that given a general problem, we may not have an easy way to compute the “complexity’’ quantity of the problem to get guarantees. This by itself is an interesting and challenging question: given a general problem, is it possible to determine if network learning can have non-vacuous guarantees? We conjecture a negative answer, but this is beyond the scope of the current work and left for future study. See more discussion in the global response above.
>
> ### Failure case
> There are two failure cases we can think of currently:
> - In  [3], they constructed a function that is easy to approximate using a 3-layer network but not approximable by any 2-layer network. Since the function is not approximable by any 2-layer network, it cannot be approximated by the gradient-induced networks as well, so OPT will be large. As a result, the final error will be large.
> - In uniform parity data distribution, considering an odd number of features rather than even, i.e., $k$ is an odd number in Assumption E.30, we can show that our gradient feature set is empty even when $p$ in Equation (5) is exponentially small, thus the OPT is a positive constant since the gradient induced network can only be constants. Meanwhile, the neural network won’t be able to learn this data distribution because its gradient is always 0 through the training, and the final error equals OPT.
> - The above two cases give examples that if the optimal model in the gradient feature class has a large risk, then the ground-truth mapping from inputs to labels is not learnable by gradient descent.
>
> [1] Daniely, A., & Vardi, G. (2020). Hardness of learning neural networks with natural weights.
>
> [2] Daniely, A., Srebro, N., & Vardi, G. (2023). Efficiently Learning Neural Networks: What Assumptions May Suffice?
>
> [3] Safran, I., Eldan, R., & Shamir, O. (2019). Depth separations in neural networks: what is actually being separated?

---

> > ### Comment · Reviewer_Sj27 · 2023-08-18
> >
> > Thank you to the authors for your response. I have decided to maintain my score, and encourage the authors to add the failure cases to the final version, perhaps in the appendix.

---

> > > ### Author Response · Authors · 2023-08-18
> > > **Thank you**
> > >
> > > Thank you for your comments and suggestions! We will add the failure cases to our revision. Thank you for your time!

---

### Official Review · Reviewer_H9VP · 2023-07-06

**Soundness:** 3 good
**Presentation:** 2 fair
**Contribution:** 3 good
**Rating:** 5
**Confidence:** 3

**Summary:**

This paper proposes a general framework of feature learning for two-layer neural networks trained with gradient descent. This framework covers a variety of classification losses and data distributions. Specifically, the authors establish that the loss of neural networks trained with gradient descent is comparable to that of networks with first layers in the direction of a simplified gradient and optimal second layers. The framework is then specialized to multiple examples, including Gaussian mixture classification and learning parity functions.

**Strengths:**

* The presented framework is rather general and unifies the approaches of several recent works, while covering distributions that were not previously covered in the literature.
* The studied problem is exciting and significant for the community as the existence of such a framework is necessary given the ad-hoc nature of current feature learning analyses.
* The related literature is covered extensively.


**Weaknesses:**

* The general guarantees provided by the framework (Theorem 3.12) are limited to the performance of the optimal network with first-layer directions approximately aligned with the gradient. Such a statement does not directly imply feature learning for general problems/distributions, as a key part of feature learning seems to be showing that the gradient directions are indeed useful. To that end, the guarantees are more or less similar to prior work and suboptimal in some cases (e.g. Gaussian mixtures).

* The sample complexities of Theorems 4.4 and 4.8 are only stated as polynomials in problem parameters, thus while outperforming kernel methods, provide limited insight in comparison with prior works. Furthermore, the $\tilde{O}(d^{1.5})$ sample complexity to learn an XOR label from a mixture of 4 Gaussians in Section 4.1.1 seems to be suboptimal, as [92] shows a sample complexity of $O(d)$ for learning the same problem.

* The presentation of the paper can be improved, with specific examples given below.

**Questions:**

* Is it possible to characterize the reason behind the suboptimality in the $\tilde{\mathcal{O}}(d^{1.5})$ sample complexity of this work in comparison with the $O(d)$ sample complexity of [92] to learn a mixture of 4 Gaussians with XOR-like structure?

* A number of figures can be added to improve the presentation of the paper, e.g. for the definition of gradient features and introducing the Gaussian mixtures and the input distribution for learning parity functions.

* Including the choices of step size in Theorem 3.12 can provide additional insights. Specifically, it seems like the theorem is still operating in the regime of a few gradient steps for the first layer, similar to prior work. A more explicit discussion of the training of the first layer would clarify the similarities and distinctions with prior work.

**Limitations:**

The authors have mostly discussed the limitations of their work. It would help the readers better understand the limitations if there is also a discussion on how realistic the training assumption for the first layer is (i.e. only a few steps or many steps), and also the potential suboptimality in the rate of learning mixtures of Gaussians in comparison with prior work.

---

> ### Author Rebuttal · Authors · 2023-08-09
>
> We thank the reviewer for providing thorough suggestions! For the theorem still operating in the regime of a few gradient steps for the first layer, please refer to the global response above. Below we address the other comments. In short, we can **improve** our results from $\tilde{O}(d^{1.5})$ to be $\tilde{O}(d)$ in sample size under the setting of a mixture of 4 Gaussians with an XOR-like structure.
>
> ### Theorem 3.12 are limited to the performance of the optimal network with first-layer directions approximately aligned with the gradient
>
> We agree that our framework cannot directly imply guarantees for general problems/distributions. We want to emphasize that our key contributions are the concept of gradient features and the idea of using networks with gradient features as baselines to quantify the learning errors. We view the current work as a first step in exploiting the full power of these ideas. Even when we use the framework with the current analysis, we can already unify the prototypical questions/analyses in existing work and also obtain interesting insights. This shows the great potential of the framework. See more discussion in the global response above.
>
>
> ### Suboptimal results in the mixture of Gaussians
>
> [92] use ODE to simulate the optimization process for the 2NN learning XOR-shaped Gaussian mixture and give convincing evidence that $O(d)$ number of samples is enough to learn the XOR-shaped Gaussian mixture, yet they did not give a rigorous convergence guarantee for this problem. We successfully derived a convergence guarantee while we required a slightly larger sample size $\tilde{O}(d^{1.5})$.
>
> Moreover, we can improve our results from $\tilde{O}(d^{1.5})$ to be $\tilde{O}(d)$ in sample size. Note that in Lemma D.17 Equation (153) and (154), it still holds if we choose $z = {\log n \over n^{-{1 \over 2}}}$. Then, all $n^{- {1\over 3}}$ in sample size will change to $ {\log n \over n^{-{1 \over 2}}}$. The probability term will change from $O(\exp(- n^{1\over 3}))$ to $O({1 \over n^2})$. Then, we can still get the final guarantee with $\tilde{O}(d)$ in a mixture of 4 Gaussians with an XOR-like structure. In our original submission, we do not provide the tightest bound, we will update it in our paper.
>
> On the other hand, as we mentioned in the limitation section, our framework may or may not recover the width or sample complexity bounds in existing work. This is because our work is analyzing general cases, and thus may not give better than or the same bounds as those in special cases since special cases have more properties that can be exploited to get potentially better bounds.
>
> We proposed the two key ideas of gradient feature and gradient feature-induced neural networks not only to show their ability to unify several current works but also to open a new direction of thinking with respect to the learning process. These notations have the potential to be extended to multi-layer gradient features and multi-step learning, and this work is only our first step.
>
> ### Figures
> Great suggestion! We made a plot about Gradient Feature under Mixture of Gaussians data in the  **[additional rebuttal pdf](https://openreview.net/attachment?id=BMPAso6Sns&name=pdf)**. We will plot more figures in the revision.

---

> > ### Comment · Reviewer_H9VP · 2023-08-13
> >
> > Thank you for your detailed response and for the nice figure! I believe that this work presents an effective framework to unify recent approaches, however, I'm keeping my original score as I'm not entirely certain how much of the high-level intuitions obtained from this work are significantly novel in comparison to the existing literature.

---

> > > ### Author Response · Authors · 2023-08-13
> > > **Thank you**
> > >
> > > We appreciate that the reviewer believes our work presents an effective framework to unify recent approaches. We will update our work with new bound for the 4-mixture-of-Gaussians, more figures, and more insights. Thank you for your time and valuable suggestions!

---

### Official Review · Reviewer_3J3m · 2023-07-07

**Soundness:** 4 excellent
**Presentation:** 3 good
**Contribution:** 3 good
**Rating:** 7
**Confidence:** 3

**Summary:**

This paper introduces a general framework for studying feature learning in two-layer NNs. This framework covers feature learning in different examples such as linear classification, mixture of Gaussians and parity functions (it also gives some intuition about the learned features). The neural network under study has one hidden layer and is initialized in a symmetric manner. The training algorithm used is mini-batch GD with fresh samples. In the first iteration, features are learned in some hidden neurons. During the next iterations both layers get updated, however the change in the hidden layer is controlled.



**Strengths:**

- The paper does not freeze the first layer's weights, instead it trains all parameters together and controls the change in the first layer's weights with small learning rate.
- The literature review is quite extensive; the paper can generally help people interested in feature learning to become familiar with this direction of research. The paper's framework has also been applied to several different learning problems.
- I personally liked the connection to the lottery ticket hypothesis. I'd suggest moving (more of) it to the main.

**Weaknesses:**

- There are a few limitations which are quite common in the current literature of deep learning theory and have been discussed in the paper: e.g., the feature learning is actually done in the first iteration of gradient descent. See below for some questions regarding the limitations of the framework.
- Personally I wonder if there are new insights given by this framework or not? (Also see below)

**Questions:**

- Q1. Is it possible to extend the results to the continuous setting and $\ell_2$ loss function (at the beginning $\ell_2$ also acts like $-y\hat{y}$)
- Q2. What are the general insights given by the framework (other than the unification)?
- Q3. Is it possible to use other activation functions?
- Q4. Why in Theorems 33 and 46 the algorithm that trains both layers is not used?

Suggestions:
- It would be really great if some intuition about the gradient feature learning framework can be provided (maybe even in the abstract).
- The parity formulation in section 4.2 was difficult to understand for me; maybe some changes would help.

**Limitations:**

This is a theoretical work and there is not negative societal impact. I think the limitations have been discussed adequately (nonetheless see above).

---

> ### Author Rebuttal · Authors · 2023-08-08
>
> We thank the reviewer for providing thorough suggestions! For the limitation of the first gradient descent learning and Q2 General insights, please refer to the global response above. Below we address the other comments.
>
> ### Q1 Continous setting and square loss
> It is possible to extend to a continuous setting and this is one of the works in progress. We can define a gradient feature distribution rather than a gradient feature set. However, we find the technical tools used in the continuous setting are pretty different from the discrete version.
>
> For the $\ell\_2$ loss, it works well in the early-stage where feature learning happens. However, in the later stage analysis (online convex learning part), we need to show that the hidden layer weights stay in a small neighborhood while the top layer weights get updated to a good solution. For the squared loss, it may still be possible to do this control. The current argument easily bounds the change for logistic/hinge losses, but it cannot be directly adopted in the case of squared loss (due to its faster growth). We conjecture a more careful step-by-step inductive argument being needed for that case.
>
>
> ### Q3 Activation functions
> Yeah, we can change the ReLU activation function to a sub-linear activation function, e.g., leaky ReLU, sigmoid, to get a similar conclusion. First, we need to introduce a corresponding gradient feature set and then we can make it by following the same analysis pipeline. For simplicity, we present ReLU only.
>
>
> ### Q4 Theorems 33 and 46
> For Theorem 33, we provided Theorem 42 as an alternative version that trains both layers. We provide Theorem 33 because (1) use it as a warm-up and (2) follow the original analysis in [1] to give a comparison.
>
> For Theorem 46: this is because we would like to **unify** previous work. [2] are very closely related to our framework: their analysis for multiple index data follows the same principle and analysis approach as our general framework, although it does not completely fit into our Theorem 3.12 due to some technical differences. We can cover it with our Theorem 3.4.
> - The same principle and analysis approach: [2] shows that the first layer learns good features by one gradient step update, which can approximate the true labels by a low-degree polynomial function. Then, a classifier (the second layer) is trained on top of the learned first layer which leads to the final guarantees. This is consistent with our framework: we first show that the first layer learns good features by one gradient step update, which can approximate the true labels, and then show a good classifier can be learned on the first layer.
> - Technical differences: First, in the second stage, [2] fix the first layer and only update the top layer which is a convex optimization. Our framework allows updates in the first layer and uses online convex learning techniques for the analysis. Second, they consider the square loss (this is used to calculate Hermite coefficients explicitly for gradients, which are useful in the low-degree polynomial function approximation). While in our online convex learning analysis, we need boundedness of the derivative of the loss to show that the first layer weights’ changes are bounded in the second stage. Given the above two technicalities, we analyze their training algorithm (Algorithm 2) which fixes the first layer weights, which currently do not fit directly into our Theorem 3.12 but can fit into Theorem 3.4.
>
> ### Parity formulation
> We will polish the formulation to make the setting more clear. We provide a high-level intuition here. We have $r$ parity functions each corresponding to a block of $k$ dimensions;  $\mathcal{X}\_{j,+}$ and $\mathcal{X}\_{j,-}$ stands for the component providing a strong signal for the $j$-th parity; $\mathcal{X}\_U$ corresponds to uniform distribution unrelated to any parity and providing weak learning signal; $A^\perp$ is the noise part. The label depends on the sum of the $r$ parity functions.
>
> [1] Barak, B., Edelman, B., Goel, S., Kakade, S., Malach, E., & Zhang, C. (2022). Hidden progress in deep learning: Sgd learns parities near the computational limit.
>
> [2] Damian, A., Lee, J., & Soltanolkotabi, M. (2022). Neural networks can learn representations with gradient descent.

---

> > ### Comment · Reviewer_3J3m · 2023-08-16
> >
> > Thanks for the response. I think the paper can benefit from including some of these discussions. I will keep my score.

---

> > > ### Author Response · Authors · 2023-08-16
> > > **Thank you**
> > >
> > > Thank you for your positive and helpful feedback! We will make sure to include these discussions in the revision.

---

### Official Review · Reviewer_zxbj · 2023-07-10

**Soundness:** 4 excellent
**Presentation:** 2 fair
**Contribution:** 3 good
**Rating:** 6
**Confidence:** 3

**Summary:**

The paper defines the concept of "gradient features" which capture the features that the network can learn after one step of gradient descent. The paper then instantiates this framework to prove optimization and generalization guarantees for various statistical learning problems.

**Strengths:**

- The paper develops a general framework which formalizes the "one step" feature learning trick which has recently become popular. This provides an easy to use framework which makes it easy to derive sharp sample complexity guarantees for a number of well defined statistical learning problems.

- The paper instantiates this framework in a number of settings (mixtures of Gaussians, parity, multi-index models) and re-derives a number of sample complexity results.

**Weaknesses:**

- It appears that this framework can only handle features that can be learned directly at initialization (i.e. through the "one-step trick"). In particular, it seems unable to handle multi-step feature learning (e.g. the merged staircase property in [1]).

[1] Abbe et al. (2022) "The merged-staircase property: a necessary and nearly sufficient condition for SGD learning of sparse functions on two-layer neural networks"

**Questions:**

- What is the role of the $s$ in the gradient feature? It seems to encode the sign of $b$ but I don't immediately see why this is important.

- For simple $k$-parity, my understanding is that the learned gradient feature $D_1$ is $1_{A}$, the indicator function for the subset $A$. This would therefore compute parity by computing the parity of $\sum_{i \in A} x_i$. Is this correct?

**Limitations:**

The authors adequately addressed the limitations of this paper.

---

> ### Author Rebuttal · Authors · 2023-08-08
>
> We thank the reviewer for providing thorough suggestions! For the "one-step trick" question, please refer to the global response above. Below we address the other comments.
>
> ### More about multi-step feature learning
> We would like to mention that the early-stage analysis (“one-step trick”) is an important and necessary foundation for multi-step analysis. In our analysis, we show that when training time is at a certain range (denoted as $T$), during online convex optimization, the first layer weights will stay in a small regime while the second layer weights will converge to a good classifier based on these gradient features. However, the NN may improve its performance if we continue training NN beyond $T$. Our gradient features are defined based on initialization, i.e., $f^{(0)}$. It is natural to think about whether we can define a new gradient feature set upon $f^{(T)}$ and whether we can get a better guarantee based on the new gradient feature set. If the answer is yes, then we could provide a framework with multi-stage/step feature learning.
>
> In [1], they introduce a very sophisticated data distribution whose properties are exploited to address the above challenges smartly. The analysis of multi-step feature learning in a general framework is still open and will be our future work, as we mentioned in our Conclusion section “While the current framework focuses on the gradient features in the early gradient steps, whether feature learning also happens in later steps and if so how to formalize that?”.
>
> ### Role of $s$
> Yes, the $s$ encodes the sign of the bias term, which is important. Recall that we do not update the bias term for simplicity. Let’s consider a simple toy example. Assume we have $f_1(x) = a_1 ReLU(w_1^\top x + 1)$, $f_2(x) = a_2 ReLU(w_2^\top x - 1)$ and $f_3(x) = a_3 ReLU(w_3^\top x + 2)$.
> - The sign of the bias term is important. We can see that we always have $a_1 ReLU(w_1^\top x + 1) \neq a_2 ReLU(w_2^\top x - 1)$ for any $a_1, w_1, a_2, w_2$. It means that $f_1(x)$ and $f_2(x)$ are intrinsically different and have different active patterns. Thus, we need to handle the sign of the bias term carefully.
> - The scaling of the bias is absorbed. On the other hand, we can see that $a_1 ReLU(w_1^\top x + 1) = a_3 ReLU(w_3^\top x + 2)$ when $a_1 = 2 a_3, 2w_1 = w_3$. It means the scale of the bias term is less important, which can be absorbed into other terms.
>
> Thus, we only need to handle bias with different signs carefully.
>
> ### $k$-parity
> Yes, it is correct that $D_1$ is $1_A$, the indicator function for the subset $A$ and we build the optimal neural network based on such directions.
>
> [1] Abbe, E., Adsera, E. B., & Misiakiewicz, T. (2022). The merged-staircase property: a necessary and nearly sufficient condition for sgd learning of sparse functions on two-layer neural networks.

---

> > ### Comment · Reviewer_zxbj · 2023-08-15
> >
> > Thank you for the clarifications. I have decided to keep my score.

---

> > > ### Author Response · Authors · 2023-08-16
> > > **Thank you**
> > >
> > > Thank you for reading our response! We are pleased that our response addressed your questions.

---

### Author Rebuttal · Authors · 2023-08-08

We thank all reviewers for their constructive and valuable feedback.

We are glad that all reviewers unanimously agree that our theoretical analysis is novel, exciting, powerful, and significant. Reviewers find that our paper provides an easy-to-use, general, and unified framework (zxbj, H9VP, Sj27) that makes it easy to derive sharp sample complexity guarantees strictly improve upon kernel methods (zxbj, 3J3m, Sj27). Reviewers believe that our framework is necessary given the ad-hoc nature of current feature learning analyses (3J3m, H9VP). We are encouraged that reviewers agree that the paper can generally help people interested in feature learning to become familiar with this direction of research (3J3m, H9VP, Sj27).

Here we address the early-stage feature learning problem raised by all reviewers and provide more insights about our framework. We will address the other comments in individual responses to each reviewer.

We agree that early-stage feature learning is a limitation of many latest feature learning analysis papers (zxbj), and we emphasize that our framework provides new insights including some for potentially going beyond the early-stage feature learning. The **challenges in the later-stage analysis** are: (1) the weights in the later stage will not be as normal as the initialization, and we need new tools to analyze their properties; (2) to show that the later-stage features eventually lead to a good solution, we may need new analysis tools for the nonconvex optimization due to the changes in the first layer weights.

On the other hand, when building the general framework, we indeed (1) pin down the key principle behind learning over different data distributions and (2) get new insights.

1. Our framework articulates the following key principles (pointed out for specific problems in existing work but not articulated more generally):
- **Role of gradient**: The gradient leads to the emergence of good features, which is useful for the learning of upper layers in later stages.
- **From features to solutions**: Learned features in early steps will not be distorted, if not improved, in later stages. The training dynamic for upper layers will eventually learn a good combination of hidden neurons based on gradient features, giving a good solution.
2. Some other interesting insights are obtained from the generality of the framework:
- To build a general framework, the meaningful error guarantees should be data-dependent, since NN learning on general data distributions is hard and data-independent guarantees will be vacuous. Comparing the optimal in a family of “ground-truth’’ functions (inspired by agnostic learning in learning theory) is a useful method to obtain the data-dependent bound. We further construct the “ground-truth’’ functions using properties of the training dynamics, i.e., gradient features. This greatly facilitates the analysis of the training dynamics and is the key to obtaining the final guarantees.
- The framework can also be viewed as **using the optimal by gradient-induced NN to measure the “complexity’’** of the problem. For easier problems, this quantity is smaller, and our framework can give a better error bound. So this provides a united way to derive guarantees for specific problems.
- It is important to validate the effectiveness of a general framework by applying it to prototypical problems. Such applications can also help clarify what’s crucial or less crucial in existing analyses for specific problems.
- For an SGD-optimized NN, its **actual representation power** is from the subset of NN based on gradient features, instead of the whole set of NN. This view helps explain the simplicity bias/implicit regularization phenomenon of NN learning in practice.
- Our framework goes **beyond NTK** as we use features from gradients rather than features from random initialization. It means features from gradients are more powerful.
3. More broadly, our framework may give new perspectives about roadmaps forward.
- We argue a new perspective about the connection between the strong representation power and the successful learning of NN. **Traditionally, the strong representation power of NN is the key reason for hardness results of NN learning**: NN has strong representation power and can encode hard learning questions, so they are hard to learn. See the proof in SQ bound from [1] or NP-hardness from [2]. The strong representation power also causes trouble for the statistical aspect: it leads to vacuous generalization bounds when traditional uniform convergence tools are used.
- **Our framework suggests a new perspective in sharp contrast: the strong representation power of NN with gradient features is actually the key to successful learning**. More concretely, the optimal error of the gradient feature-induced NN being small (i.e., strong representation power for a given data distribution) can lead to a small guarantee, which is the key to successful learning.
- The above new perspective suggests a different analysis road than traditional ones. Traditional analysis typically first reasons about the optimal based on the whole function class, i.e. the ground truth, then analyze how NN learns proper features and reaches the optimal. In contrast, our framework defines feature family first, and then reasons about the optimal based on it.
- Our framework provides the foundation for future work on analyzing gradient-based NN learning, which may inspire future directions including but not limited to (1) defining a new feature family for 2-layer NN rather than gradient feature, (2) considering deep NN and introducing new gradient features (e.g., gradient feature notion for upper layers), (3) defining different gradient feature family at different training stages (e.g., gradient feature notion for later stages).

[1] Daniely, A., & Malach, E. (2020). Learning parities with neural networks.

[2] Blum, A., & Rivest, R. (1988). Training a 3-node neural network is NP-complete.

---

### Decision · Program_Chairs · 2023-09-21

**Decision:**

Accept (poster)

**Comment:**

The paper unifies and extends several prior work on feature learning using the "one-step" large batch gradient descent trick via the concept of "gradient features". The key idea is to quantify the useful features that one large step of gradient descent can recover from initialization. On the technical front, the analysis follows similar analyses in the literature with the added extension of training both layers in the second phase of learning. The paper also connects the observations to LTH and simplicity bias among others which are interesting.

The main limitations of the current work is the one-step analysis compared to the multi-step analysis that have been performed for certain settings in prior work (staircase/multi-index), and the lack of clarity on what new insights this framework can have beyond the few examples discussed in the paper (which were somewhat understood before). Despite these limitations, the reviewers and I unanimously agree that this paper provides a useful unification which future work can rely on to build more complex analyses. Therefore, I recommend accepting the paper. I encourage the authors to incorporate the improved bounds for the mixture of Gaussians case, the plot from the rebuttal, and other relevant discussion points that came up in the exchanges.